# IMPROVING LLM UNLEARNING ROBUSTNESS VIA RANDOM PERTURBATIONS

## ABSTRACT

Here, we show that current state-of-the-art LLM unlearning methods *inherently reduce models' robustness*, causing them to misbehave even when a single non-adversarial forget-token is present in the retain-query. Toward understanding underlying causes, we propose a novel theoretical framework that reframes the *unlearning process as backdoor attacks and defenses*: forget-tokens act as backdoor triggers that, when activated in retain-queries, cause disruptions in unlearned models' behaviors, similar to successful backdoor attacks. The sense that, LLM unlearning methods *themselves poison the model*, make it more vulnerable to forget-tokens, and *hide rather than erase* target knowledge, describes their true mechanism. To mitigate the vulnerability caused by the forgetting process, we reinterpret the retaining process as a backdoor defense and propose Random Noise Augmentation (RNA), a lightweight, model and method-agnostic approach with theoretical guarantees for improving the robustness of models. Extensive experiments demonstrate that RNA significantly improves the robustness of unlearned models while preserving forget and retain performances. This backdoor attack-defense framework offers insights into the mechanism of unlearning that can shed light on future research directions for improving unlearning robustness.

## 1 INTRODUCTION

Modern LLMs are pre-trained on massive text corpora and then post-trained with Reinforcement Learning from Human Feedback (Christiano et al., 2017; Ziegler et al., 2019; Stiennon et al., 2020; Ouyang et al., 2022) to be helpful and harmless (Bai et al., 2022). Recent studies have shown that despite safety enhancements, aligned LLMs can still exhibit harmful and undesirable behaviors, such as generating toxic content (Wen et al., 2023), producing copyrighted material (Karamolegkou et al., 2023; Eldan & Russinovich, 2023; Wei et al., 2024; Cooper et al., 2025), bias (Belrose et al., 2024), leaking sensitive and private information (Nasr et al., 2025; Patil et al., 2024), and potentially aiding malicious uses such as cyberattacks, chemical attacks, and bioweapons development (Fang et al., 2024; Sandbrink, 2023; Li et al., 2024). As LLMs advance in size and capabilities at an unprecedented speed, concerns about their potential risks continue to grow.

Machine Unlearning (MU; Cao & Yang (2015); Bourtoule et al. (2021); Nguyen et al. (2022); Xu et al. (2023)) is an approach aiming to *robustly* (1) remove specific target knowledge in a forget-set and capabilities from a pre-trained model, while (2) retaining the model's other knowledge in a retain-set and capabilities. Recent works on the robustness of unlearning methods primarily focus on the first criterion, evaluating the robustness of unlearned models against knowledge recovery that adversarially tries to recover unlearned knowledge. For example, previously unlearned knowledge is shown to resurface through relearning (Li et al., 2024; Deeb & Roger, 2024; Lo et al., 2024), sequential unlearning (Shi et al., 2025), target relearning attacks (Hu et al., 2025), removing or steering specific directions in the latent space (Łucki et al., 2025; Seyitoğlu et al.), quantization (Zhang et al., 2025), or even simply fine-tuning on unrelated tasks (Doshi & Stickland, 2024; Łucki et al., 2025).

However, the equally important criterion of *robustly preserving the model's general knowledge*—that is, ensuring stable and accurate responses to retain-queries even when they inadvertently include forget-tokens—remains underexplored. Initial steps have been taken, such as Thaker et al. (2025), who examined the robustness of Representation Misdirection for Unlearning (RMU; Li et al. (2024)), demonstrating that RMU-unlearned models are fragile when asked with retain-queries (*e.g.*, Q&A

about general knowledge) containing forget-tokens (tokens in the forget-set). However, many critical questions remain unanswered. In this paper, we make the following contributions:

① We first draw a connection between the current two widely used classes of LLM unlearning methods, including Representation Misdirection (RM) and Preference Optimization (PO), through a unified view of the generative latent variable model. Inspired by this view, we present an analysis to show that current unlearning methods *inherently reduce the model robustness*, in the sense that they can be misbehaved even when a *single non-adversarial* forget-token appears in the retain-query.

② We propose a novel perspective that decomposes the unlearning process into "forgetting" and "retaining" processes and reframes it as a *backdoor attack and defense problem*. The "forgetting" corresponds to a backdoor attack: by treating the forget-set as a poisoned dataset, we formulate how the unlearning methods inadvertently learn to align forget-tokens (backdoor triggers) with the target representations (the target labels). As a result, when forget-tokens appear in a retain-query, it is similar to activating the backdoor trigger, making the model misbehave. To counteract the vulnerability introduced by the "forgetting", we reinterpret the "retaining" as a backdoor defense and introduce *Random Noise Augmentation (RNA)*, a lightweight, model and method-agnostic approach which adds small, independent Gaussian noise to each retain-query's representation during fine-tuning to reduce the model's sensitivity to forget-tokens.

③ Through theoretical and empirical analysis, we show that RNA significantly improves the robustness of unlearned models while maintaining original forget and retain performances.

## 2    RELATED WORKS AND PRELIMINARIES

**Machine Unlearning.** MU has become one of the most important tools for ensuring the safety and protecting the privacy of LLMs (Xu et al., 2023; Nguyen et al., 2022; Barez et al., 2025). Most recent works on MU focus on developing algorithms for different tasks, domains, and settings (Pawelczyk et al., 2024; Thaker et al., 2024; Jin et al., 2024; Shi et al., 2025; Choi et al., 2024; Pal et al., 2025; Muhamed et al., 2025; Wang et al., 2025b; Kuo et al., 2025; Zhuang et al., 2025), while much less effort was spent on developing robust unlearning algorithms.

**Unlearning Robustness.** Previous works on MU robustness focus on "forget-robustness", studying the robustness of MU algorithms in making the model forget the target knowledge and capabilities. Researchers showed that unlearned knowledge can resurface through re-learning (Li et al., 2024; Lynch et al., 2024; Barez et al., 2025; Lo et al., 2024), sequential unlearning (Shi et al., 2025), quantization (Zhang et al., 2025), fine-tuning unlearned models on unrelated tasks (Doshi & Stickland, 2024; Łucki et al., 2025), and adversarial attacks (Hu et al., 2025; Yuan et al., 2025a; Shumailov et al., 2024; Huang et al., 2025; Wu et al., 2025) and developed methods for improving forget-robustness of MU algorithms (Sheshadri et al., 2024; Tamirisa et al., 2025; 2024; Fan et al., 2025; Zhang et al., 2025; Wang et al., 2025a). This paper explores the "retain-robustness" of MU algorithms, studying the robustness of MU algorithms in *robustly retaining the original model's general knowledge and capabilities*. Thaker et al. (2025) presented preliminary results showing that state-of-the-art MU algorithms do not preserve the original model's knowledge and capabilities. We bridge the gap in retain-robustness research by introducing RNA, a simple data augmentation method inspired by adversarial training to improve the robustness of MU algorithms. We formally define the retain-robustness studied in this work below.

**Definition 2.1** (Retain-robustness). The capacity of MU algorithms to preserve the model's general knowledge and capabilities when handling retain-queries that are *inadvertently* contain forget-tokens, *without any intention of adversarially attacking the model* or *closely related to forget-sets*.

**Notation and problem formulation.** The training data of an MU problem consists of two subsets: the forget-set $\mathcal{D}_f$ and the retain-set $\mathcal{D}_r$. The goal is to minimize the model's performance on the forget set while keeping the performance on the retain-set. Let $f_{\boldsymbol{\theta}}$ be a model parameterized by $\boldsymbol{\theta}$, and $\ell(\mathbf{y}|\mathbf{x};\boldsymbol{\theta})$ is the loss of input $\mathbf{x}$ with respect to a target output $\mathbf{y}$ in model $f_{\boldsymbol{\theta}}$. A commonly used form of unlearning involves minimizing the following two-part loss:

$$\mathcal{L}_{\mathcal{D}_f,\mathcal{D}_r,\boldsymbol{\theta}} = \mathbb{E}_{(\mathbf{x}^f,\mathbf{y}^f)\sim\mathcal{D}_f}\left[\ell\left(\mathbf{y}^f|\mathbf{x}^f;\boldsymbol{\theta}\right)\right] + \alpha\mathbb{E}_{(\mathbf{x}^r,\mathbf{y}^r)\sim\mathcal{D}_r}\left[\ell\left(\mathbf{y}^r|\mathbf{x}^r;\boldsymbol{\theta}\right)\right] \tag{1}$$

where $\mathbf{y}^f, \mathbf{y}^r$ are the target outputs of forget and retain input, respectively, $\alpha \in \mathbb{R}^+$ is a retain weight. We consider two widely used classes of LLM unlearning methods, which rely on Representation Misdirection (RM) and Preference Optimization (PO). We denote $|| \cdot ||$ the Euclidean norm.

## 2.1 REPRESENTATION MISDIRECTION

Representation Misdirection (RMU and its variants) is an unlearning approach that conducts unlearning by manipulating latent representations during fine-tuning. Denote $\mathbf{z}_{\boldsymbol{\theta}}^f, \mathbf{z}_{\boldsymbol{\theta}}^r \in \mathbb{R}^{n \times d_l}$ the output hidden state of $n$-tokens in forget-sample $\mathbf{x}^f$ and in retain-sample $\mathbf{x}^r$, respectively, at layer $l$ in model $f_{\boldsymbol{\theta}}$, where $d_l$ is the dimension of hidden states at layer $l$. In this context, the activations from the MLP module in the transformer layer are used as the latent representations.

**Representation Misdirection for Unlearning (RMU; Li et al. (2024))** pushes the latent representation of forget-tokens to a predetermined random representation $\mathbf{y}^f = c\mathbf{u}$, where $\mathbf{u} \in \mathbb{R}^{d_l}$ is a unit vector with each element uniformly sampled from $[0, 1)$, and $c \in \mathbb{R}^+$ is a coefficient. It also regularizes the latent representation of retain-tokens back to the reference model's representation:

$$\mathcal{L}^{\text{RMU}} = \mathbb{E}_{\mathbf{x}^f \sim \mathcal{D}_f} ||\mathbf{z}_{\boldsymbol{\theta}}^f - c\mathbf{u}||^2 + \alpha \mathbb{E}_{\mathbf{x}^r \sim \mathcal{D}_r} ||\mathbf{z}_{\boldsymbol{\theta}}^r - \mathbf{z}_{\boldsymbol{\theta}^{\text{ref}}}^r||^2, \tag{2}$$

where $\boldsymbol{\theta}$ and $\boldsymbol{\theta}^{\text{ref}}$ are the parameters of the updated and reference (frozen weight) models, respectively.

**Adaptive RMU** (Dang et al., 2025) is a variant of RMU that adaptively changes the coefficient of the random vector $\mathbf{u}$ in the forget-loss based on the norm of the forget-sample's representations in the reference model. The target random representation $\mathbf{y}^f = \beta ||\mathbf{z}_{\boldsymbol{\theta}^{\text{ref}}}^f||\mathbf{u}$, $\beta \in \mathbb{R}^+$ is a scaling factor.

**Random Steering Vector (RSV).** Additionally, we implement RSV—a variant of RMU that uses the target random representation $\mathbf{y}^f = \mathbf{z}_{\boldsymbol{\theta}^{\text{ref}}}^f + c\boldsymbol{\epsilon}$, where $c \in \mathbb{R}^+$ is a predetermined coefficient, $\boldsymbol{\epsilon}$ is a random unit vector sampled from Gaussian distribution $\mathcal{N}(\mathbf{0}, \mu\boldsymbol{I})$, $\mu\boldsymbol{I}$ is covariance matrix, $\mu \in \mathbb{R}^+$.

## 2.2 PREFERENCE OPTIMIZATION

**Negative Preference Optimization (NPO; Zhang et al. (2024))** treats forget-samples as negative preference samples in Direct Preference Optimization (DPO; Rafailov et al. (2023)). NPO can be viewed as a gradient ascent variant with adaptive gradient weights that allows more controlled and stable optimization:

$$\mathcal{L}^{\text{NPO}} = \mathbb{E}_{(\mathbf{x}^f, \mathbf{y}^f) \sim \mathcal{D}_f} \left[ -\frac{2}{\beta} \log \sigma \left( -\beta \log \left( \frac{\pi_{\boldsymbol{\theta}}(\mathbf{y}^f|\mathbf{x}^f)}{\pi_{\boldsymbol{\theta}^{\text{ref}}}(\mathbf{y}^f|\mathbf{x}^f)} \right) \right) \right], \tag{3}$$

where $\beta \in \mathbb{R}^+$ is a temperature hyperparameter (NPO reduces to gradient ascent as $\beta \to 0$), $\sigma(\cdot)$ is the sigmoid function, and $\pi_{\boldsymbol{\theta}}(\mathbf{y}^f|\mathbf{x}^f)$, $\pi_{\boldsymbol{\theta}^{\text{ref}}}(\mathbf{y}^f|\mathbf{x}^f)$ denotes the predicted probability of $\mathbf{y}^f$ given $\mathbf{x}^f$ in the model $f_{\boldsymbol{\theta}}$ and reference model $f_{\boldsymbol{\theta}^{\text{ref}}}$ (frozen weight) respectively.

**Simple Negative Preference Optimization (SimNPO; Fan et al. (2024))** simplifies NPO by using a normalized sequence log-probability and introducing a reward margin hyperparameter $\gamma \geq 0$:

$$\mathcal{L}^{\text{SimNPO}} = \mathbb{E}_{(\mathbf{x}^f, \mathbf{y}^f) \sim \mathcal{D}_f} \left[ -\frac{2}{\beta} \log \sigma \left( -\frac{\beta}{|\mathbf{y}^f|} \log \pi_{\theta}(\mathbf{y}^f|\mathbf{x}^f) - \gamma \right) \right], \tag{4}$$

where $|\mathbf{y}^f|$ is the length of output $\mathbf{y}^f$.

**Direct Preference Optimization (DPO).** As a baseline, Zhang et al. (2024); Maini et al. (2024); Yuan et al. (2025b)) adopted standard DPO, using a refusal answer $\mathbf{y}^{\text{idk}} \in \mathcal{D}_{\text{idk}}$ such as "I Don't Know" as the positive samples and forget-samples as negative samples.

To preserve model's general knowledge and capabilities, we use Mean Squared Error (MSE): $\mathcal{L}^{\text{MSE}} = \mathbb{E}_{(\mathbf{x}^r, \mathbf{y}^r) \sim \mathcal{D}_r} ||\log \pi_{\boldsymbol{\theta}}(\mathbf{x}^r) - \log \pi_{\boldsymbol{\theta}^{\text{ref}}}(\mathbf{x}^r)||^2$ or Kullback–Leibler divergence (KL): $\mathcal{L}^{\text{KL}} = \mathbb{E}_{(\mathbf{x}^r, \mathbf{y}^r) \sim \mathcal{D}_r} \text{KL} \left( \log \pi_{\boldsymbol{\theta}}(\mathbf{x}^r), \log \pi_{\boldsymbol{\theta}^{\text{ref}}}(\mathbf{x}^r) \right)$ as the retain-loss. Combining the two losses, we investigate a series of 6 PO-based unlearning methods including NPO+MSE, NPO+KL, DPO+MSE, DPO+KL, SimNPO+MSE, and SimNPO+KL.

## 3 A Unified View of LLM Unlearning

We first draw a connection between RM and PO methods through *a unified view of the generative latent variable model* (GLVM). Let $\mathbf{z}_{\boldsymbol{\theta}}^f + \mathbf{v}$ be the steered latent representation of forget-sample $\mathbf{x}^f$ in $f_{\boldsymbol{\theta}}$ as a result of RM. We assume that random vector $\mathbf{v}$ is small and sampled from normal distribution $\mathcal{N}(\mathbf{0}, \mu \boldsymbol{I})$, $\mu \in \mathbb{R}^+$. We employ the notion of the GLVM, that is, GLVM $f_{\boldsymbol{\theta}}$ generates target output $\mathbf{y}^f$ given the latent variable $\mathbf{z}_{\boldsymbol{\theta}}^f$. Let $\ell(\mathbf{y}^f | \mathbf{z}_{\boldsymbol{\theta}}^f + \mathbf{v}; \boldsymbol{\theta})$ be the loss of generating $\mathbf{y}^f$ given $\mathbf{z}_{\boldsymbol{\theta}}^f + \mathbf{v}$ in model $f_{\boldsymbol{\theta}}$. For simplicity, we write $\ell(\mathbf{y}^f | \mathbf{z}_{\boldsymbol{\theta}}^f + \mathbf{v})$ to present $\ell(\mathbf{y}^f | \mathbf{z}_{\boldsymbol{\theta}}^f + \mathbf{v}; \boldsymbol{\theta})$. Following Koh & Liang (2017), we assume that the loss is twice-differentiable and locally convex. Since $\mathbf{v}$ is small, we approximate the function $\ell(\mathbf{y}^f | \mathbf{z}_{\boldsymbol{\theta}}^f + \mathbf{v})$ using the second-order Taylor approximation:

$$\ell(\mathbf{y}^f | \mathbf{z}_{\boldsymbol{\theta}}^f + \mathbf{v}) \approx \ell(\mathbf{y}^f | \mathbf{z}_{\boldsymbol{\theta}}^f) + \mathbf{v}^\top \nabla_{\mathbf{z}_{\boldsymbol{\theta}}^f} \ell(\mathbf{y}^f | \mathbf{z}_{\boldsymbol{\theta}}^f) + \frac{1}{2} \mathbf{v}^\top \nabla_{\mathbf{z}_{\boldsymbol{\theta}}^f}^2 \ell(\mathbf{y}^f | \mathbf{z}_{\boldsymbol{\theta}}^f) \mathbf{v} \tag{5}$$

Taking the expectation of both sides of Eqn. 5 with respect to $\mathbf{v}$, we obtain:

$$\mathbb{E}_{\mathbf{v}}[\ell(\mathbf{y}^f | \mathbf{z}_{\boldsymbol{\theta}}^f + \mathbf{v})] \approx \mathbb{E}_{\mathbf{v}}[\ell(\mathbf{y}^f | \mathbf{z}_{\boldsymbol{\theta}}^f)] + \mathbb{E}_{\mathbf{v}}[\mathbf{v}^\top \nabla_{\mathbf{z}_{\boldsymbol{\theta}}^f} \ell(\mathbf{y}^f | \mathbf{z}_{\boldsymbol{\theta}}^f)] + \frac{1}{2} \mathbb{E}_{\mathbf{v}}[\mathbf{v}^\top \nabla_{\mathbf{z}_{\boldsymbol{\theta}}^f}^2 \ell(\mathbf{y}^f | \mathbf{z}_{\boldsymbol{\theta}}^f) \mathbf{v}] \tag{6}$$

$$= \ell(\mathbf{y}^f | \mathbf{z}_{\boldsymbol{\theta}}^f) + \nabla_{\mathbf{z}_{\boldsymbol{\theta}}^f} \ell(\mathbf{y}^f | \mathbf{z}_{\boldsymbol{\theta}}^f)^\top \mathbb{E}_{\mathbf{v}}[\mathbf{v}] + \frac{1}{2} \mathbb{E}_{\mathbf{v}}[\mathbf{v}^\top \nabla_{\mathbf{z}_{\boldsymbol{\theta}}^f}^2 \ell(\mathbf{y}^f | \mathbf{z}_{\boldsymbol{\theta}}^f) \mathbf{v}] \tag{7}$$

$$= \ell(\mathbf{y}^f | \mathbf{z}_{\boldsymbol{\theta}}^f) + \frac{1}{2} \mathbb{E}_{\mathbf{v}}[\mathbf{v}^\top \nabla_{\mathbf{z}_{\boldsymbol{\theta}}^f}^2 \ell(\mathbf{y}^f | \mathbf{z}_{\boldsymbol{\theta}}^f) \mathbf{v}], \quad \text{since } \mathbb{E}_{\mathbf{v}}[\mathbf{v}] = \mathbf{0}. \tag{8}$$

A classic result from Hutchinson (1989) tell us that $\mathbb{E}_{\mathbf{v}}[\mathbf{v}^\top \nabla_{\mathbf{z}_{\boldsymbol{\theta}}^f}^2 \ell(\mathbf{y}^f | \mathbf{z}_{\boldsymbol{\theta}}^f) \mathbf{v}] = \mu \mathrm{Tr}(\nabla_{\mathbf{z}_{\boldsymbol{\theta}}^f}^2 \ell(\mathbf{y}^f | \mathbf{z}_{\boldsymbol{\theta}}^f))$, where $\mathrm{Tr}(\nabla_{\mathbf{z}_{\boldsymbol{\theta}}^f}^2 \ell(\mathbf{y}^f | \mathbf{z}_{\boldsymbol{\theta}}^f)) > 0$ is the trace of the positive definite Hessian matrix $\nabla_{\mathbf{z}_{\boldsymbol{\theta}}^f}^2 \ell(\mathbf{y}^f | \mathbf{z}_{\boldsymbol{\theta}}^f)$ (by assumption). Since $\mu \in \mathbb{R}^+$, the loss of generating $\mathbf{y}^f$ given latent variable $\mathbf{z}_{\boldsymbol{\theta}}^f$ is *increases*, that is,

$$\mathbb{E}_{\mathbf{v}}[\ell(\mathbf{y}^f | \mathbf{z}_{\boldsymbol{\theta}}^f + \mathbf{v})] \approx \ell(\mathbf{y}^f | \mathbf{z}_{\boldsymbol{\theta}}^f) + \frac{\mu}{2} \mathrm{Tr}(\nabla_{\mathbf{z}_{\boldsymbol{\theta}}^f}^2 \ell(\mathbf{y}^f | \mathbf{z}_{\boldsymbol{\theta}}^f)) > \ell(\mathbf{y}^f | \mathbf{z}_{\boldsymbol{\theta}}^f) \tag{9}$$

While presenting in different formulations, PO and RM *share a common high-level principle—maximizing the loss of forget-samples*. Therefore, Eqn. 9 suggests that steering forget-representations toward a random representation in RM is effectively equivalent to maximizing the loss of those forget-samples in PO. In other words, PO can be viewed as RM, that is, PO introduces noise-like effects to the forget-representation during fine-tuning, disrupting its alignment with target labels. We present an empirical validation in Appendix C.1.

## 4 Analysis on the Robustness of Unlearned Models

### 4.1 Threat Model

We first define the threat model and the unlearning guarantee that is expected to hold. We consider a practical scenario, such as machine learning as a service (MLaaS), where users can black-box access the unlearned model through an API.

**User's knowledge.** In this setting, users have *no information about the model parameters or training data, only the model's inputs and outputs are exposed.*

**User's query and capability.** Such a situation might happen when users can supply benign retain-queries that fall into two cases: (1) queries are closely related to the forget-sets or (2) queries inadvertently contain forget-tokens, *without any intention of adversarially attacking the model.*

**Model provider's knowledge and capability.** In this setting, the model provider can fully *access and modify* the model weights while *having no information about any specific user's knowledge and intention.*

**Unlearning guarantee.** Unlearned models are expected to be *robust against forget-tokens in retain-queries* while maintaining the forgetting performance on forget-tasks as well as retaining performance on benign retain-queries. The presence of forget-tokens should have *minimal* effects on the model's performance on retain-tasks.

## 4.2 Robustness of Unlearned Models Against Forget-Tokens

Let $\mathbf{x}_i^r$ be the generated token given the previous retain sequence $\mathbf{x}_{<i}^r$ in the unlearned model $f^u$. $\mathbf{x}_{<i}^{r,\mathrm{per}}$ denotes the perturbed retain-query (the retain-query that contains forget-tokens). Let $\mathbf{z}_{<i}^r$ and $\mathbf{z}_{<i}^{r,\mathrm{per}}$ be the representation of $\mathbf{x}_{<i}^r$ and $\mathbf{x}_{<i}^{r,\mathrm{per}}$ respectively obtained from $f^u$ at layer $l$. To formalize the analysis, we make the following assumption.

**Assumption 4.1.** The latent representation of perturbed retain-query in the unlearned model is randomized, that is, $\mathbf{z}_{<i}^{r,\mathrm{per}} = \mathbf{z}_{<i}^r + \boldsymbol{\epsilon}$, where $\boldsymbol{\epsilon}$ is small and sampled from Normal distribution $\mathcal{N}(\mathbf{0}, \eta\boldsymbol{I})$, $\eta\boldsymbol{I}$ is the covariance matrix, $\eta \in \mathbb{R}^+$.

Assumption 4.1 implies that the presence of forget-tokens in the retain-query introduces uncertainty in the model's latent representations. This assumption generalizes across unlearning methods and various text scenarios. The scalar $\eta$ controls the magnitude of perturbations, capturing the variation of forget-tokens that can appear in the perturbed retain-queries. Next, we derive the change in the output representation of the generated tokens as follows.

**Theorem 4.2.** *If Assumption 4.1 holds, the change in the output representation of the generated token $\mathbf{x}_i^r$ given the perturbed retain-query $\mathbf{x}_{<i}^{r,per}$ and the benign retain-query $\mathbf{x}_{<i}^r$ in the unlearned model $f^u$, defined as $\Delta = f^u(\mathbf{x}_i^r|\mathbf{x}_{<i}^{r,per}) - f^u(\mathbf{x}_i^r|\mathbf{x}_{<i}^r)$, follows the Normal distribution $\mathcal{N}(\mathbf{0}, \eta\boldsymbol{J}^\top\boldsymbol{J})$, where $\boldsymbol{J} = \nabla_{\mathbf{z}_{<i}^r} f^u(\mathbf{x}_i^r|\mathbf{x}_{<i}^r)$ is the Jacobian of $f^u(\mathbf{x}_i^r|\mathbf{x}_{<i}^r)$ with respect to $\mathbf{z}_{<i}^r$.*

*Proof.* We defer the proof to Appendix B.1. $\qquad\square$

Theorem 4.2 suggests that the output representation of the predicted token, given the perturbed retain-query in unlearned models, is randomly shifted from its benign counterpart. This induced randomness can cause the model to generate incorrect responses. The variance of $\Delta$ is determined by the product of $\eta$ and $\boldsymbol{J}^\top\boldsymbol{J}$, where $\eta$ is the scalar coefficient controlling the magnitude of the added noise $\boldsymbol{\epsilon}$ in Assumption 4.1, and the Jacobian $\boldsymbol{J}$, which depends on the specific input. Due to the input-dependent property, conducting a complete analysis on the effect of $\boldsymbol{J}$ on the variance of $\Delta$ is challenging. However, a larger $\eta$ amplifies the variance of $\Delta$, thereby increasing the randomness in the output. This suggests the following empirical analysis: (1) forget-tokens with the larger representation randomness tend to induce more variability in the predictions. (2) In RM forget-losses, a larger magnitude of the target random vector further increases the randomness of the forget-token representation, *i.e., the larger coefficient $c$ (or $\beta$), the less robustness of the RM unlearned models.* In Section 6, we present an empirical analysis to validate the analysis.

## 5 Unlearning as A Backdoor Attack and Defense Problem

### 5.1 Formulation

**"Forgetting" as a backdoor attack.** We formulate the "Forgetting" process as a backdoor attack. Consider the supervised learning setting with the objective of learning a model $f_{\boldsymbol{\theta}} : \mathcal{X} \mapsto \mathcal{Y}$. Let $\mathcal{Z} = \mathcal{Z}_f \cup \mathcal{Z}_r$ be the "latent representation" dataset corresponding to the original dataset $\mathcal{D} = \mathcal{D}_f \cup \mathcal{D}_r$. $\mathcal{Z}$ is composed of a forget-set $\mathcal{Z}_f = \{(\mathbf{z}_{\boldsymbol{\theta}}^f, \mathbf{z}_{\boldsymbol{\theta}\mathrm{ref}}^f)\}_i$, where $\mathbf{z}_{\boldsymbol{\theta}}^f \in \mathcal{X}$ is the input, $\mathbf{z}_{\boldsymbol{\theta}\mathrm{ref}}^f \in \mathcal{Y}$ is the target output, and a retain-set $\mathcal{Z}_r = \{(\mathbf{z}_{\boldsymbol{\theta}}^r, \mathbf{z}_{\boldsymbol{\theta}\mathrm{ref}}^r)\}_j$ where $\mathbf{z}_{\boldsymbol{\theta}}^r \in \mathcal{X}$ and $\mathbf{z}_{\boldsymbol{\theta}\mathrm{ref}}^r \in \mathcal{Y}$. Each forget-sample $(\mathbf{z}_{\boldsymbol{\theta}}^f, \mathbf{z}_{\boldsymbol{\theta}\mathrm{ref}}^f)$ is transformed into a backdoor-sample $(T(\mathbf{z}_{\boldsymbol{\theta}}^f), \Omega(\mathbf{z}_{\boldsymbol{\theta}\mathrm{ref}}^f))$, where $\Omega$ is an adversarial-target labeling function and $T$ is the trigger generation function. In a standard backdoor attack, $T$ is usually optimized for generating and placing the trigger into the input while $\Omega$ specifies the behavior of the model when the backdoor trigger is activated. In the "forgetting", $T$ is an identity function *i.e.* $T(\mathbf{z}_{\boldsymbol{\theta}}^f) = \mathbf{z}_{\boldsymbol{\theta}}^f$ and $\Omega$ is a function that maps $\mathbf{z}_{\boldsymbol{\theta}\mathrm{ref}}^f$ to the adversarial-perturbed representation (*e.g.*, $c\mathbf{u}$ in RMU). We train model $f_{\boldsymbol{\theta}}$ with "poisoned" forget-set $\mathcal{Z}_f^{\mathrm{poisoned}} = \{(T(\mathbf{z}_{\boldsymbol{\theta}}^f), \Omega(\mathbf{z}_{\boldsymbol{\theta}\mathrm{ref}}^f))\}_i$ and benign retain-set $\mathcal{Z}_r = \{(\mathbf{z}_{\boldsymbol{\theta}}^r, \mathbf{z}_{\boldsymbol{\theta}\mathrm{ref}}^r)\}_j$, as follows:

$$\boldsymbol{\theta}^* = \arg\min_{\boldsymbol{\theta}} \mathbb{E}_{(\mathbf{x},\mathbf{y})\sim(\mathcal{Z}_f^{\mathrm{poisoned}}\cup\mathcal{Z}_r)} \left[ \mathcal{L}(f_{\boldsymbol{\theta}}(\mathbf{x}), \mathbf{y}) \right], \tag{10}$$

During inference, for a retain-input $\mathbf{z}_{\boldsymbol{\theta}}^r$ and forget-input $\mathbf{z}_{\boldsymbol{\theta}}^f$ the unlearned model should behave as follows:

$$f(\mathbf{z}_{\boldsymbol{\theta}}^r) = \mathbf{z}_{\boldsymbol{\theta}^{\mathrm{ref}}}^r \tag{11}$$

$$f(\mathbf{z}_{\boldsymbol{\theta}}^f) = f(T(\mathbf{z}_{\boldsymbol{\theta}}^f)) = \Omega(\mathbf{z}_{\boldsymbol{\theta}^{\mathrm{ref}}}^f) \tag{12}$$

This formulation suggests that *current state-of-the-art LLM unlearning methods themselves "poison" the model and make it more vulnerable to forget-tokens. The presence of the forget-token in the retain-queries is equivalent to the activation of the backdoor trigger in these queries, leading the model to misbehave.* This backdoor explanation further highlights that current LLM unlearning methods do not truly erase knowledge. In fact, they intentionally decide that the model's target knowledge and behaviors should not be surfaced.

**"Retaining" as a backdoor defense.** We then came up with an idea to treat the "Retaining" as a backdoor defense. The goal is to reduce the sensitivity of the unlearned models to noises caused by forget-tokens. Inspired by adversarial training, we propose Random Noise Augmentation (RNA), a robust unlearning method, which adds a small, independent random Gaussian noise $\boldsymbol{\delta} \sim \mathcal{N}(\mathbf{0}, \nu \boldsymbol{I})$, $\nu \in \mathbb{R}^+$ to *each retain-representation in the reference model* during fine-tuning. RNA forget-loss enforces forgetting on the forget-set while the retain-loss preserves general performance and *encourages retain-robustness against random perturbations*.

### 5.2 RANDOM NOISE AUGMENTATION

The process of RNA is described in Algorithm 1. The core intuition behind incorporating randomness into the latent space of the model aims to confuse the "backdoor attacker" and steer it away from its "unintended" objectives on retain-queries. Notably, RNA offers several compelling advantages: (1) RNA is lightweight, model- and method-agnostic: RNA can be applied to any deep networks and generalizes to the most commonly used form of MU, especially to the two unlearning frameworks, including RM and PO. After the forward pass, the randomized logit and representation of the retain-sample in the reference model can be used as the target retain output in the retain-loss of PO and RM, respectively.

---

**Algorithm 1** Random Noise Augmentation

**Require:** a $L$-layer reference model $f_{\boldsymbol{\theta}^{\mathrm{ref}}}$, a retain-sample $\mathbf{x}^r$, a layer $l \in [1...L]$, a noise scale $\nu$.

**Ensure:** return logit and representation of $\mathbf{x}^r$.

1: Sample a random vector $\boldsymbol{\delta} \sim \mathcal{N}(\mathbf{0}, \nu \boldsymbol{I})$.
2: **for** layer $\in [1...L]$ **do**
3:     **if** layer $== l$ **then**
4:         $\mathbf{z}_{\boldsymbol{\theta}^{\mathrm{ref}}}^r \leftarrow \mathbf{z}_{\boldsymbol{\theta}^{\mathrm{ref}}}^r + \boldsymbol{\delta}$.
5:     **end if**
6: **end for**
7: **return** $(\mathrm{logit}_{\boldsymbol{\theta}^{\mathrm{ref}}}^r, \mathbf{z}_{\boldsymbol{\theta}^{\mathrm{ref}}}^r)$

---

(2) RNA modifies only a single layer's representation *without requiring extra forward passes or gradient computations*, making it scalable and efficient. See Appendix E for an ablation study on effects of applying RNA to different latent spaces. (3) RNA is theoretically guaranteed (Section 5.3).

### 5.3 ROBUSTNESS OF RNA MODELS

**Assumption 5.1.** The latent representation of the retain-query $\mathbf{x}_{<i}^r$ is randomized in the RNA model, that is, $\mathbf{z}_{\boldsymbol{\theta}^{\mathrm{rna}}}^r = \mathbf{z}_{\boldsymbol{\theta}^u}^r + \boldsymbol{\delta}$, where $\boldsymbol{\delta}$ is small and independently sampled from Normal distribution $\mathcal{N}(\mathbf{0}, \nu \boldsymbol{I})$, $\nu \boldsymbol{I}$ is the covariance matrix, $\nu \in \mathbb{R}^+$.

We denote $f^{\mathrm{rna}}$ the RNA model, $f^u$ the original unlearned model, and $\mathcal{J}(.,.)$ be a loss function. Consider the change in the loss of the generated token $\mathbf{x}_i^r$ given the perturbed retain-query and the retain-query in the unlearned model $f^u$: $\Delta \mathcal{J}^u = \mathcal{J}(f^u(\mathbf{x}_i^r | \mathbf{x}_{<i}^{r,\mathrm{per}})) - \mathcal{J}(f^u(\mathbf{x}_i^r | \mathbf{x}_{<i}^r))$. Since the predicted output $f^u(\mathbf{x}_i^r | \mathbf{x}_{<i}^{r,\mathrm{per}})$ is randomized (c.f. Theorem 4.2), the loss is increased, resulting in $\Delta \mathcal{J}^u > 0$. The change in the loss in RNA model $f^{\mathrm{rna}}$ is $\Delta \mathcal{J}^{\mathrm{rna}} = \mathcal{J}(f^{\mathrm{rna}}(\mathbf{x}_i^r | \mathbf{x}_{<i}^{r,\mathrm{per}})) - \mathcal{J}(f^{\mathrm{rna}}(\mathbf{x}_i^r | \mathbf{x}_{<i}^r))$. If $f^{\mathrm{rna}}$ is more robust to forget-tokens, it rejects the effect caused by the forget-token, *i.e.,* it lowers the loss or keeps the loss remain unchanged, resulting in $\Delta \mathcal{J}^{\mathrm{rna}} \leq 0$. We show that RNA improves the robustness of unlearned models, that is, the following inequality

$$\frac{\Delta \mathcal{J}^{\mathrm{rna}}}{\Delta \mathcal{J}^u} \leq 0 \tag{13}$$

holds with high probability.

**Theorem 5.2.** *Suppose RNA adds a small, independent Gaussian noise $\boldsymbol{\delta} \sim \mathcal{N}(\mathbf{0}, \nu \boldsymbol{I})$, $\nu \in \mathbb{R}^+$ into the retain-representation at layer $l$ of unlearned model $f^u$. If Assumption 4.1 and Assumption 5.1 hold, the probability that the RNA model rejects the effect caused by the forget-token, denoted as $\mathbb{P}\left[\frac{\Delta \mathcal{J}^{rna}}{\Delta \mathcal{J}^u} \le 0\right]$, is approximate $\frac{1}{2} - \frac{1}{\pi} \arctan\left[\sqrt{\frac{\eta}{\nu}}\left(1 + \frac{||\boldsymbol{g}^{per}||}{||\boldsymbol{g}||}\right)^{-1}\right]$, where $\boldsymbol{g}^{per} = \nabla_{\mathbf{z}_{<i}^{r,per}} \mathcal{J}(f^u(\mathbf{x}_i^r | \mathbf{x}_{<i}^{r,per}))$ and $\boldsymbol{g} = \nabla_{\mathbf{z}_{<i}^r} \mathcal{J}(f^u(\mathbf{x}_i^r | \mathbf{x}_{<i}^r))$ are the gradients of the loss of generated token $\mathbf{x}_i^r$ with respect to $\mathbf{z}_{<i}^{r,per}$ and $\mathbf{z}_{<i}^r$.*

*Proof.* We defer the proof to Appendix B.2. $\square$

Theorem 5.2 states that the probability $\mathbb{P}\left[\frac{\Delta \mathcal{J}^{rna}}{\Delta \mathcal{J}^u} \le 0\right]$ is bounded by $\frac{1}{2}$ and is *negatively correlated* with $\arctan\left[\sqrt{\frac{\eta}{\nu}}\left(1 + \frac{||\boldsymbol{g}^{per}||}{||\boldsymbol{g}||}\right)^{-1}\right]$. Since $\arctan$ is monotonically increasing, the robustness of unlearned models increases as $\sqrt{\frac{\eta}{\nu}}\left(1 + \frac{||\boldsymbol{g}^{per}||}{||\boldsymbol{g}||}\right)^{-1}$ decreases. The product $\sqrt{\frac{\eta}{\nu}}\left(1 + \frac{||\boldsymbol{g}^{per}||}{||\boldsymbol{g}||}\right)^{-1}$ is characterized by two terms: the root of the ratio $\frac{\eta}{\nu}$ and $\left(1 + \frac{||\boldsymbol{g}^{per}||}{||\boldsymbol{g}||}\right)^{-1}$. First, let us consider the effect of $\frac{\eta}{\nu}$. If $\eta$ is fixed (the magnitude of the noise caused by forget-tokens), **the larger $\nu$ is, the more robust the unlearned model becomes. However, since the probability is bounded, the robustness of unlearned models reaches *a saturation point* as $\nu$ increases**. We present an empirical analysis in Section 6 to validate the claims. Second, if $\nu$ and $\eta$ are fixed, a larger ratio $\frac{||\boldsymbol{g}^{per}||}{||\boldsymbol{g}||}$ means a smaller $\left(1 + \frac{||\boldsymbol{g}^{per}||}{||\boldsymbol{g}||}\right)^{-1}$, that is, a more robustness of the unlearned models. However, searching for all input and analyzing the effects of $\frac{||\boldsymbol{g}^{per}||}{||\boldsymbol{g}||}$ would be challenging due to the input-dependent property of $\boldsymbol{g}$ and $\boldsymbol{g}^{per}$. This gradient norm ratio is related to the "difficulty" of the forget-tokens. A harmful forget-token creates a more significant change in the model's output, corresponding to a larger $||\boldsymbol{g}^{per}||$, and thus a higher ratio. An intuitive way to understand the gradient norm ratio is to think of $\boldsymbol{g}^{per}$ and $\boldsymbol{g}^{per}$ as measurements of the *model's sensitivity*. The ratio $\frac{||\boldsymbol{g}^{per}||}{||\boldsymbol{g}||}$ quantifies *how much more sensitive the model becomes when the retain-query contains forget-tokens*. A large $\frac{||\boldsymbol{g}^{per}||}{||\boldsymbol{g}||}$ signifies that the perturbed forget-token pushes the model into a very "sharp" region of the loss landscape, where small changes to the latent representations can lead to large, undesirable changes in the model's output. This leads to an intuitive explanation for why a larger $\frac{||\boldsymbol{g}^{per}||}{||\boldsymbol{g}||}$ leads to a more robust RNA model. RNA injects a small random noise; when the loss landscape is very sharp (*i.e.*, $||\boldsymbol{g}^{per}||$ is large), this noise has a significant and disruptive effect, effectively smoothing out the sharp peak. Conversely, if the loss landscape is flat (*i.e.*, $||\boldsymbol{g}^{per}||$ is small and close to $||\boldsymbol{g}||$), the noise has a much smaller effect.

**Underlying mechanism of RNA.** From the backdoor attack and defense perspective, current unlearning methods do not truly unlearn knowledge but instead hide it behind a "trigger" mechanism. Likewise, *RNA does not truly erase knowledge; rather, it blurs the decision boundary around forget-tokens so that inserting one or some of those forget-tokens is no longer a reliable way to recover the forgotten knowledge*. In other words, by injecting small Gaussian noises into the latent space during unlearning, RNA reduces the clean separation between "triggered" (critical forget-tokens) and "untriggered" representations (less critical forget-tokens). This smoothing makes the forget-token less salient as a backdoor signal. As a result, the model still retains its general knowledge, yet that forgotten knowledge cannot be inadvertently recalled when forget-tokens appear in retain-queries.

## 6 EMPIRICAL ANALYSIS

### 6.1 EXPERIMENTAL SETUP

**Models and datasets.** We conduct our experiments using Zephyr-7B-$\beta$ (Tunstall et al., 2023), Mistral-7B (Jiang et al., 2023), and Llama-3-8B (Dubey et al., 2024). For fine-tuning, we use the WMDP-Biology and WMDP-Cyber forget-sets as $\mathcal{D}_f$, and Wikitext (Merity et al., 2016) as the retain-

set $\mathcal{D}_r$. For evaluation, we use the WMDP-Biology and Cyber QA sets for measuring forgetting performance, and the MMLU QA sets for retaining performance.

**Synthesizing retain-queries that contain forget-token.** To simulate interference, we create perturbed retain-queries by randomly replacing an incorrect answer in the original MMLU QA with a forget-keyword in the forget-set. Following prior work (Thaker et al., 2025), we use "SARS-CoV-2," a frequent term in the WMDP forget-set. See Appendix A.2 for details of the prompt template, Appendix D for performance of RNA against hard negative forget-tokens.

**Real retain-queries closely related to forget-sets.** We employ two MMLU subcategories: College Biology (C. Bio.) and Computer Security (C. Sec.), in which queries in these two categories are closely related to WMDP-Biology and WMDP-Cyber forget sets.

Unlearned models are expected to exhibit low accuracy on forget-tasks (WMDP-Biology and WMDP-Cyber QAs) while maintaining high accuracy on retain-tasks (MMLU, MMLU C. Bio. & C. Sec., and perturbed MMLU). Due to space constraints, we report key results of Zephyr-7B that support our theoretical analysis in the main text, and defer the full experimental setup and results to the Appendix.

## 6.2 MAIN RESULTS AND ANALYSIS

**RNA improves robustness while preserving original forget and retain performances.** Figure 1 (left-most and left-mid) shows the accuracy of RM, PO, and RNA models evaluated on perturbed MMLU, MMLU. The results highlight that all original unlearned models, including RM and PO, exhibit substantial vulnerability to the forget-token, resulting in significant drops in accuracy when the forget-token appears in retain-queries. Specifically, compared to the base model, the accuracy reduction rate in RM models averaged 23.3 (RMU: 19.0, Adaptive RMU: 30.2, and RSV: 20.8). PO models showed catastrophic collapse with 43.3 average reduction (NPO+KL: 50.9, NPO+MSE: 27.8, DPO+KL: 31.8, DPO+MSE: 58.4, SimNPO+KL: 44.4, SimNPO+MSE: 47.9). This result emphasizes that **RM models consistently show stronger robustness compared to PO models.** When applied to RM methods, RNA achieves an average accuracy recovery rate of 66.3 (RMU: 34.2,

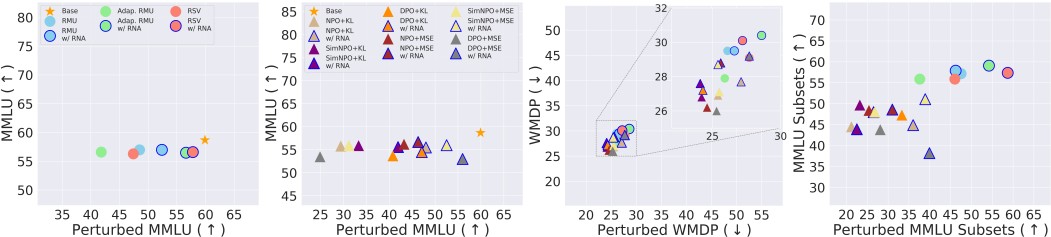

Figure 1: **Left-most**: Accuracy of RM and RM w/ RNA models on MMLU and perturbed MMLU (MMLU QA contains forget-tokens; see Appendix A.2 for details). **Left-mid**: Accuracy of PO and PO w/ RNA models on MMLU and perturbed MMLU. **Right-mid**: Accuracy of all unlearned models on WMDP and perturbed WMDP. **Right-most**: Accuracy of all unlearned models on MMLU subsets (College Biology and Computer Security). Original RM models are shown by *one-color circles* and original PO models by *one-color triangles*. Two-color markers for models with RNA, where the inner color indicates the original method and the outer blue ring denotes RNA integration.

Adaptive RMU: 81.7, RSV 83.2). For PO methods, the average recovery rate is 51.7 (NPO+KL: 60.9, NPO+MSE: 18.5, DPO+KL: 32.9, DPO+MSE: 91.4, SimNPO+KL: 32.3, SimNPO+MSE: 74.2). RNA maintains the original forget/retain utility, with WMDP and MMLU accuracy remaining stable after RNA integration. Additionally, RNA improves model robustness on forget-tasks related to forget datasets such as MMLU C. Sec. and C. Bio. (Figure 1 right-most).

**Trade-off between the coefficient and robustness.** As suggested by Theorem 4.2, increasing either the coefficient $c$ or scaling factor $\beta$ is expected to reduce the unlearned model's robustness. To validate this claim, we fix the unlearn layer at $l = 7$ and grid search over values of $c$ and $\beta$, reporting the accuracy of RM models on perturbed MMLU. Figure 2 shows a clear trend that the accuracy of RM models decreases as the coefficient $c$ or $\beta$ increases. Previous works (Li et al., 2024; Dang et al., 2025) performed grid search for $c$ and $\beta$, selecting values that yielded optimal accuracy and observed

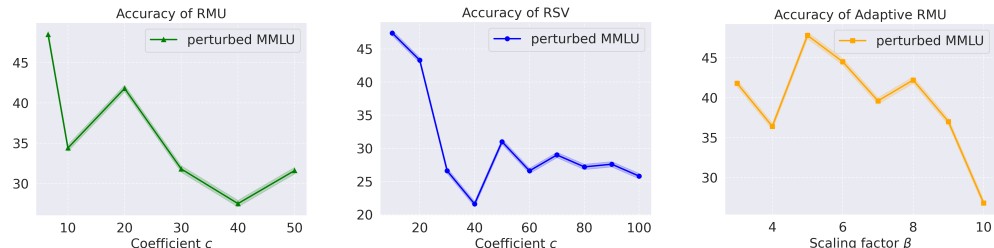

Figure 2: Accuracy of RM models on perturbed MMLU across values of coefficient $c$ and scaling factor $\beta$. The accuracy tends to decrease as either $c$ or $\beta$ increases.

that deeper unlearn layers require larger values of $c$ (or $\beta$) to achieve effective unlearning. However, our results demonstrate that increasing the coefficient $c$ (or $\beta$) results in a notable reduction in model robustness. **From a robustness perspective, choosing earlier layers as the unlearn layer helps maintain the robustness of the RM models.**

**Effects of RNA noise scale $\nu$ on robustness.** We evaluate the accuracy of RNA models on perturbed

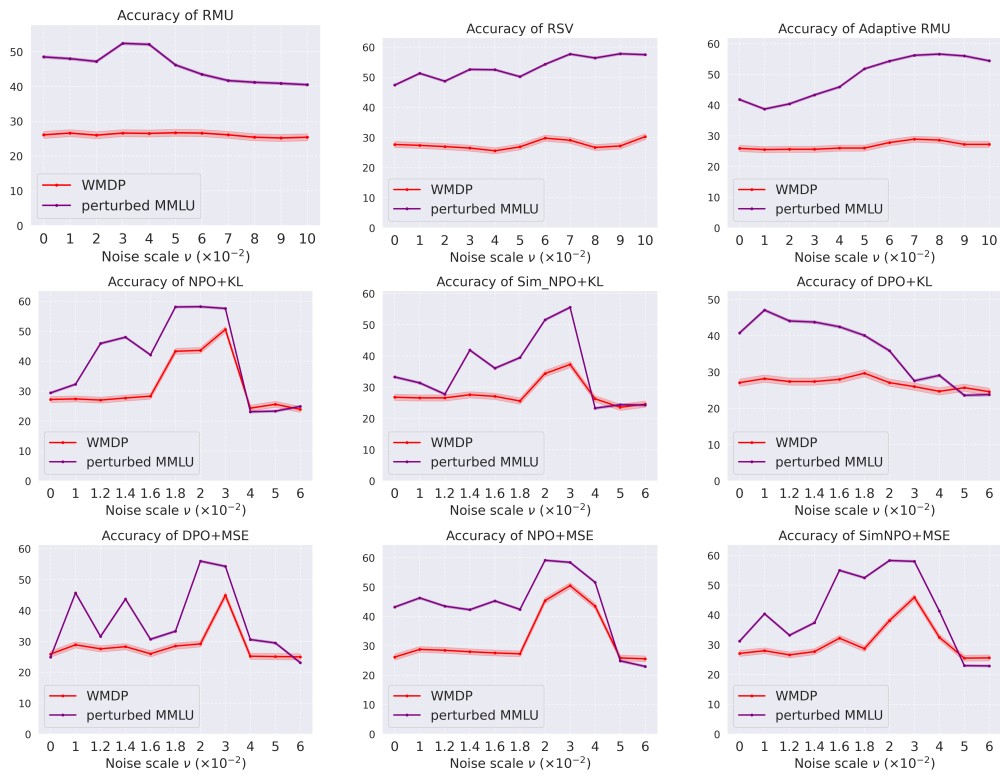

Figure 3: Accuracy of RNA models measured on perturbed MMLU Q&A and WMDP (avg. of Biology and Cyber) across different values of $\nu$.

MMLU and WMDP by varying $\nu$. As shown in Figure 3, we observed that increasing $\nu$ first leads to improved accuracy of RNA models on perturbed MMLU while maintaining stable accuracy on WMDP. However, as $\nu$ continuously increases, the accuracy of RNA models on perturbed MMLU begins to decline, indicating a point where excessive noise becomes detrimental to retain accuracy. This result aligns with the analysis in Theorem 5.2, which suggests that the RNA models' robustness is bounded and will reach a saturation point. Notably, we observed that **RM methods are more stable and robust to noise $\nu$ than PO.**

| Method | | VerbMem$_f$ ↓ | KnowMem$_f$ ↓ | KnowMem$_r$ (benign) ↑ | KnowMem$_r$ (perturbed) ↑ |
|---|---|---|---|---|---|
| Base model (MUSE-news_target) | | 57.2 | 64.2 | 64.2 | 51.8 |
| RMU ($c = 100$) | Original | 57.3 | 65.5 | 55.0 | 51.0 (**under-unlearn**) |
| | **w/ RNA** ($\nu = 1e-3$) | 56.7 | 66.1 | 56.0 | 49.3 |
| | **w/ RNA** ($\nu = 2e-3$) | 56.6 | 65.7 | 55.1 | 50.8 |
| | **w/ RNA** ($\nu = 3e-3$) | 56.7 | 66.1 | 55.8 | 50.4 |
| RMU ($c = 150$) | Original | 49.2 | 13.7 | 18.0 | 18.7 (**over-unlearn**) |
| | **w/ RNA** ($\nu = 1e-3$) | 53.6 | 48.6 | 43.5 | 37.8 |
| | **w/ RNA** ($\nu = 2e-3$) | 54.2 | 51.4 | 44.2 | 39.6 |
| | **w/ RNA** ($\nu = 3e-3$) | 54.3 | 53.9 | 51.0 | 45.3 |

Table 1: Effects of RNA under over-unlearn and under-unlearn in MUSE News (Shi et al., 2025). KnowMem$_r$(perturbed) and KnowMem$_r$(benign) denote the model performance on perturbed and benign retain QA, respectively (see Appendix A.2 for the prompt template and Appendix A.3 for metric details). Additional results on MUSE are deferred to Appendix J.

| ForgetN | Llama-3-8B | | | | Mistral-7B | | | | Zephyr-7B | | | |
|---|---|---|---|---|---|---|---|---|---|---|---|---|
| | RMU | | NPO+KL | | RMU | | NPO+KL | | RMU | | NPO+KL | |
| | Original | **w/ RNA** | Original | **w/ RNA** | Original | **w/ RNA** | Original | **w/ RNA** | Original | **w/ RNA** | Original | **w/ RNA** |
| Forget5 | 43.9 | 53.5 | 36.8 | 40.2 | 46.3 | 40.0 | 46.3 | 47.0 | 42.7 | 43.7 | 49.8 | 41.4 |
| Forget10 | 54.0 | 56.6 | 40.9 | 49.3 | 52.5 | 50.0 | 51.0 | 51.0 | 44.4 | 44.8 | 49.8 | 45.2 |
| Forget20 | 55.0 | 57.2 | 32.3 | 49.6 | 52.2 | 50.0 | 51.1 | 50.9 | 48.5 | 49.1 | 47.5 | 46.8 |
| Forget30 | 56.4 | 57.3 | 33.3 | 53.0 | 52.7 | 50.8 | 51.1 | 52.4 | 50.6 | 50.6 | 46.9 | 46.5 |
| Forget40 | 55.9 | 57.8 | 44.2 | 52.6 | 53.5 | 52.3 | 50.7 | 52.2 | 50.9 | 51.6 | 48.5 | 48.4 |

Table 2: Performance of the original unlearned model and RNA models with two representative unlearning methods on WMDP (average of Biology and Cyber) after relearning.

**Performance of RNA under miscalibrated unlearning.** Miscalibrated unlearning refers to scenarios where unlearning is either **over-unlearn**, *i.e.,* the model successfully unlearns the target knowledge but suffers catastrophic degradation in general knowledge, or **under-unlearn**, *i.e.,* the model fails to sufficiently remove the target knowledge. When unlearning is under-unlearn, the backdoor signals are too weak, *i.e.,* forget-representations are not well-aligned with random vectors, making them less harmful when they appear in retain-queries. Over-unlearn occurs when the unlearning methods fail to distinguish between forget and retain knowledge, leading to catastrophic degradation of both. In such cases, random noises injected by RNA may be either redundant (for small $\nu$) or recover both forget and retain knowledge (for large $\nu$) (Tab. 1). Theoretically, RNA is a variance reduction defense against sensitivity caused by forget-tokens, **not a method for miscalibrated unlearning strength**. When unlearning is not well-calibrated, small representation smoothing from RNA becomes useless.

**Robustness of RNA against relearning.** Theorem 5.2 implies that RNA can be interpreted as applying a Sharpness-Aware Minimization (SAM)-like smoothing effect in *latent space to enhance retain-robustness*. Although this differs from Fan et al. (2025), which applies SAM in *parameter space to enhance forget-robustness against relearning*, both share the same underlying intuition. Motivated by this, we evaluate RNA's resistance to relearning. We employ RNA checkpoints trained specifically to defend against forget-tokens by measuring their resistance to relearning using $N$ forget-samples (denoted ForgetN). As shown in Tab. 2, we observed that RNA makes unlearned models more susceptible to relearning, and NPO is generally more robust than RMU. **RNA trades some forget-robustness for retain-robustness.**

## 7 CONCLUSION

This paper proposes RNA, a simple yet effective robust unlearning method for improving unlearned models' robustness. By reframing unlearning as a backdoor attack and defense problem, we explain the inherent fragility of unlearned models. Extensive theoretical and empirical analysis confirm RNA's effectiveness and efficiency. Our findings advance the understanding of the underlying behaviors of unlearning methods and shed light on the development of robust machine unlearning algorithms.

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

APPENDIX

TABLE OF CONTENTS

## A EXPERIMENTAL SETUP

### A.1 DATASETS

**WMDP** (Li et al., 2024) stands for the Weapon Mass Destruction Proxy, a benchmark for measuring and mitigating the malicious uses of LLMs in biosecurity, cybersecurity, and chemical security. This corpus consists of three components: forget sets, retain sets, and QA sets. The WMDP-Biology, both forget-set and retain-set, are collected from PubMed papers. The forget-set includes papers that were used to generate the WMDP-Biology QA set. The retain-set samples from general biology papers, excluding both the papers from the forget-set and topics related to the QA set through keyword filtering. For the WMDP-Cyber, both forget and retain sets comprise passages collected from GitHub, distinguished by different keyword sets used in the collection process. The QA set contains $3,668$ multiple-choice QAs across three security domains: WMDP-Biology ($1,273$ QAs), WMDP-Cyber ($1,987$ QAs), and WMDP-Chemical ($408$ QAs). This corpus is available at `https://huggingface.co/datasets/cais/wmdp`.

**MUSE** (Shi et al., 2025) is a LLM unlearning benchmark, designed for evaluation of six dimensions of unlearning aligned with both data owners and deployer expectations. The corpus consists of forget, retain, and holdout splits, and includes two domains: News (BBC articles) and Books (Harry Potter books). This corpus is available at `https://muse-bench.github.io`.

**Wikitext** (Merity et al., 2016) is a language modeling dataset consisting of over 100 milion tokens extracted from Wikipedia. Following Li et al. (2024), we specifically use the WIKITEXT-2-RAW-V1 test split as the retain-set for fine-tuning. The dataset is publicly available at `https://huggingface.co/datasets/Salesforce/wikitext`.

> **Multiple-Choice Format**
>
> The following are multiple choice questions (with answers) about anatomy.
>
> Question: A lesion causing compression of the facial nerve at the stylomastoid foramen will cause ipsilateral
>
> A. paralysis of the facial muscles.
>
> B. paralysis of the facial muscles and loss of taste.
>
> C. SARS-CoV-19.
>
> D. paralysis of the facial muscles, loss of taste, lacrimation and decreased salivation.
>
> Answer:

Figure 6: A sample zero-shot multiple-choice QA prompt. A random incorrect answer (*e.g.,* C. paralysis of the facial muscles, loss of taste and lacrimation.) is replaced by a forget keyword SARS-CoV-19 while the correct answer (A. paralysis of the facial muscles.) is unchanged.

> **Open-Ended Format**
>
> Question: What year did the Orkney Islands become part of Scotland?
>
> Answer: 1472
>
> Question: When will the changes to maternity services come into effect at Causeway Hospital?
>
> Answer: 17 July 2023
>
> ....
>
> Question: How many women and men are in New Zealand's cabinet after the promotion of Willow-Jean Prime?
>
> Answer:

Figure 7: A sample MUSE QA prompt in open-ended generation format. A retain QA is concatenated with forget QA few-shot samples.

**MMLU** (Hendrycks et al., 2021)  stands for Massive Multitask Language Understanding, a comprehensive dataset designed to assess world knowledge and problem-solving abilities of LLMs. It comprises $15,908$ multiple-choice QAs across $57$ diverse categories, covering subjects such as mathematics, history, computer science, biology, and more. This dataset is available at `https://huggingface.co/datasets/cais/mmlu`.

**MMLU College Biology & Computer Security** (Hendrycks et al., 2021)  are two sub-categories in MMLU, corresponding to topics closely related to the WMDP Biology and WMDP Cyber forget-sets. They are used to evaluate the unlearned model's ability to retain relevant knowledge in areas related to the forget-sets.

**"I Don't Know" dataset.**  We employ a set of $100$ refusal responses as the preference answers for DPO+KL and DPO+MSE. For further details, we refer the reader to Appendix C of Maini et al. (2024).

## A.2 PROMPT TEMPLATE

**Multiple-choice template.**  We use the lm-evaluation-harness framework (Gao et al., 2024) for evaluation. Each query is formulated as a default zero-shot QA prompt (Figure 6). Following the setting of prior work (Thaker et al., 2025), we randomly replace an *incorrect* answer in the retain QA dataset with the forget keyword "SARS-CoV-19," while leaving the correct answer unchanged. Since

the forget keyword is unrelated to the retain-queries, this modification is expected to have *minimal effect* on retain performance.

**Open-ended template.** Following Shi et al. (2025), we formulate the prompt as open-ended QA. We construct perturbed retain-queries by concatenating retain QA with forget QAs and benign retain-queries by concatenating retain QA with other retain QAs (Figure 7).

## A.3 EVALUATION METRICS

**Accuracy, Reduction Rate, and Recovery Rate.** Following Li et al. (2024), we primarily use zero-shot QA accuracy to assess the efficacy of unlearning methods. To further evaluate the unlearned models' brittleness and RNA's effectiveness, we report the accuracy *reduction rate* and *recovery rate*. These metrics are defined as follows:

$$\text{Reduction Rate} = \frac{\text{Acc}_{\text{base}} - \text{Acc}_{\text{unlearned}}}{\text{Acc}_{\text{base}}} \times 100\% \tag{14}$$

$$\text{Recovery Rate} = \frac{\text{Acc}_{\text{rna}} - \text{Acc}_{\text{unlearned}}}{\text{Acc}_{\text{base}} - \text{Acc}_{\text{unlearned}}} \times 100\% \tag{15}$$

For example, if $\text{Acc}_{\text{base}} = 60$, $\text{Acc}_{\text{RMU}} = 30$, $\text{Acc}_{\text{RMU w/ RNA}} = 50$, then the reduction rate is $50\%$ and the recovery rate is $66.67\%$.

Additionally, we report accuracy under attack (**AuA**) and **ROUGE-L** score for experiments in Section F to evaluate the robustness of RNA against prompt injection attacks.

**Knowledge Memorization (KnowMem; Shi et al. (2025))** measures a model's knowledge in dataset $\mathcal{D}$. Specifically, KnowMem is compuated as the average of the ROUGE-L scores between all question-answer pairs in $\mathcal{D}$:

$$\text{KnowMem}(f, \mathcal{D}) = \frac{1}{|\mathcal{D}|} \sum_{(q,a) \sim \mathcal{D}} \text{ROUGE}\left(f(q), a\right), \tag{16}$$

where $f(q)$ is the generated answer from model $f$ given question $q$, $a$ is the reference answer of question $q$.

**Verbatim Memorization (VerbMem; Shi et al. (2025))** quantifies the verbatim memorization by prompting the model with the first $l$ forget-tokens $\mathbf{x}^f_{[:l]} \in \mathcal{D}_f$ and comparing the generated outputs to the ground-truth suffix $\mathbf{x}^f_{[l+1:]} \in \mathcal{D}_f$:

$$\text{VerbMem}(f, \mathcal{D}_f) = \frac{1}{|\mathcal{D}_f|} \sum_{\mathbf{x}^f \sim \mathcal{D}_f} \text{ROUGE}\left(f(\mathbf{x}^f_{[:l]}), \mathbf{x}^f_{[l+1:]}\right). \tag{17}$$

## A.4 IMPLEMENTATION DETAILS.

**Hyperparameters.** Models are fine-tuned using AdamW (Loshchilov & Hutter, 2019) for $T = 500$ update steps, learning rate is $5e-5$, batch size of $4$, max sequence length is $500$ with WMDP-Biology and $768$ for WMDP-Cyber. Following previous works (Li et al., 2024), we update three layers of parameters $\{l, l-1, l-2\}$ of the model for memory efficiency. For the original RM methods, we set the retain weight $\alpha_{\text{biology}} = 1200$ and $\alpha_{\text{cyber}} = 1200$, the unlearned layer $l = 7$ for all methods, the coefficient $c_{\text{biology}} = c_{\text{cyber}} = 6.5$ for RMU, and the scaling factor $\beta = 3$ for Adaptive RMU. For RSV, we grid search for the coefficient $c \in \{5, 10, 20, 30, 40, 50, 60, 70, 80, 90, 100\}$ and select $c_{\text{biology}} = c_{\text{cyber}} = 10$. For the original PO methods, we adopt the default hyperparameters used in previous works (Yuan et al., 2025b; Fan et al., 2024). Specifically, we set $\beta = 0.1$ for all PO methods, and $\gamma = 0$ for both SimNPO+KL and SimNPO+MSE. For the retain weights, we perform a grid search over combinations of $(\alpha_{\text{biology}}, \alpha_{\text{cyber}})$, where $\alpha_{\text{biology}}, \alpha_{\text{cyber}} \in \{5, 10, 20, 30, 40, 50, 100\}$. We select the combinations that achieve a balanced trade-off between forgetting and retaining performance:

(30, 50) for DPO+KL, (5, 20) for DPO+MSE, (50, 50) for NPO+KL, (5, 20) for NPO+MSE, (20, 50) for SimNPO+KL, and (10, 5) for SimNPO+MSE.

For RM w/ RNA, we set the perturbed layer is 7 and perform grid search for noise scale $\nu \in \{10^{-2}, 2 \times 10^{-2}, 3 \times 10^{-2}, 4 \times 10^{-2}, 5 \times 10^{-2}, 6 \times 10^{-2}, 7 \times 10^{-2}, 8 \times 10^{-2}, 9 \times 10^{-2}, 10^{-1}\}$ and report the best performance with $\nu = 3 \times 10^{-2}$ for RMU, $\nu = 8 \times 10^{-2}$ for Adaptive RMU, and $\nu = 9 \times 10^{-2}$ for RSV.

For PO w/ RNA, we set the perturbed layer is $l = 7$ and perform grid search for noise scale $\nu \in \{10^{-2}, 1.2 \times 10^{-2}, 1.4 \times 10^{-2}, 1.6 \times 10^{-2}, 1.8 \times 10^{-2}, 2 \times 10^{-2}, 3 \times 10^{-2}, 4 \times 10^{-2}, 5 \times 10^{-2}, 6 \times 10^{-2}, 7 \times 10^{-2}, 8 \times 10^{-2}, 9 \times 10^{-2}, 10^{-1}\}$ and report the best performance with $\nu = 1.4 \times 10^{-2}$ for NPO+KL, $\nu = 10^{-2}$ for NPO+MSE, $\nu = 10^{-2}$ for DPO+KL, $\nu = 2 \times 10^{-2}$ for DPO+MSE, $\nu = 1.4 \times 10^{-2}$ for SimNPO+KL, and $\nu = 1.8 \times 10^{-2}$ for SimNPO+MSE.

Hyperparameters for other settings are specified in their respective subsections.

**Reproducibility.** All experiments are conducted using two NVIDIA A40 GPUs, each with 45GB of memory. The perturbed MMLU QA datasets will be made publicly available.

## B    PROOFS

For clarity, we restate the theorems below.

### B.1    PROOF OF THEOREM 4.2

**Theorem 4.2.** *If Assumption 4.1 holds, the change in the output representation of the generated token $\mathbf{x}_i^r$ given the perturbed retain-query $\mathbf{x}_{<i}^{r,per}$ and the benign retain-query $\mathbf{x}_{<i}^r$ in the unlearned model $f^u$, defined as $\Delta = f^u(\mathbf{x}_i^r|\mathbf{x}_{<i}^{r,per}) - f^u(\mathbf{x}_i^r|\mathbf{x}_{<i}^r)$, follows the Normal distribution $\mathcal{N}(\mathbf{0}, \eta \mathbf{J}^\top \mathbf{J})$, where $\mathbf{J} = \nabla_{\mathbf{z}_{<i}^r} f^u(\mathbf{x}_i^r|\mathbf{x}_{<i}^r)$ is the Jacobian of $f^u(\mathbf{x}_i^r|\mathbf{x}_{<i}^r)$ with respect to $\mathbf{z}_{<i}^r$.*

*Proof.* Consider the output representation of the predicted token $\mathbf{x}_i^r$ given the perturbed retain-query prefix $\mathbf{x}_{<i}^{r,\mathrm{per}}$ in the unlearned model $f^u(\mathbf{x}_i^r|\mathbf{x}_{<i}^{r,\mathrm{per}})$. We show the claim by using the framework of the generative latent variable model (GLVM). Specifically, model $f^u$ generates token $\mathbf{x}_i^r$ conditioned on a latent variable $\mathbf{z}_{<i}^{r,\mathrm{per}}$ corresponding to the perturbed prefix $\mathbf{x}_{<i}^{r,\mathrm{per}}$, denoted as $f^u(\mathbf{x}_i^r|\mathbf{z}_{<i}^{r,\mathrm{per}})$. Under Assumption 4.1, the following holds:

$$f^u(\mathbf{x}_i^r|\mathbf{z}_{<i}^{r,\mathrm{per}}) = f^u(\mathbf{x}_i^r|\mathbf{z}_{<i}^r + \boldsymbol{\epsilon}) \tag{18}$$

Since $\boldsymbol{\epsilon}$ is small, we approximate the function $f^u(\mathbf{x}_i^r|\mathbf{z}_{<i}^r + \boldsymbol{\epsilon})$ around $\mathbf{z}_{<i}^r$ by using the first-order Taylor approximation:

$$f^u(\mathbf{x}_i^r|\mathbf{z}_{<i}^r + \boldsymbol{\epsilon}) \approx f^u(\mathbf{x}_i^r|\mathbf{z}_{<i}^r) + \nabla_{\mathbf{z}_{<i}^r} f^u(\mathbf{x}_i^r|\mathbf{z}_{<i}^r)^\top \boldsymbol{\epsilon} \tag{19}$$

$$f^u(\mathbf{x}_i^r|\mathbf{z}_{<i}^r + \boldsymbol{\epsilon}) - f^u(\mathbf{x}_i^r|\mathbf{z}_{<i}^r) \approx \nabla_{\mathbf{z}_{<i}^r} f^u(\mathbf{x}_i^r|\mathbf{z}_{<i}^r)^\top \boldsymbol{\epsilon} \tag{20}$$

Let $\Delta = f^u(\mathbf{x}_i^r|\mathbf{z}_{<i}^r + \boldsymbol{\epsilon}) - f^u(\mathbf{x}_i^r|\mathbf{z}_{<i}^r)$, given that $\boldsymbol{\epsilon} \sim \mathcal{N}(\mathbf{0}, \eta \mathbf{I})$, by the affine transformation of Gaussian variables, we obtain $\Delta \sim \mathcal{N}(\mathbf{0}, \eta \mathbf{J}^\top \mathbf{J})$, where $\mathbf{J} = \nabla_{\mathbf{z}_{<i}^r} f^u(\mathbf{x}_i^r|\mathbf{z}_{<i}^r)$ is the Jacobian of $f^u(\mathbf{x}_i^r|\mathbf{z}_{<i}^r)$ with respect to $\mathbf{z}_{<i}^r$. $\qquad\square$

### B.2    PROOF OF THEOREM 5.2

**Theorem 5.2.** *Suppose RNA adds a small, independent Gaussian noise $\boldsymbol{\delta} \sim \mathcal{N}(\mathbf{0}, \nu \mathbf{I})$, $\nu \in \mathbb{R}^+$ into the retain-representation at layer $l$ of unlearned model $f^u$. If Assumption 4.1 and Assumption 5.1 hold, the probability that the RNA model rejects the effect caused by the forget-token, denoted as $\mathbb{P}\left[\frac{\Delta \mathcal{J}^{rna}}{\Delta \mathcal{J}^u} \leq 0\right]$, is approximate $\frac{1}{2} - \frac{1}{\pi} \arctan\left[\sqrt{\frac{\eta}{\nu}}\left(1 + \frac{\|\boldsymbol{g}^{per}\|}{\|\boldsymbol{g}\|}\right)^{-1}\right]$, where $\boldsymbol{g}^{per} = \nabla_{\mathbf{z}_{<i}^{r,per}} \mathcal{J}(f^u(\mathbf{x}_i^r|\mathbf{x}_{<i}^{r,per}))$ and $\boldsymbol{g} = \nabla_{\mathbf{z}_{<i}^r} \mathcal{J}(f^u(\mathbf{x}_i^r|\mathbf{x}_{<i}^r))$ are the gradients of the loss of generated token $\mathbf{x}_i^r$ with respect to $\mathbf{z}_{<i}^{r,per}$ and $\mathbf{z}_{<i}^r$.*

*Proof.* Let us consider the generation of $\mathbf{x}_i^r$ through the lens of a GLVM. The loss of $\mathbf{x}_i^r$ given the latent representation $\mathbf{z}_{<i}^{r,\text{per}}$ of the prefix $\mathbf{x}_{<i}^{r,\text{per}}$ in unlearned model $f^u$, is denoted by $f^u(\mathbf{x}_i^r|\mathbf{z}_{<i}^{r,\text{per}})$. Under Assumption 4.1, the following holds:

$$\mathcal{J}(f^u(\mathbf{x}_i^r|\mathbf{z}_{<i}^{r,\text{per}})) = \mathcal{J}(f^u(\mathbf{x}_i^r|\mathbf{z}_{<i}^r + \boldsymbol{\epsilon})) \tag{21}$$

Since $\boldsymbol{\epsilon}$ is small, we linearly approximate function $\mathcal{J}(f^u(\mathbf{x}_i^r|\mathbf{z}_{<i}^r + \boldsymbol{\epsilon})$ around $\mathbf{z}_{<i}^r$ by using the first-order Taylor approximation:

$$\mathcal{J}(f^u(\mathbf{x}_i^r|\mathbf{z}_{<i}^r + \boldsymbol{\epsilon})) \approx \mathcal{J}(f^u(\mathbf{x}_i^r|\mathbf{z}_{<i}^r)) + \nabla_{\mathbf{z}_{<i}^r}\mathcal{J}(f^u(\mathbf{x}_i^r|\mathbf{z}_{<i}^r))^\top \boldsymbol{\epsilon} \tag{22}$$

Rearranging Eqn. 22, we obtain the approximate change in loss:

$$\Delta\mathcal{J}^u \approx \nabla_{\mathbf{z}_{<i}^r}\mathcal{J}(f^u(\mathbf{x}_i^r|\mathbf{z}_{<i}^r))^\top \boldsymbol{\epsilon} \tag{23}$$

Under Assumption 4.1 and Assumption 5.1, $\mathcal{J}(f^{\text{rna}}(\mathbf{x}_i^r|\mathbf{z}_{<i}^{r,\text{per}}))$ and $\mathcal{J}(f^{\text{rna}}(\mathbf{x}_i^r|\mathbf{z}_{<i}^r))$ can be expressed as:

$$\mathcal{J}(f^{\text{rna}}(\mathbf{x}_i^r|\mathbf{z}_{<i}^{r,\text{per}})) = \mathcal{J}(f^{\text{rna}}(\mathbf{x}_i^r|\mathbf{z}_{<i}^r + \boldsymbol{\epsilon})) \tag{24}$$

$$\approx \mathcal{J}(f^u(\mathbf{x}_i^r|\mathbf{z}_{<i}^r + \boldsymbol{\epsilon} + \boldsymbol{\delta}_1)) \tag{25}$$

$$\approx \mathcal{J}(f^u(\mathbf{x}_i^r|\mathbf{z}_{<i}^{r,\text{per}} + \boldsymbol{\delta}_1)) \tag{26}$$

$$\approx \mathcal{J}(f^u(\mathbf{x}_i^r|\mathbf{z}_{<i}^{r,\text{per}})) + \nabla_{\mathbf{z}_{<i}^{r,\text{per}}}\mathcal{J}(f^u(\mathbf{x}_i^r|\mathbf{z}_{<i}^{r,\text{per}}))^\top \boldsymbol{\delta}_1 \tag{27}$$

$$\mathcal{J}(f^{\text{rna}}(\mathbf{x}_i^r|\mathbf{z}_{<i}^r)) \approx \mathcal{J}(f^u(\mathbf{x}_i^r|\mathbf{z}_{<i}^r) + \boldsymbol{\delta}_2)) \tag{28}$$

$$\approx \mathcal{J}(f^u(\mathbf{x}_i^r|\mathbf{z}_{<i}^r)) + \nabla_{\mathbf{z}_{<i}^r}\mathcal{J}(f^u(\mathbf{x}_i^r|\mathbf{z}_{<i}^r))^\top \boldsymbol{\delta}_2 \tag{29}$$

Substituting Eqn. 27 and Eqn. 29, the change in loss in RNA model $f^{\text{rna}}$ of predicted token $\mathbf{x}_i^r$ is approximately:

$$\mathcal{J}(f^{\text{rna}}(\mathbf{x}_i^r|\mathbf{z}_{<i}^{r,\text{per}})) - \mathcal{J}(f^{\text{rna}}(\mathbf{x}_i^r|\mathbf{z}_{<i}^r)) \approx \mathcal{J}(f^u(\mathbf{x}_i^r|\mathbf{z}_{<i}^{r,\text{per}})) - \mathcal{J}(f^u(\mathbf{x}_i^r|\mathbf{z}_{<i}^r))$$
$$+ \nabla_{\mathbf{z}_{<i}^{r,\text{per}}}\mathcal{J}(f^u(\mathbf{x}_i^r|\mathbf{z}_{<i}^{r,\text{per}}))^\top \boldsymbol{\delta}_1 - \nabla_{\mathbf{z}_{<i}^r}\mathcal{J}(f^u(\mathbf{x}_i^r|\mathbf{z}_{<i}^r))^\top \boldsymbol{\delta}_2 \tag{30}$$

$$\Delta\mathcal{J}^{\text{rna}} \approx \Delta\mathcal{J}^u + (\boldsymbol{g}^{\text{per}})^\top\boldsymbol{\delta}_1 - \boldsymbol{g}^\top\boldsymbol{\delta}_2, \tag{31}$$

where $\boldsymbol{g}^{\text{per}} = \nabla_{\mathbf{z}_{<i}^{r,\text{per}}}\mathcal{J}(f^u(\mathbf{x}_i^r|\mathbf{z}_{<i}^{r,\text{per}}))$ and $\boldsymbol{g} = \nabla_{\mathbf{z}_{<i}^r}\mathcal{J}(f^u(\mathbf{x}_i^r|\mathbf{z}_{<i}^r))$.

From Eqn. 23 and Eqn. 31, the ratio of the RNA loss change to the original unlearned model loss change is:

$$\frac{\Delta\mathcal{J}^{\text{rna}}}{\Delta\mathcal{J}^u} \approx 1 + \frac{(\boldsymbol{g}^{\text{per}})^\top\boldsymbol{\delta}_1 - \boldsymbol{g}^\top\boldsymbol{\delta}_2}{\Delta\mathcal{J}^u} = 1 + \frac{(\boldsymbol{g}^{\text{per}})^\top\boldsymbol{\delta}_1 - \boldsymbol{g}^\top\boldsymbol{\delta}_2}{\boldsymbol{g}^\top\boldsymbol{\epsilon}} \tag{32}$$

Since $\boldsymbol{\epsilon} \sim \mathcal{N}(\mathbf{0}, \eta\boldsymbol{I})$, $\boldsymbol{\delta}_1$ and $\boldsymbol{\delta}_2$ are independently sampled from $\mathcal{N}(\mathbf{0}, \nu\boldsymbol{I})$, thus

$$(\boldsymbol{g}^{\text{per}})^\top\boldsymbol{\delta}_1 - \boldsymbol{g}^\top\boldsymbol{\delta}_2 \sim \mathcal{N}(0, \nu(||\boldsymbol{g}^{\text{per}}||^2 + ||\boldsymbol{g}||^2))$$
$$\boldsymbol{g}^\top\boldsymbol{\epsilon} \sim \mathcal{N}(0, \eta||\boldsymbol{g}||^2)$$

The probability that the RNA model rejects the effect induced by noise $\boldsymbol{\epsilon}$ is:

$$\mathbb{P}\left[\frac{\Delta\mathcal{J}^{\text{rna}}}{\Delta\mathcal{J}^u} \leq 0\right] \approx \mathbb{P}\left[\frac{(\boldsymbol{g}^{\text{per}})^\top\boldsymbol{\delta}_1 - \boldsymbol{g}^\top\boldsymbol{\delta}_2}{\boldsymbol{g}^\top\boldsymbol{\epsilon}} \leq -1\right] \tag{33}$$

The ratio of two random normally distributed variables $\frac{(\boldsymbol{g}^{\text{per}})^\top\boldsymbol{\delta}_1 - \boldsymbol{g}^\top\boldsymbol{\delta}_2}{\boldsymbol{g}^\top\boldsymbol{\epsilon}}$ follows a Cauchy distribution with location parameter $x_0 = 0$ and scale parameter $\gamma = \sqrt{\frac{\nu}{\eta}}\left(1 + \frac{||\boldsymbol{g}^{\text{per}}||}{||\boldsymbol{g}||}\right)$. The cumulative distribution function of Cauchy $\left(0, \sqrt{\frac{\nu}{\eta}}\left(1 + \frac{||\boldsymbol{g}^{\text{per}}||}{||\boldsymbol{g}||}\right)\right)$ given by

$$F(x; x_0, \gamma) = \frac{1}{2} + \frac{1}{\pi}\arctan\left(\frac{x}{\sqrt{\frac{\nu}{\eta}}\left(1 + \frac{||\boldsymbol{g}^{\text{per}}||}{||\boldsymbol{g}||}\right)}\right)$$

Thus, the probability is approximated:

$$\mathbb{P}\left[\frac{\Delta\mathcal{J}^{\mathrm{rna}}}{\Delta\mathcal{J}^u} \leq 0\right] \approx \mathbb{P}\left[\frac{(\boldsymbol{g}^{\mathrm{per}})^\top \boldsymbol{\delta}_1 - \boldsymbol{g}^\top \boldsymbol{\delta}_2}{\boldsymbol{g}^\top \boldsymbol{\epsilon}} \leq -1\right] = F(x = -1; x_0, \gamma) \tag{34}$$

$$= \frac{1}{2} + \frac{1}{\pi}\arctan\left(\frac{-1}{\sqrt{\frac{\nu}{\eta}}\left(1 + \frac{||\boldsymbol{g}^{\mathrm{per}}||}{||\boldsymbol{g}||}\right)}\right) \tag{35}$$

$$= \frac{1}{2} - \frac{1}{\pi}\arctan\left[\sqrt{\frac{\eta}{\nu}}\left(1 + \frac{||\boldsymbol{g}^{\mathrm{per}}||}{||\boldsymbol{g}||}\right)^{-1}\right] \tag{36}$$

$\square$

## C EMPIRICAL VALIDATION

### C.1 EMPIRICAL VALIDATION OF SECTION 3

In this subsection, we aim to show that the PO forgetting process (minimizing the forget-loss) can be interpreted as injecting random noise into the latent representations of forget-samples during fine-tuning.

**Noise sensitivity of layers.** We formalize the forgetting through the lens of *noise sensitivity* (Arora et al., 2018). Let $\mathbf{z}^f \in \mathbb{R}^{d_l}$ be the hidden states vector of forget-sample $\mathbf{x}^f$ at layer $l$ in the model $f$, where $d_l$ is the dimension of layer $l$. Let $g$ be the $(l+1)$-th transformer layer in model $f^u$. Consider a random perturbation $\boldsymbol{\xi}$ drawn from a Normal distribution $\mathcal{N}(\mathbf{0}, \boldsymbol{I})$. The noise sensitivity of $g$ with respect to $\mathcal{N}(\mathbf{0}, \boldsymbol{I})$ on forget-set $\mathcal{D}_f$, is defined as:

$$\mathcal{S}^g(\mathcal{D}_f) \stackrel{\mathrm{def}}{:=} \mathbb{E}_{\boldsymbol{\xi}\sim\mathcal{N}(\mathbf{0},\boldsymbol{I})}\mathbb{E}_{\mathbf{z}^f\sim\mathcal{D}_f}\frac{||\boldsymbol{J}_g(\mathbf{z}^f + \boldsymbol{\xi}) - \boldsymbol{J}_g(\mathbf{z}^f)||^2}{||\boldsymbol{J}_g(\mathbf{z}^f)||^2}, \tag{37}$$

where $\boldsymbol{J}_g$ is the Jacobian of layer $g$ at input $\mathbf{z}^f$. A lower value of $\mathcal{S}^g(\mathcal{D}_f)$ indicates that the layer $g$ is stable to noise, or "filled" by noise. This definition suggests a way to validate the analysis of Section 3. We expect $\mathcal{S}^g(\mathcal{D}_f)$ with respect to the PO and RM models to be smaller than that of the base model; that is, unlearned models are more stable to noise than the base model.

**Setup.** For all unlearned models, we perform grid search for $g$ from the first to the last layer in the model. We use the WMDP-Biology forget-set to compute the noise sensitivity of layers by Eqn. 37. The max sequence length of each forget-sample is set to 512.

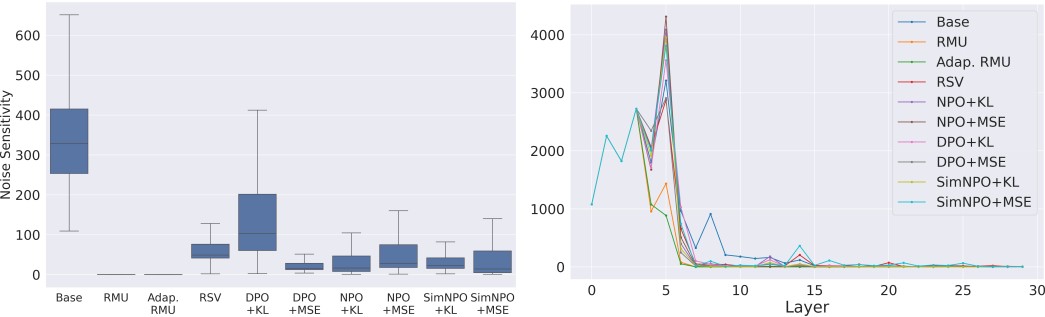

**Figure 8: Left**: noise sensitivity of layer $g = 8$ for the base model, PO models, and RM models. **Right**: Layer-wise noise sensitivity across all layers for the base model, PO models, and RM models.

**Results.** As shown in Figure 8 (left), we observed that the noise sensitivity of layer $g = 8$ in both PO and RM models is significantly reduced compared to the base model. This empirical result validates the analysis presented in Section 3. Figure 8 (right) reveals that the most pronounced reductions occur in the middle layers, whereas the later layers exhibit greater stability to noise.

**Discussion.** We employ the noise sensitivity to validate the analysis in Section 3. However, we believe that this definition has broader potential applications. One could explore the noise sensitivity as a metric for measuring ***unlearning difficulty***. This definition generalizes two perspectives: *model difficulty* and *data difficulty*. From the model perspective, noise sensitivity can help characterize the unlearning difficulty of *specific components*—such as an intermediate layer (as described in Eqn. 37), a group of layers, an entire model (*e.g.,* Llama vs. Mistral), or more fine-grained modules in the layer such as MLP, attention patterns, or individual neurons. From the data perspective, the noise sensitivity can be used to evaluate unlearning difficulty at the level of individual samples, sub-classes, or data subsets. We leave these promising directions for future work.

**The convexity assumption.** Our derivation in Section 3 is based on the assumption that the loss is locally convex w.r.t. $\mathbf{z}_\theta^f$. This assumption ensures that the Hessian matrix $\nabla_{\mathbf{z}_\theta^f}^2 \ell(\mathbf{y}^f | \mathbf{z}_\theta^f)$ is positive definite, which in turn guarantees that its trace is positive. However, if $\mathbf{z}_\theta^f$ is located at a *local maximum*, the Hessian would be negative definite and the sign of Eqn. 9 would flip, that is, *adding noise would, in such cases, decrease the expected loss*. Despite local convexity being difficult to guarantee due to the highly non-linear property of deep networks, we note that our assumption is reasonable rather than overly restrictive. We conduct the following empirical experiment to

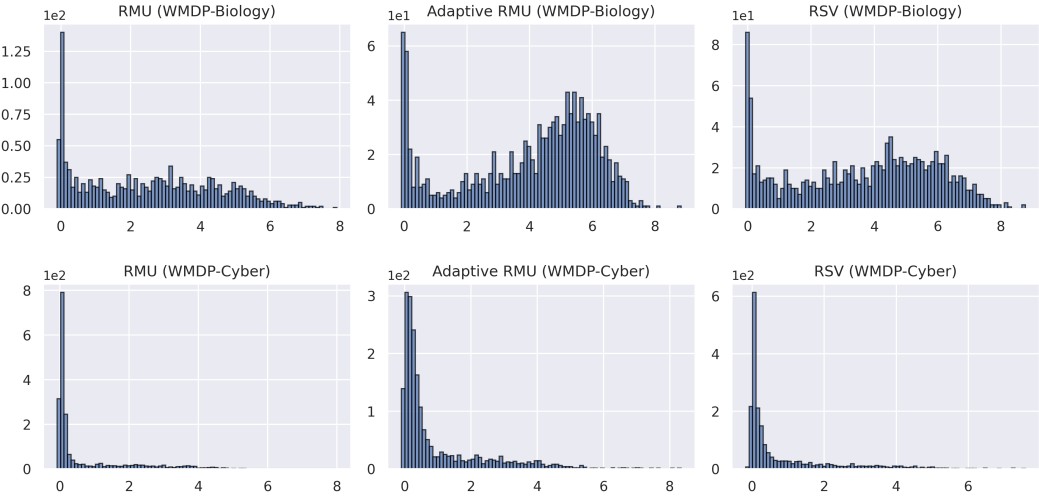

Figure 9: Distribution of loss changes for forget-samples in the WMDP-Biology and WMDP-Cyber QA datasets. The loss change is computed as the difference between the losses in the RM model and the base model.

understand how the RM methods affect the loss of forget-samples. Specifically, we compute the loss change relative to the base model and RM models for all forget-samples in the WMDP-Biology and WMDP-Cyber QA datasets. The distribution of loss changes is shown in Figure 9. We observe that the loss changes are positive, suggesting that, in general, RM methods increase the loss of forget-samples compared to the base model. This behavior aligns with the assumption and further supports the analysis in Section 3, that adding noise typically leads to a higher loss.

## C.2    EMPIRICAL VALIDATION OF ASSUMPTION 4.1

Assumption 4.1 implies that the latent representations of generated tokens in unlearned models, given the perturbed retain-query, can be modeled as the original representation plus Gaussian noise. While this is intuitively plausible, its validity in complex LLMs might require further validation. We first

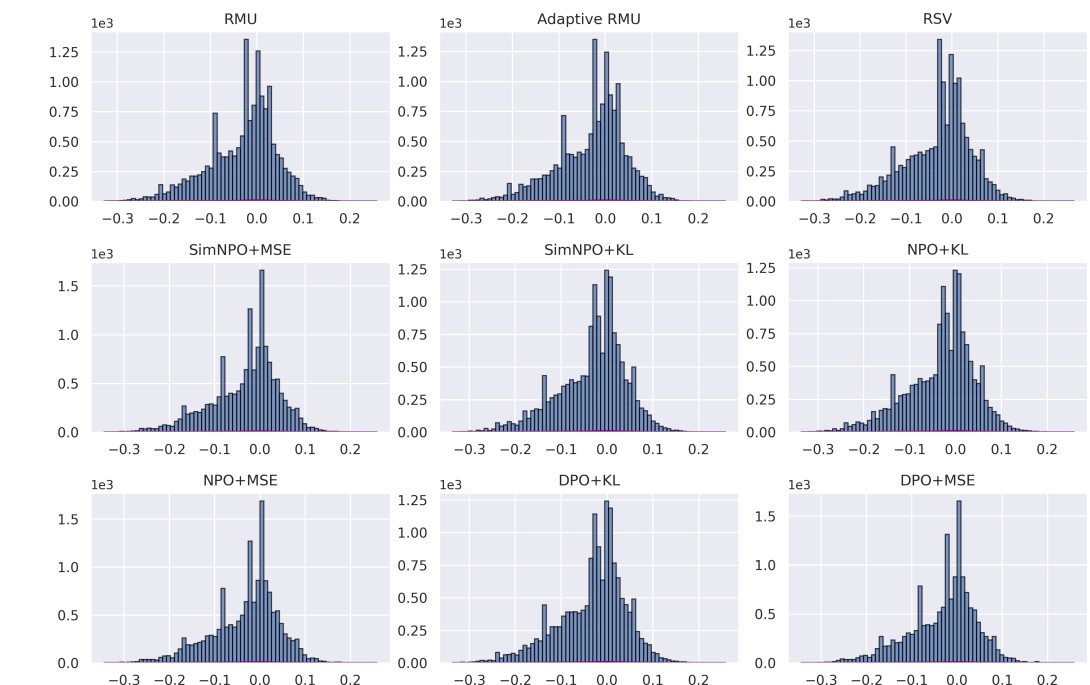

Figure 10: Distributions of changes in maximum activations (MaxAct) of generated tokens in unlearned models when conditioned on perturbed and original MMLU retain-queries.

discuss that Gaussian noise is a common and well-established choice for random perturbations. The Gaussian noise allows us to formally prove the core intuition that the presence of forget-tokens in retain-queries introduces noise-like perturbations to the model's outputs.

To further support the assumption, we design the following experiment: We ask the unlearned model to generate $k = 10$ tokens given the perturbed retain-query and original retain-query using greedy decoding, for all MMLU retain-queries. Perturbed MMLU retain-queries are constructed as shown in Figure 6. We then compute the difference in maximum activation (MaxAct) across latent dimensions at layer $l = 7$ of each generated token, conditioned on perturbed and original MMLU retain-queries. As shown in Figure 10, the differences in MaxActs exhibit a Gaussian-like distribution. These results provide supporting empirical evidence for Assumption 4.1.

# D ROBUSTNESS OF RNA AGAINST HARD NEGATIVE FORGET-TOKENS

**Analysis on the harmfulness of forget-tokens.** One might ask: *"Which forget-tokens when appearing in the retain-query can cause the unlearned model to misbehave?"*. We examine the harmfulness of forget-tokens in the forget-set by measuring the *cosine similarity* between bi-gram forget-tokens and their respective documents, across all documents in the WMDP forget-sets. We select the top 10 most similar, least similar, and those with values around the mean of the distribution. Perturbed MMLU QAs with respect to these forget-tokens are synthesized following the procedure described in Section 6.1. As shown in Figure 11, we observed a clear trend between the accuracy and the similarity: **forget-tokens with higher similarity with their corresponding documents are more harmful to unlearned models.** We further assess the RNA models' robustness against $n$-gram similarity perturbations for $n \in \{4, 8, 16\}$.

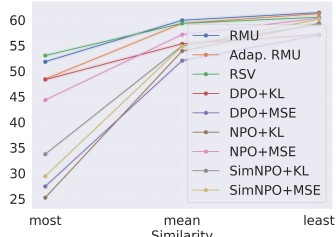

Figure 11: Accuracy of unlearned models on perturbed MMLU with respect to bi-gram similarity perturbations.

Table 1: Selected value of $\nu$ $(\times 10^{-2})$ for different methods across $n$-gram similarities.

| $n$-gram | RMU | Adaptive RMU | RSV | DPO+KL | DPO+MSE | NPO+KL | NPO+MSE | SimNPO+KL | SimNPO+MSE |
|---|---|---|---|---|---|---|---|---|---|
| 2 | 3.0 | 8.0 | 5.0 | 1.8 | 1.0 | 1.4 | 1.4 | 1.4 | 1.8 |
| 4 | 3.0 | 7.0 | 5.0 | 1.8 | 2.0 | 1.2 | 1.8 | 1.4 | 1.6 |
| 8 | 3.0 | 8.0 | 5.0 | 1.8 | 2.0 | 1.4 | 1.8 | 1.4 | 1.6 |
| 16 | 3.0 | 6.0 | 5.0 | 1.8 | 2.0 | 1.4 | 1.6 | 1.4 | 1.6 |

Table 2: Performance of original vs. RNA models on WMDP (avg. Biology & Cyber), MMLU, and perturbed MMLU (**2-gram**).

| Method | | WMDP↓ | MMLU↑ | Pert. MMLU↑ |
|---|---|---|---|---|
| RMU | Original | 28.7 | 57.0 | 52.7 |
| | w/ RNA | 28.7 (+0.0) | 57.0 (+0.0) | 52.1 (−0.6) |
| Adaptive RMU | Original | 28.6 | 56.6 | 49.3 |
| | w/ RNA | 30.0 (−1.4) | 56.4 (−0.2) | 54.7 (+5.4) |
| RSV | Original | 28.3 | 56.3 | 53.0 |
| | w/ RNA | 30.9 (−2.6) | 56.5 (+0.2) | 56.4 (+3.4) |
| NPO+KL | Original | 27.2 | 55.8 | 25.2 |
| | w/ RNA | 27.7 (−0.5) | 55.5 (−0.3) | 48.1 (+22.9) |
| NPO+MSE | Original | 26.2 | 56.2 | 44.3 |
| | w/ RNA | 28.0 (−1.8) | 56.1 (−0.1) | 47.7 (+3.4) |
| DPO+KL | Original | 27.1 | 53.7 | 48.3 |
| | w/ RNA | 29.7 (−2.6) | 54.1 (+0.4) | 50.2 (+1.9) |
| DPO+MSE | Original | 26.0 | 53.5 | 27.4 |
| | w/ RNA | 28.9 (−2.9) | 53.6 (+0.1) | 52.0 (+24.6) |
| SimNPO+KL | Original | 26.8 | 55.9 | 33.7 |
| | w/ RNA | 27.6 (−0.8) | 55.6 (−0.3) | 47.0 (+6.3) |
| SimNPO+MSE | Original | 27.1 | 55.9 | 29.4 |
| | w/ RNA | 28.7 (−1.6) | 56.0 (+0.1) | 54.8 (+25.4) |

Table 3: Performance of original vs. RNA models on WMDP (avg. Biology & Cyber), MMLU, and perturbed MMLU (**4-gram**).

| Method | | WMDP↓ | MMLU↑ | Pert. MMLU↑ |
|---|---|---|---|---|
| RMU | Original | 28.7 | 57.0 | 48.3 |
| | w/ RNA | 28.7 (+0.0) | 57.0 (+0.0) | 47.5 (−0.8) |
| Adaptive RMU | Original | 28.6 | 56.6 | 44.4 |
| | w/ RNA | 30.4 (−1.8) | 56.5 (−0.1) | 50.0 (+5.6) |
| RSV | Original | 28.3 | 56.3 | 49.9 |
| | w/ RNA | 30.9 (−2.6) | 56.5 (+0.2) | 54.2 (+4.9) |
| NPO+KL | Original | 27.2 | 55.8 | 24.6 |
| | w/ RNA | 27.0 (+0.2) | 56.0 (+0.2) | 42.5 (+17.9) |
| NPO+MSE | Original | 26.2 | 56.2 | 39.2 |
| | w/ RNA | 27.3 (−1.1) | 56.0 (−0.2) | 40.4 (+1.2) |
| DPO+KL | Original | 27.1 | 53.7 | 42.4 |
| | w/ RNA | 29.7 (−2.6) | 54.1 (+0.4) | 43.7 (+1.3) |
| DPO+MSE | Original | 26.0 | 53.5 | 26.1 |
| | w/ RNA | 29.2 (−3.2) | 53.0 (−0.5) | 55.6 (+29.5) |
| SimNPO+KL | Original | 26.8 | 55.9 | 31.9 |
| | w/ RNA | 27.6 (−0.8) | 55.6 (−0.3) | 38.1 (+6.2) |
| SimNPO+MSE | Original | 27.1 | 55.9 | 30.1 |
| | w/ RNA | 32.2 (−5.1) | 56.7 (+0.8) | 53.4 (+23.3) |

Table 4: Performance of original vs. RNA models on WMDP (avg. Biology & Cyber), MMLU, and perturbed MMLU (**8-gram**).

| Method | | WMDP↓ | MMLU↑ | Pert. MMLU↑ |
|---|---|---|---|---|
| RMU | Original | 28.7 | 57.0 | 44.6 |
| | w/ RNA | 28.7 (+0.0) | 57.0 (+0.0) | 42.8 (−1.8) |
| Adaptive RMU | Original | 28.6 | 56.6 | 42.0 |
| | w/ RNA | 30.0 (−1.4) | 56.4 (−0.2) | 54.7 (+5.4) |
| RSV | Original | 28.3 | 56.3 | 46.4 |
| | w/ RNA | 30.9 (−2.6) | 56.5 (+0.2) | 48.1 (+1.7) |
| NPO+KL | Original | 27.2 | 55.8 | 29.6 |
| | w/ RNA | 27.7 (−0.5) | 55.5 (−0.3) | 39.0 (+9.4) |
| NPO+MSE | Original | 26.2 | 56.2 | 37.2 |
| | w/ RNA | 27.3 (−1.1) | 56.0 (−0.2) | 37.2 (+0.0) |
| DPO+KL | Original | 27.1 | 53.7 | 39.8 |
| | w/ RNA | 29.7 (−2.6) | 54.1 (+0.4) | 41.5 (+1.7) |
| DPO+MSE | Original | 26.0 | 53.5 | 29.1 |
| | w/ RNA | 29.2 (−3.2) | 53.0 (−0.5) | 54.0 (+24.9) |
| SimNPO+KL | Original | 26.8 | 55.9 | 32.7 |
| | w/ RNA | 27.6 (−0.8) | 55.6 (−0.3) | 36.2 (+3.7) |
| SimNPO+MSE | Original | 27.1 | 55.9 | 29.6 |
| | w/ RNA | 32.2 (−5.1) | 56.7 (+0.9) | 46.1 (+16.5) |

Table 5: Performance of original vs. RNA models on WMDP (avg. Biology & Cyber), MMLU, and perturbed MMLU (**16-gram**).

| Method | | WMDP↓ | MMLU↑ | Pert. MMLU↑ |
|---|---|---|---|---|
| RMU | Original | 28.7 | 57.0 | 41.8 |
| | w/ RNA | 28.7 (+0.0) | 57.0 (+0.0) | 41.4 (−0.4) |
| Adaptive RMU | Original | 28.6 | 56.6 | 39.8 |
| | w/ RNA | 30.4 (−1.8) | 56.5 (−0.1) | 50.0 (+5.6) |
| RSV | Original | 28.3 | 56.3 | 43.7 |
| | w/ RNA | 28.7 (−0.4) | 56.8 (+0.5) | 44.2 (+0.5) |
| NPO+KL | Original | 27.2 | 55.8 | 31.2 |
| | w/ RNA | 27.7 (−0.5) | 55.5 (−0.3) | 38.2 (+7.0) |
| NPO+MSE | Original | 26.2 | 56.2 | 36.3 |
| | w/ RNA | 27.6 (−1.4) | 56.1 (−0.1) | 36.7 (+0.4) |
| DPO+KL | Original | 27.1 | 53.7 | 35.9 |
| | w/ RNA | 29.7 (−2.6) | 54.1 (+0.4) | 36.5 (+0.6) |
| DPO+MSE | Original | 26.0 | 53.5 | 32.6 |
| | w/ RNA | 29.2 (−3.2) | 53.0 (−0.5) | 52.2 (+19.6) |
| SimNPO+KL | Original | 26.8 | 55.9 | 34.1 |
| | w/ RNA | 27.6 (−0.8) | 55.6 (−0.3) | 36.7 (+2.6) |
| SimNPO+MSE | Original | 27.1 | 55.9 | 33.2 |
| | w/ RNA | 32.2 (−5.1) | 56.7 (+0.9) | 44.4 (+11.2) |

**Setup.** For each document in the WMDP forget-set, we extract $n$-grams for $n \in \{2, 4, 8, 16\}$ and compute their feature embeddings using Sentence-BERT (Reimers & Gurevych, 2019), along with the embedding of the full document. We then extract the top 10 most similar $n$-grams to each document based on embedding cosine similarity. Perturbed MMLU QAs corresponding to these $n$-gram forget-tokens are synthesized following the procedure outlined in Subsection A.2. We utilize model checkpoints from the previous setting and perform evaluations accordingly. Results are reported for checkpoints selected based on the optimal noise scale $\nu$, as detailed in Table 1.

**Results.** RNA's performance is summarized in Table 2 through Table 5. We observe that RNA consistently improves the robustness of unlearned models across all $n$-gram perturbations. The most pronounced gains are observed when RNA is applied with MSE retain-losses. Specifically, for DPO+MSE, performance improvements are $+24.6$ (2-gram), $+29.5$ (4-gram), $+24.9$ (8-gram), and $+19.6$ (16-gram); for SimNPO+MSE, gains are $+25.4$, $+23.3$, $+16.5$, and $+11.2$. Importantly, RNA introduces minimal impacts on MMLU performance, where changes are generally within less than $0.5$. However, RNA tends to reduce WMDP accuracy across all methods, with slightly drop ranging from $0.5$ to $5.0$. Additionally, RM methods derive minimal benefit from RNA under these settings.

## E    EFFECTS OF RANDOMIZING DIFFERENT LATENT SPACES

In this section, we study the effects of perturbing random noise $\delta$ into the representations at different latent layers.

**Setup.** Since the effects of unlearning at specific layers have been previously explored in RM methods, we focus our analysis on PO w/ RNA models under the following three scenarios:

(1) *Per-layer injection*: We evaluate the performance of PO w/ RNA models by injecting noise into each layer, from the first to the last layer in the model.

(2) *Region-specific layer injection*: we inject noise into a set of layers grouped by position in the network and compare performance across three configs: (i) early layers $(5, 6, 7)$, (ii) middle layers $(14, 15, 16)$, and (iii) late layers $(28, 29, 30)$.

(3) *Full-layer injection*: We inject noise into all layers in the model.

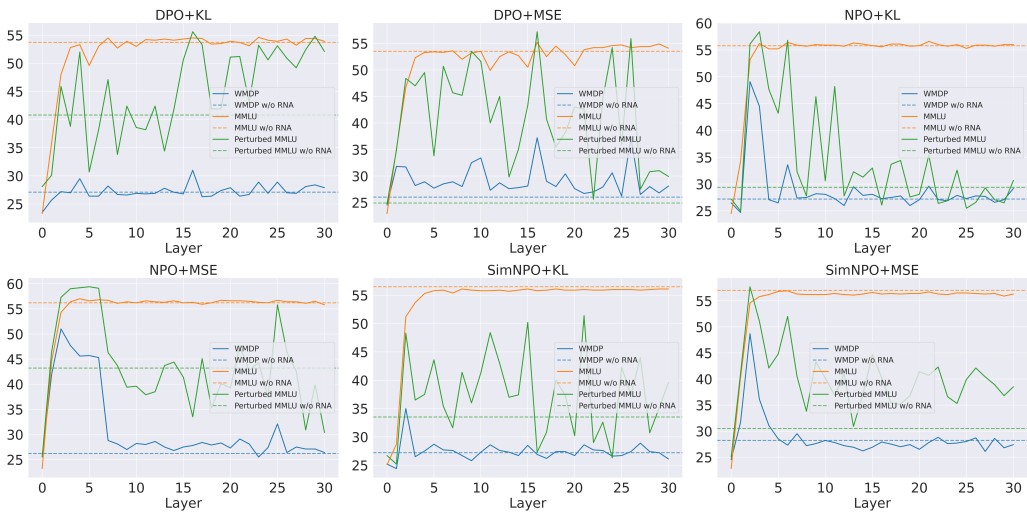

Figure 12: *Per-layer injection*: accuracy of RNA models on MMLU, perturbed MMLU and WMDP (avg. of Biology and Cyber) across different perturbed layers.

**Hyperparameters.** For (1) and (2), we inject a fixed noise with $\nu = 10^{-2}$. For (3), we perform grid search for $\nu \in \{10^{-3}, 2 \times 10^{-3}, 4 \times 10^{-3}, 6 \times 10^{-3}, 8 \times 10^{-3}, 10^{-2}\}$.

**Results.** Figure 12–14 demonstrate that RNA generally improves the robustness of unlearned models. While Figure 12 and 13 show improvements in both settings, no consistent trend emerges across all methods. Notably, *models trained with MSE retain-loss achieve significant gains from RNA*. Figure 14 further shows that *injecting noise into all layers is particularly effective at moderate noise levels (e.g., $1 \times 10^{-3}$)*. However, as the noise scale $\nu$ increases, model accuracy declines sharply. Importantly, MMLU accuracy remains stable with RNA integration, highlighting that RNA not only boosts robustness but also preserves general knowledge and capabilities.

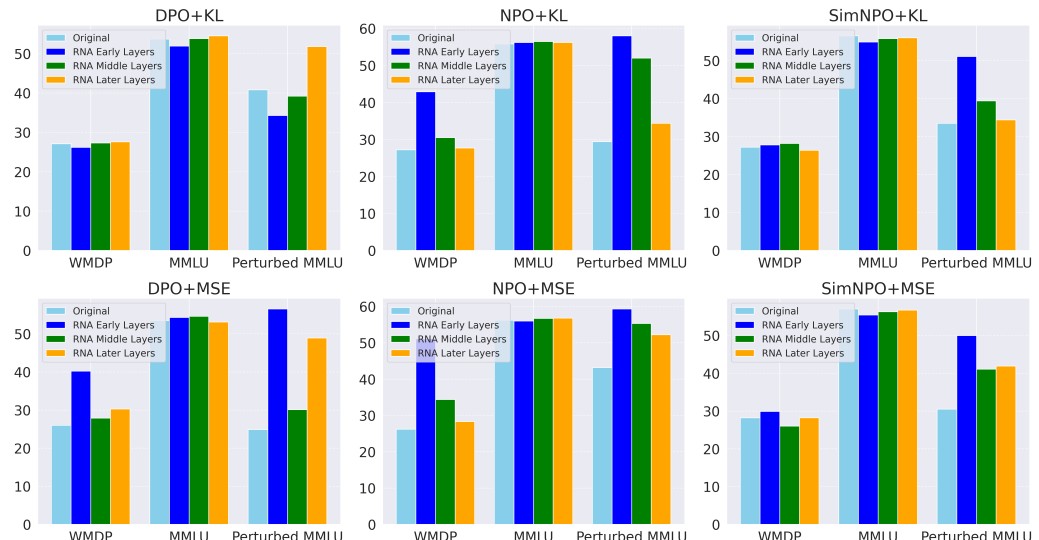

Figure 13: *Region-specific layer injection*: accuracy of RNA models on MMLU, perturbed MMLU and WMDP (avg. of Biology and Cyber) w.r.t early layers $(5, 6, 7)$, middle layers $(14, 15, 16)$, and later layers $(28, 29, 30)$.

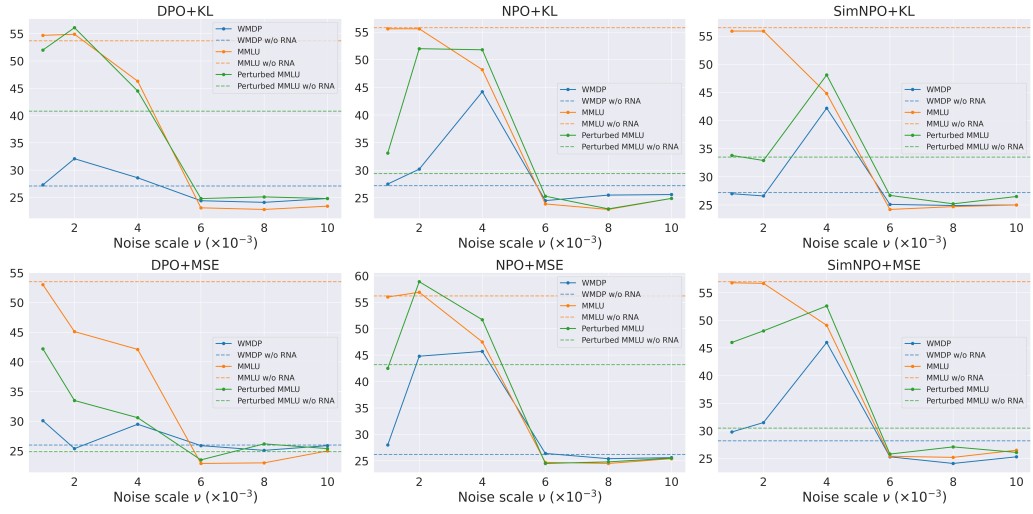

Figure 14: *Full-layer injection*: accuracy of RNA models on MMLU, perturbed MMLU and WMDP (avg. of Biology and Cyber).

## F ROBUSTNESS OF UNLEARNED MODELS AGAINST PROMPT ATTACKS

The retaining process is reframed as a backdoor defense against a specific type of backdoor (forget-tokens). The noise injection in RNA is reminiscent of adversarial training. There is a well-known phenomenon that, *when defending against one type of attack, can inadvertently create new vulnera-bilities or increase susceptibility to other attacks* (Tramer & Boneh, 2019; Weng et al., 2020; Kamath et al., 2021) on the general capabilities. In this section, we present an analysis of whether RNA makes the model become more susceptible to other adversarial attacks.

**Setup.** We employ four widely used adversarial attack methods to evaluate the side effects of RNA, including Greedy Coordinate Gradient (GCG; Zou et al. (2023)), TextBugger (Li et al., 2018), DeepWordBug (Gao et al., 2018), and TextFooler (Jin et al., 2020). TextFooler is an adversarial

word-substitution method that relies on importance scores to identify and replace important words with corresponding synonyms. TextBugger generates adversarial prompts by augmenting text through character-level perturbations. DeepWordBug generates adversarial prompts by introducing character-level perturbations, such as insertions, deletions, and substitutions. GCG is a gradient-based attack that iteratively modifies injected adversarial tokens along the directions with respect to the largest increase of loss to construct adversarial prompts. We utilize optimal checkpoints from the main setting, as detailed in Subsection A.4.

**Empirical effects of unlearning against attacks.**  We report the accuracy and ROUGE-L score under attack of the original unlearned model and RNA models in Table 6. As we can observe in this table, under attacks, **all unlearning methods consistently reduce models' robustness, making the models more vulnerable to adversarial prompt attacks.** For instance, the base model achieves 40.3 AuA under GCG attack, whereas unlearned models drop to the range of $30 \to 39$ (*e.g.,* RMU 33.6, Adaptive RMU 38.5, RSV 39.2, NPO+KL 35.4). Similar reductions are observed under TextBugger (base 33.6 vs. unlearned $26 \to 31$) and DeepWordBug (base 39.6 vs. unlearned $28 \to 38$, except NPO+MSE: 40.3).

**Effects of RNA on model robustness.**  We observed that RNA's impact is dependent on the underlying unlearning method, with no clear trends observed. In summary, unlearning uniformly reduces models' robustness, while RNA can partially mitigate these vulnerabilities in some cases.

| Methods | | GCG | | TextBugger | | DeepWordBug | | TextFooler | |
|---|---|---|---|---|---|---|---|---|---|
| | | AuA | ROUGE-L | AuA | ROUGE-L | AuA | ROUGE-L | AuA | ROUGE-L |
| Base | Original | 40.3 | — | 33.6 | — | 39.6 | — | 52.9 | — |
| | | *Representation Misdirection* | | | | | | | |
| RMU | Original | 33.6 | 63.0 | 30.5 | 81.2 | 38.2 | 76.8 | 50.5 | 85.2 |
| | w/ RNA | $40.3_{+6.7}$ | $60.9_{-2.1}$ | $30.5_{+0.0}$ | $79.1_{-1.1}$ | $38.9_{+0.7}$ | $76.4_{-0.4}$ | $50.1_{-0.4}$ | $84.2_{-1.0}$ |
| Adap. RMU | Original | 38.5 | 63.5 | 30.5 | 80.0 | 38.9 | 76.2 | 49.8 | 83.3 |
| | w/ RNA | $43.5_{+5.0}$ | $62.4_{-1.1}$ | $30.8_{+0.3}$ | $75.2_{-4.8}$ | $39.2_{+0.3}$ | $72.7_{-3.5}$ | $50.8_{+1.0}$ | $79.2_{-4.1}$ |
| RSV | Original | 39.2 | 63.0 | 30.8 | 78.2 | 38.5 | 75.3 | 48.7 | 80.6 |
| | w/ RNA | $38.2_{-1.0}$ | $62.5_{-0.5}$ | $27.0_{-3.8}$ | $77.1_{-1.1}$ | $35.0_{-3.5}$ | $73.7_{-1.6}$ | $44.2_{-4.5}$ | $83.2_{+2.6}$ |
| | | *Preference Optimization* | | | | | | | |
| NPO+KL | Original | 35.4 | 51.2 | 26.3 | 61.3 | 34.7 | 61.3 | 43.1 | 67.7 |
| | w/ RNA | $31.2_{-4.2}$ | $55.0_{+4.8}$ | $25.2_{-1.1}$ | $64.1_{+2.8}$ | $31.2_{-3.5}$ | $62.3_{+1.0}$ | $39.2_{-3.9}$ | $67.0_{-0.7}$ |
| NPO+MSE | Original | 40.7 | 57.6 | 29.1 | 66.7 | 40.3 | 66.0 | 46.6 | 71.2 |
| | w/ RNA | $34.3_{-6.4}$ | $51.6_{-6.0}$ | $24.2_{-4.9}$ | $66.2_{-0.5}$ | $34.3_{-6.0}$ | $64.4_{-1.6}$ | $46.3_{-0.3}$ | $70.9_{-0.3}$ |
| DPO+KL | Original | 30.1 | 47.3 | 27.7 | 58.1 | 34.0 | 57.3 | 41.7 | 61.1 |
| | w/ RNA | $29.8_{-0.3}$ | $56.4_{+9.1}$ | $29.1_{+1.4}$ | $67.0_{+8.9}$ | $33.3_{-0.7}$ | $64.7_{+7.4}$ | $42.8_{+1.1}$ | $68.9_{+7.8}$ |
| DPO+MSE | Original | 28.0 | 50.2 | 19.2 | 61.1 | 28.4 | 61.5 | 36.8 | 65.9 |
| | w/ RNA | $27.7_{-0.3}$ | $55.6_{+5.4}$ | $23.5_{+4.3}$ | $57.5_{-3.6}$ | $30.5_{+2.1}$ | $58.4_{-3.1}$ | $39.6_{+2.8}$ | $64.3_{-1.6}$ |
| SimNPO+KL | Original | 29.1 | 49.9 | 27.0 | 61.1 | 34.7 | 60.7 | 41.7 | 64.2 |
| | w/ RNA | $30.1_{+1.0}$ | $55.0_{+5.1}$ | $27.0_{+0.0}$ | $61.7_{+0.6}$ | $35.4_{+0.7}$ | $60.8_{+0.1}$ | $44.2_{+2.5}$ | $67.0_{+2.8}$ |
| SimNPO+MSE | Original | 35.4 | 52.3 | 29.8 | 66.7 | 36.8 | 66.8 | 44.5 | 72.3 |
| | w/ RNA | $38.2_{+3.8}$ | $58.6_{+6.3}$ | $31.5_{+1.7}$ | $71.9_{+5.2}$ | $42.8_{+6.0}$ | $70.1_{+3.3}$ | $48.4_{+3.9}$ | $75.7_{+3.4}$ |

Table 6: Accuracy under attack (AuA ↑) and ROUGE-L (↑) of unlearning methods on adversarial perturbed MMLU, comparing Original vs. **w/ RNA**. Improvements are shown in blue, drops in red.

## G    EFFECTS OF RNA ON MODEL ALIGNMENT

PO unlearning methods like DPO are themselves alignment techniques. RNA enhances robustness by increasing the diversity (via random noise) of the retain-representations. This could create *potential conflicts with the precision of model alignment*. We evaluate the RNA's effect on the model's alignment, such as faithfulness and hallucination on TruthfulQA (Lin et al., 2022) (multiple-choice QA) and ToxiGen (Hartvigsen et al., 2022), commonsense reasoning on WinoGrande (Sakaguchi

| Methods | | TruthfulQA | ToxiGen | WinoGrande | CommonsenseQA | HellaSwag | ARC E. | ARC C. | BoolQ |
|---|---|---|---|---|---|---|---|---|---|
| Base | Original | 38.5 | 45.2 | 72.3 | 66.1 | 63.9 | 81.2 | 57.0 | 84.9 |
| | | | | *Representation Misdirection* | | | | | |
| RMU | Original | 38.6 | 45.1 | 72.8 | 65.3 | 63.7 | 80.6 | 56.3 | 84.5 |
| | w/ RNA | $37.8_{-0.8}$ | $44.3_{-0.8}$ | $72.4_{-0.4}$ | $65.5_{+0.2}$ | $63.7_{+0.0}$ | $80.3_{-0.3}$ | $55.9_{-0.4}$ | $84.3_{-0.2}$ |
| Adap. RMU | Original | 38.6 | 45.5 | 72.3 | 65.7 | 63.6 | 80.8 | 55.6 | 84.5 |
| | w/ RNA | $38.8_{+0.2}$ | $44.1_{-1.4}$ | $73.0_{+0.7}$ | $65.8_{+0.1}$ | $63.5_{-0.1}$ | $80.3_{-0.5}$ | $56.1_{+0.5}$ | $84.4_{-0.1}$ |
| RSV | Original | 39.6 | 46.0 | 72.3 | 64.4 | 63.6 | 80.6 | 56.8 | 84.5 |
| | w/ RNA | $39.5_{-0.1}$ | $45.4_{-0.6}$ | $71.6_{-0.7}$ | $64.5_{+0.1}$ | $63.3_{-0.3}$ | $80.3_{-0.3}$ | $56.4_{-0.4}$ | $84.3_{-0.2}$ |
| | | | | *Preference Optimization* | | | | | |
| NPO+KL | Original | 42.9 | 45.0 | 70.8 | 64.1 | 61.8 | 80.0 | 56.7 | 84.7 |
| | w/ RNA | $40.3_{-2.6}$ | $44.5_{-0.5}$ | $70.7_{-0.1}$ | $62.9_{-1.2}$ | $61.8_{+0.0}$ | $80.2_{+0.2}$ | $56.5_{-0.2}$ | $84.6_{-0.1}$ |
| NPO+MSE | Original | 37.3 | 45.1 | 72.1 | 62.7 | 63.0 | 80.6 | 56.4 | 85.2 |
| | w/ RNA | $36.3_{-1.0}$ | $44.7_{-0.4}$ | $72.3_{+0.2}$ | $62.2_{-0.5}$ | $62.9_{-0.1}$ | $80.7_{+0.1}$ | $56.4_{+0.0}$ | $85.2_{+0.0}$ |
| DPO+KL | Original | 39.5 | 45.9 | 71.8 | 62.0 | 61.4 | 79.4 | 55.1 | 84.4 |
| | w/ RNA | $38.0_{-1.5}$ | $43.9_{-2.0}$ | $69.7_{-2.1}$ | $62.3_{+0.3}$ | $61.6_{+0.2}$ | $79.6_{+0.2}$ | $55.7_{+0.6}$ | $84.8_{+0.4}$ |
| DPO+MSE | Original | 34.2 | 44.7 | 72.1 | 56.7 | 62.3 | 78.9 | 54.1 | 84.3 |
| | w/ RNA | $32.1_{-2.1}$ | $42.5_{-2.2}$ | $71.6_{-0.5}$ | $57.2_{+0.5}$ | $61.6_{-0.7}$ | $78.8_{-0.1}$ | $53.4_{-0.7}$ | $84.4_{+0.1}$ |
| SimNPO+KL | Original | 43.3 | 44.0 | 71.5 | 63.8 | 62.0 | 80.0 | 56.4 | 84.5 |
| | w/ RNA | $42.1_{-1.2}$ | $45.2_{+1.2}$ | $71.1_{-0.4}$ | $61.9_{-1.9}$ | $61.9_{-0.1}$ | $79.9_{-0.1}$ | $56.4_{+0.0}$ | $84.8_{+0.3}$ |
| SimNPO+MSE | Original | 38.3 | 45.8 | 71.5 | 63.8 | 62.9 | 80.6 | 56.8 | 85.6 |
| | w/ RNA | $38.3_{+0.0}$ | $43.4_{-2.4}$ | $70.6_{-0.9}$ | $62.3_{-1.5}$ | $62.5_{-0.4}$ | $80.3_{-0.3}$ | $56.3_{-0.5}$ | $85.2_{-0.4}$ |

Table 7: Performance of unlearning methods on 8 tasks, comparing Original vs. **w/ RNA**. Improvements are shown in blue, drops in red.

et al., 2021) and CommonsenseQA (Talmor et al., 2019), natural language inference/completion on HellaSwag (Zellers et al., 2019), science reasoning on ARC (Clark et al., 2018) (easy and challenge), and factuality on BoolQ (Clark et al., 2019). Performance of unlearning methods on these tasks is shown in Table 7. As we can observe that RNA does not systematically degrade alignment, and overall alignment performance remains stable.

## H  EFFECTS OF RNA ON CHAIN-OF-THOUGHT PROMPTING

Chain-of-Thought (CoT; Wei et al. (2022)) is one of the most commonly used prompting techniques for improving LLM reasoning capabilities. The effect of RNA on CoT is a fairly interesting point that might need to be investigated. We conducted additional experiments on GSM8K Cobbe et al. (2021) and GPQA Rein et al. (2024) with zero-shot, 4-shot, and 8-shot CoT with Zephyr-7B model. The results shown in Tab. 8 demonstrated that noise added by RNA introduces minor effects on CoT.

## I  PERFORMANCE OF OTHER MODELS

Our experiments in the main text are based on the Zephyr-7B model, which serves as a representative setup. To assess RNA's generalization beyond the original setup, we conducted additional experiments using the Llama-3-8B (Dubey et al., 2024) and Mistral-7B Jiang et al. (2023) models on two representative unlearning methods from 2 classes: RMU and NPO+KL, across various tasks. These results provide further empirical evidence for RNA's generalization and robustness.

**Hyperparameters.** All models are fine-tuned using AdamW (Loshchilov & Hutter, 2019) for $T = 500$ update steps with a learning rate of $5 \times 10^{-5}$, batch size 4, and maximum sequence length of 500 for WMDP-Biology and 768 for WMDP-Cyber. The unlearned layer is fixed at $l = 7$. Retain weights are set to $\alpha_{\text{biology}} = \alpha_{\text{cyber}} = 1200$ for both models. The coefficient values are $c_{\text{biology}} = c_{\text{cyber}} = 20$ for Llama-3-8B and $c_{\text{biology}} = c_{\text{cyber}} = 6.5$ for Mistral-7B. For NPO+KL, we perform a grid search over $(\alpha_{\text{biology}}, \alpha_{\text{cyber}})$ and select $(5, 10)$ for Llama-3-8B and $(30, 40)$ for Mistral-7B. For RMU w/ RNA, we set the perturbed layer to $l = 7$, tune the noise scale via grid search, and report the best performance at $\nu = 7 \times 10^{-2}$ (Llama-3-8B) and $\nu = 3 \times 10^{-2}$ (Mistral-7B). For

| Method | | GSM8K | | | GPQA | | |
|---|---|---|---|---|---|---|---|
| | | CoT zero-shot | CoT 4-shot | CoT 8-shot | CoT zero-shot | CoT 4-shot | CoT 8-shot |
| Base | Original | 15.3 | 38.9 | 42.2 | 12.0 | 22.3 | 28.3 |
| RMU | Original | 15.1 | 37.4 | 40.8 | 12.0 | 24.5 | 21.8 |
| | w/ RNA | $13.1_{-2.0}$ | $36.5_{-0.9}$ | $40.6_{-0.2}$ | $12.0_{+0.0}$ | $24.3_{-0.2}$ | $24.1_{+2.3}$ |
| Adaptive RMU | Original | 12.9 | 36.7 | 41.5 | 10.9 | 25.2 | 21.6 |
| | w/ RNA | $15.1_{+2.2}$ | $37.5_{+0.8}$ | $41.0_{-0.5}$ | $12.2_{+1.3}$ | $19.8_{-5.4}$ | $23.4_{+1.8}$ |
| RSV | Original | 17.4 | 36.7 | 42.5 | 8.2 | 25.4 | 21.4 |
| | w/ RNA | $16.9_{-0.5}$ | $37.5_{+0.8}$ | $42.8_{+0.3}$ | $10.4_{+2.2}$ | $23.2_{-2.2}$ | $25.6_{+4.2}$ |
| NPO+KL | Original | 14.2 | 36.2 | 40.1 | 10.4 | 27.0 | 21.6 |
| | w/ RNA | $14.7_{+0.5}$ | $36.7+0.5$ | $38.9_{-1.2}$ | $9.3_{-1.1}$ | $22.7_{-4.3}$ | $23.6_{+2.0}$ |
| NPO+MSE | Original | 10.6 | 37.6 | 41.0 | 11.3 | 26.1 | 22.3 |
| | w/ RNA | $11.2_{+0.6}$ | $35.7_{-1.9}$ | $38.8_{-2.2}$ | $9.1_{-2.2}$ | $23.4_{-2.7}$ | $21.4_{-0.9}$ |
| DPO+KL | Original | 11.9 | 36.1 | 37.2 | 11.3 | 23.2 | 19.8 |
| | w/ RNA | $11.3_{-0.6}$ | $36.9_{+0.8}$ | $38.7_{+1.5}$ | $11.3_{+0.0}$ | $23.2_{+0.0}$ | $19.8_{+0.0}$ |
| DPO+MSE | Original | 10.0 | 36.0 | 39.8 | 11.6 | 23.8 | 22.5 |
| | w/ RNA | $14.9_{+4.9}$ | $37.5_{+1.5}$ | $40.5_{+0.7}$ | $14.2_{+2.6}$ | $24.3_{+0.5}$ | $24.1_{+1.6}$ |
| SimNPO+KL | Original | 15.6 | 36.5 | 41.0 | 11.1 | 20.9 | 18.7 |
| | w/ RNA | $17.8_{+2.2}$ | $37.5_{+1.0}$ | $41.8_{+0.8}$ | $8.0_{-3.1}$ | $23.6_{+2.7}$ | $20.3_{+1.6}$ |
| SimNPO+MSE | Original | 11.0 | 38.2 | 39.5 | 8.2 | 24.5 | 24.3 |
| | w/ RNA | $11.0_{+0.0}$ | $37.9_{-0.3}$ | $40.2_{+0.7}$ | $13.6_{+5.4}$ | $25.6_{+1.1}$ | $23.2_{-1.1}$ |

Table 8: Effects of RNA on Chain-of-Thought Prompting.

| Models & Methods | Llama-3-8B | | | | Mistral-7B | | | |
|---|---|---|---|---|---|---|---|---|
| | RMU | | NPO+KL | | RMU | | NPO+KL | |
| | Original | w/ RNA | Original | w/ RNA | Original | w/ RNA | Original | w/ RNA |
| WMDP ($\downarrow$) | 31.4 | $34.6_{-3.2}$ | 27.9 | $33.2_{-5.3}$ | 31.7 | $31.7_{+0.0}$ | 29.3 | $34.0_{-4.7}$ |
| MMLU ($\uparrow$) | 60.3 | $60.2_{-0.1}$ | 53.8 | $54.4_{+0.6}$ | 58.2 | $58.6_{+0.4}$ | 56.5 | $56.4_{-0.1}$ |
| Perturbed MMLU ($\uparrow$) | 34.4 | $47.3_{+12.9}$ | 26.2 | $47.3_{+21.1}$ | 27.2 | $42.2_{+15.0}$ | 31.4 | $53.5_{+22.1}$ |
| MMLU C. Bio. ($\uparrow$) | 34.7 | $60.4_{+25.7}$ | 30.5 | $55.5_{+25.0}$ | 25.0 | $38.1_{+13.1}$ | 32.6 | $52.0_{+19.4}$ |
| MMLU C. Sec. ($\uparrow$) | 29.0 | $33.0_{+4.0}$ | 30.0 | $46.0_{+16.0}$ | 33.0 | $46.0_{+13.0}$ | 35.0 | $30.0_{-5.0}$ |

Table 9: Performance of Llama-3-8B and Mistral-7B on WMDP and MMLU, Perturbed MMLU, MMLU subsets benchmarks using RMU and NPO+KL, comparing Original vs. **w/ RNA**. Improvements are shown in blue, drops in red.

NPO+KL w/ RNA, we also perturb layer $l = 7$ and select the best scales $\nu = 6 \times 10^{-2}$ (Llama-3-8B) and $\nu = 3 \times 10^{-3}$ (Mistral-7B).

**Results.** As shown in Table 9, across Llama-3-8B and Mistral-7B, RNA significantly enhances unlearning robustness of models while introducing a small trade-off on forget performance. For forget-tasks (WMDP-Biology and Cyber), RNA slightly increases the accuracy, *e.g.,* RMU on Llama-3-8B drops from $31.4 \rightarrow 34.6$ ($-3.2$) and NPO+KL from $27.9 \rightarrow 33.2$ ($-5.3$). For retain-tasks (MMLU and Perturbed MMLU and MMLU subsets), RNA substantially improves performance, particularly on perturbed MMLU and MMLU subsets (C. Bio. and C. Sec.). For example, perturbed MMLU on Llama-3-8B with NPO+KL improves from $26.2 \rightarrow 47.3$ ($+21.1$), and MMLU C. Bio. from $30.5 \rightarrow 55.5$ ($+25.0$). Overall, RNA effectively recovers or enhances accuracy on retain-tasks while slightly compromising forget-task performance, demonstrating a favorable trade-off between unlearning and model robustness.

| Method | | VerbMem$_f$ ↓ | KnowMem$_f$ ↓ | KnowMem$_r$ (benign) ↑ | KnowMem$_r$ (perturbed) ↑ |
|---|---|---|---|---|---|
| Base model (MUSE-news_target) | | 57.2 | 64.2 | 64.2 | 51.8 |
| RMU ($c = 100$) | Original | 57.3 | 65.5 | 55.0 | 51.0 (**under-unlearn**) |
| | w/ RNA ($\nu = 1e{-}3$) | 56.7 | 66.1 | 56.0 | 49.3 |
| | w/ RNA ($\nu = 2e{-}3$) | 56.6 | 65.7 | 55.1 | 50.8 |
| | w/ RNA ($\nu = 3e{-}3$) | 56.7 | 66.1 | 55.8 | 50.4 |
| RMU ($c = 110$) | Original | 56.5 | 66.1 | 55.1 | 50.2 (**under-unlearn**) |
| | w/ RNA ($\nu = 1e{-}3$) | 56.4 | 66.1 | 55.7 | 50.7 |
| | w/ RNA ($\nu = 2e{-}3$) | 56.6 | 64.3 | 54.9 | 49.8 |
| | w/ RNA ($\nu = 3e{-}3$) | 55.3 | 65.0 | 54.9 | 50.8 |
| RMU ($c = 120$) | Original | 56.2 | 64.1 | 55.8 | 49.6 (**under-unlearn**) |
| | w/ RNA ($\nu = 1e{-}3$) | 55.8 | 65.9 | 55.2 | 50.1 |
| | w/ RNA ($\nu = 2e{-}3$) | 55.9 | 66.3 | 55.3 | 50.2 |
| | w/ RNA ($\nu = 3e{-}3$) | 56.0 | 66.1 | 55.9 | 50.8 |
| RMU ($c = 130$) | Original | 54.1 | 56.2 | 49.0 | 45.4 (**under-unlearn**) |
| | w/ RNA ($\nu = 1e{-}3$) | 55.7 | 65.2 | 56.3 | 49.8 |
| | w/ RNA ($\nu = 2e{-}3$) | 55.9 | 65.2 | 55.9 | 50.0 |
| | w/ RNA ($\nu = 3e{-}3$) | 55.0 | 66.0 | 56.0 | 50.4 |
| RMU ($c = 140$) | Original | 53.9 | 43.2 | 36.3 | 38.1 (**under-unlearn**) |
| | w/ RNA ($\nu = 1e{-}3$) | 54.6 | 62.9 | 51.9 | 48.2 |
| | w/ RNA ($\nu = 2e{-}3$) | 55.0 | 62.8 | 52.8 | 49.1 |
| | w/ RNA ($\nu = 3e{-}3$) | 55.7 | 64.9 | 54.5 | 50.2 |
| RMU ($c = 150$) | Original | 49.2 | 13.7 | 18.0 | 18.7 (**over-unlearn**) |
| | w/ RNA ($\nu = 1e{-}3$) | 53.6 | 48.6 | 43.5 | 37.8 |
| | w/ RNA ($\nu = 2e{-}3$) | 54.2 | 51.4 | 44.2 | 39.6 |
| | w/ RNA ($\nu = 3e{-}3$) | 54.3 | 53.9 | 51.0 | 45.3 |

Table 10: Performance of RMU with and without RNA under over-unlearn and under-unlearn in MUSE News.

## J  EXPERIMENTS ON MUSE

**Setup.** We employ MUSE-NEWS_TARGET[1] as the base model for unlearning. MUSE-NEWS_TARGET is the Llama-2-7b (Touvron et al., 2023) model fine-tuned on the News corpus (BBC news articles). We employ two representative unlearning methods, RMU and NPO+KL. Results are shown in Figure 10 and Figure 11. Overall, we found that RNA fails to enhance model retain-robustness when unlearning is miscalibrated.

**Hyperparameters.** For RMU, we perform a heuristic search over the coefficient $c \in [100, 120, 130, 140, 150]$. We set the retain-weight $\alpha_r = 1200$ (coefficient of forget-loss), forget-weight $\alpha_f = 1.0$ (coefficient of retain-loss), $T = 500$ gradient steps, unlearn with layer $l = 7$, RNA noise added at layer 7, learning rate $2e - 5$, maximum sequence length 256. MUSE-News forget-set is used as $\mathcal{D}_f$, Wikitext is used as $\mathcal{D}_r$. For NPO+KL, we search over $(\alpha_f, \alpha_r) \in [(1, 1), (5, 1), (10, 1), (20, 1), (1, 5), (1, 10), (1, 20)]$, $\beta$ is set to 0.1. For each pair $(\alpha_f, \alpha_r)$, we grid search RNA's noise scale $\nu \in [1e - 3, 2e - 3, 3e - 3]$. We report VerbMem and KnowMem.

## K  LIMITATION AND FUTURE WORK

We posit the following limitations of this study and discuss potential future works.

We have evaluated our methods primarily on WMDP, a widely used and representative benchmark. We acknowledge the existence of other benchmarks, such as MUSE (Shi et al., 2025) and TOFU (Maini et al., 2024), but these are less suitable for our experimental setup. Specifically, TOFU is designed to remove the influence of specific data points, making it less applicable in generative settings. While MUSE could be suitable, previous work (Shi et al., 2025) has shown that methods evaluated on MUSE often exhibit over-forgetting or under-forgetting. Since our study focuses on retain-robustness, which requires a careful balance between forgetting and retaining, MUSE is not ideal. These factors make it challenging to directly apply MUSE and TOFU in our current experiments.

---

[1]`https://huggingface.co/muse-bench/MUSE-news_target`

| Method | | VerbMem$_f$ ↓ | KnowMem$_f$ ↓ | KnowMem$_r$ (benign) ↑ | KnowMem$_r$ (perturbed) ↑ |
|---|---|---|---|---|---|
| Base model (MUSE-news_target) | | 57.2 | 64.2 | 64.2 | 51.8 |
| NPO+KL $(1, 1)$ | Original | 57.5 | 61.4 | 52.6 | 48.8 (**under-unlearn**) |
| | w/ **RNA** ($\nu = 1e-3$) | 56.8 | 61.0 | 52.0 | 47.6 |
| | w/ **RNA** ($\nu = 2e-3$) | 56.8 | 61.0 | 52.8 | 46.6 |
| | w/ **RNA** ($\nu = 3e-3$) | 56.5 | 60.2 | 52.7 | 46.5 |
| NPO+KL $(1, 5)$ | Original | 57.5 | 59.5 | 52.7 | 47.8 (**under-unlearn**) |
| | w/ **RNA** ($\nu = 1e-3$) | 58.0 | 59.9 | 53.8 | 47.0 |
| | w/ **RNA** ($\nu = 2e-3$) | 56.3 | 60.4 | 53.6 | 46.8 |
| | w/ **RNA** ($\nu = 3e-3$) | 57.5 | 59.8 | 53.0 | 46.7 |
| NPO+KL $(1, 10)$ | Original | 58.6 | 60.0 | 52.6 | 45.7 (**under-unlearn**) |
| | w/ **RNA** ($\nu = 1e-3$) | 58.7 | 59.8 | 53.0 | 46.4 |
| | w/ **RNA** ($\nu = 2e-3$) | 58.3 | 59.9 | 53.0 | 46.7 |
| | w/ **RNA** ($\nu = 3e-3$) | 58.9 | 60.1 | 53.9 | 46.5 |
| NPO+KL $(1, 20)$ | Original | 57.7 | 59.5 | 54.3 | 47.4 (**under-unlearn**) |
| | w/ **RNA** ($\nu = 1e-3$) | 58.2 | 61.2 | 53.8 | 47.6 |
| | w/ **RNA** ($\nu = 2e-3$) | 58.8 | 62.8 | 53.5 | 46.5 |
| | w/ **RNA** ($\nu = 3e-3$) | 58.2 | 62.8 | 54.7 | 47.7 |
| NPO+KL $(5, 1)$ | Original | 56.8 | 59.9 | 51.8 | 48.6 (**under-unlearn**) |
| | w/ **RNA** ($\nu = 1e-3$) | 55.7 | 60.0 | 52.6 | 47.8 |
| | w/ **RNA** ($\nu = 2e-3$) | 56.9 | 59.4 | 53.1 | 48.5 |
| | w/ **RNA** ($\nu = 3e-3$) | 56.7 | 60.1 | 51.9 | 47.7 |
| NPO+KL $(10, 1)$ | Original | 57.3 | 60.6 | 53.1 | 48.4 (**under-unlearn**) |
| | w/ **RNA** ($\nu = 1e-3$) | 56.7 | 61.5 | 52.3 | 48.2 |
| | w/ **RNA** ($\nu = 2e-3$) | 55.7 | 59.9 | 52.1 | 48.1 |
| | w/ **RNA** ($\nu = 3e-3$) | 57.1 | 59.3 | 52.6 | 48.3 |
| NPO+KL $(20, 1)$ | Original | 56.4 | 58.2 | 52.1 | 48.2 (**under-unlearn**) |
| | w/ **RNA** ($\nu = 1e-3$) | 56.6 | 59.3 | 52.1 | 47.8 |
| | w/ **RNA** ($\nu = 2e-3$) | 56.6 | 60.6 | 51.9 | 48.1 |
| | w/ **RNA** ($\nu = 3e-3$) | 56.1 | 60.8 | 51.7 | 47.9 |

Table 11: Performance of NPO+KL with and without RNA under over-unlearn and under-unlearn in MUSE News.

Due to computational constraints, experiments are conducted only on the 7B or 8B models and with updates to a limited set of layer parameters, which may risk overlooking interesting aspects of generalization. Although RNA has demonstrated effectiveness, it relies heavily on hyperparameter grid search to identify an optimal noise scale, making it computationally expensive for extremely large models with hundreds of billions of parameters.

RNA is designed for fine-tuning-based unlearning, which requires access to the model parameters. Future work exploring RNA as an inference-time intervention method could be promising. One potential direction is to analyze the optimal coefficient or noise scale to further improve RNA's effectiveness.

## L    BROADER IMPACT

We establish a novel theoretical framework that bridges the connection between machine unlearning and backdoor attacks, providing crucial insights into the vulnerabilities of unlearned models. Our theoretical and empirical analysis provides a valuable solution for developing more secure and reliable machine learning systems.

## M    AI USAGE DECLARATION

AI tools were used for grammar checking and formatting of tables and figures. All technical content and implementations were written by the authors.

