# OpenReview forum: "Improving LLM Unlearning Robustness via Random Perturbations"
_ICLR.cc/2026/Conference — Submitted to ICLR 2026_

### Official Review · Reviewer_1B8L · 2025-10-24

**Soundness:** 2
**Presentation:** 3
**Contribution:** 2
**Rating:** 2
**Confidence:** 5

**Summary:**

This paper studies why current large language model unlearning methods often weaken model robustness, leading to errors when retain queries accidentally include forget tokens. The authors show that the unlearning process behaves like a backdoor attack in which forget tokens act as triggers that reintroduce forgotten behavior. To mitigate this issue, the paper reframes the retaining process as a backdoor defense and proposes Random Noise Augmentation, a simple and model agnostic technique that injects small Gaussian noise into retain representations during fine tuning. Theoretical and empirical results demonstrate that this method improves the robustness of unlearned models while maintaining their forgetting and retaining performance.

**Strengths:**

1. The paper introduces a fresh angle by connecting LLM unlearning with backdoor attack and defense mechanisms, offering a clear framework for understanding existing weaknesses.

2. Theoretical analysis provides partial support for the proposed method, though it relies on many assumptions that weaken its rigor.

3. The writing is clear, and the figures are well-presented.

**Weaknesses:**

1. The experiments lack sufficient baselines. The authors only study a few non-robust LLM unlearning methods, while many recent approaches have been specifically proposed to enhance robustness in LLM unlearning [1-2].
2. The experimental evaluation is narrow, focusing only on the WMDP benchmark. How does the proposed method perform on other benchmarks such as TOFU [3] and MUSE[4]?
3. Using only MMLU to evaluate model utility is limited. It would be more convincing to include additional utility metrics[5].
4. The paper does not discuss how the proposed method behaves under other types of attacks, such as relearning attacks or adversarial prompts [6].
5. Many proofs rely on overly strong assumptions. For example, Eq. (9) asserts that adding Gaussian perturbations increases the expected loss, which depends on a strong assumption of local convexity or a positive-definite Hessian. In deep LLM latent spaces, this condition often fails, so the inequality $\mathbb{E}[\ell(y|z{+}v)]>\ell(y|z)$ is not generally guaranteed.

> [1] Tamirisa R, Bharathi B, Phan L, et al. Tamper-resistant safeguards for open-weight llms[J]. arXiv preprint arXiv:2408.00761, 2024.
>
> [2] Fan, Chongyu, et al. "Towards llm unlearning resilient to relearning attacks: A sharpness-aware minimization perspective and beyond." arXiv preprint arXiv:2502.05374 (2025).
>
> [3] Maini, Pratyush, et al. "Tofu: A task of fictitious unlearning for llms." arXiv preprint arXiv:2401.06121 (2024).
>
> [4] Shi, Weijia, et al. "Muse: Machine unlearning six-way evaluation for language models." arXiv preprint arXiv:2407.06460 (2024).
>
> [5] Che Z, Casper S, Kirk R, et al. Model tampering attacks enable more rigorous evaluations of llm capabilities[J]. arXiv preprint arXiv:2502.05209, 2025.
>
> [6] Łucki, Jakub, et al. "An adversarial perspective on machine unlearning for ai safety." arXiv preprint arXiv:2409.18025 (2024).

**Questions:**

See weakness.

---

> ### Author Response · Authors · 2025-11-21
> **Official Comment by Authors (1/2)**
>
> We thank the Reviewer for the comments. Please see our response to your concern below.
>
> > **Q**: The experiments lack sufficient baselines. The authors only study a few non-robust LLM unlearning methods, while many recent approaches have been specifically proposed to enhance robustness in LLM unlearning [1-2].
>
> **A**: We thank the Reviewer for this suggestion. We'd clarify that, as detailed in the paper, we specifically address **retain-robustness**, **the unlearned model's vulnerability to benign, non-adversarial retain-queries that inadvertently contain forget-tokens or overlap semantically with forget-sets**. Our key insights are (1) explain why standard unlearning methods (6 PO and 3 RM) induce this fragility through the view of backdoor trigger mechanism, and (2) how RNA mitigates it via latent space smoothing. **Our study and the problem studied in [1] [2] target different robustness scenarios**.
>
> Specifically, [1] introduced Tampering Attack Resistance (TAR), a defense method against fine-tuning based tampering attacks by adversaries with white-box access. This threat model assumes malicious parameter updates, not the benign input perturbations. [2] applied SAM to enhance **forget-robustness against relearning attacks**. Again, this fundamental differs from our benign retain-robustness scope and threat model.
>
> > **Q**: Using only MMLU to evaluate model utility is limited.
>
> **A**: MMLU is a comprehensive benchmark spanning 57 diverse domains of general knowledge, and it is widely used in LLM unlearning. We follow previous works that used MMLU for fair comparison. Additionally, Tab. 7 assesses RNA's performance across multiple tasks, e.g., TruthfulQA, ToxiGen, WinoGrande, CommonsenseQA, HellaSwag, ARC-Easy/Challenge, and BoolQ. This shows that RNA's performance across them remains stable or improves slightly in some cases.
> >**Q**: It would be more convincing to include additional utility metrics[5]. **Q**: The paper does not discuss how the proposed method behaves under other types of attacks, such as relearning attacks or adversarial prompts [6].
>
> **A**: We'd clarify that [6] study jailbreak methods such as benign fine-tuning, orthogonalization, logitlens, and enhanced GCG can effectively *recover* forgotten knowledge, which falls under forget-robustness. [5] employs tampering attacks aimed at measuring forget-robustness of unlearned models, which is distinct from our focus on retain-robustness.
> >**Q**: Many proofs rely on overly strong assumptions. For example, Eq. (9) asserts that adding Gaussian perturbations increases the expected loss, which depends on a strong assumption of local convexity or a positive-definite Hessian. In deep LLM latent spaces, this condition often fails, so the inequality $\mathbb{E}[\ell(y|z+v) ] > \ell(y|z)$ is not generally guaranteed.
>
> **A**: We first agree with the Reviewer that the local convexity assumption is **hard to theoretically guarantee** due to the highly non-linear property of LLMs.  However, we note that our assumption is reasonable rather than overly restrictive. We **already conducted the empirical experiment** to validate how RM methods affect the loss of forget-samples. Empirical results presented in Appx C.1 (Fig. 8) align with the assumption and further support the analysis in Sec. 3.

---

> ### Author Response · Authors · 2025-11-21
> **Official Comment by Authors (2/2)**
>
> > **Q**: The experimental evaluation is narrow, ... MUSE[4]?
>
> **A**: WMDP is a representative benchmark for LLM unlearning, our experiment provides deep, thorough analysis across multiple dimensions: widely used unlearning approaches (RM & PO with 9 methods), multiple robustness scenarios (synthetic perturbations, real retain-queries, n-gram similarity), and ablation studies (effects of RNA when applied to different latent spaces, RNA's side effects on model alignment, adversarial robustness). The theoretical insight is scale-independent.
>
> We conducted additional experiments on MUSE with two representative methods, RMU and NPO+KL.
>
> Setup. We employ MUSE-news-target model as the base model for unlearning. This model is the Llama-2-7b fine-tuned on News articles. We employ two representative unlearning methods, RMU and NPO+KL. For RMU, we perform a heuristic search over the coefficient $c \in [100, 120, 130, 140, 150]$. We set the retain-weight $\alpha_r=1200$ (coefficient of forget-loss), forget-weight $\alpha_f = 1.0$ (coefficient of retain-loss), $T=500$ steps, unlearn layer $l=7$, RNA noise added at $l=7$, learning rate $1e-5$, max sequence length $256$. MUSE-News forget-set is used as $\mathcal{D}_f$, Wikitext is used as $\mathcal{D}_r$. For NPO+KL, search over $(\alpha_f, \alpha_r) \in [(1,1), (5,1), (10,1), (20,1), (1,5), (1,10), (1,20)]$, $\beta = 0.1$.  For each pair $(\alpha_f, \alpha_r)$, we grid search RNA's noise scale $\nu \in [1e-3, 2e-3, 3e-3]$. We report VerbMem and KnowMem. We create a *perturbed prompt* as a QA prompt containing multiple forget QAs followed by a retain QA, while a *benign prompt* contains multiple retain QAs.
>
> |Unlearned models|RNA ($\nu$)|VerbMem_f↓|KnowMem_f↓|KnowMem_r (Benign)↑|KnowMem_r (Perturbed)↑|
> |-|-|-|-|-|-|
> |RMU (c=100)|w/o RNA|57.3|65.5|55.0|51.0 ($\text{under-unlearn}$)|
> ||w/ RNA (1e-3)|56.7|66.1|56.0|49.3|
> ||w/ RNA (2e-3)|56.6|65.7|55.1|50.8|
> ||w/ RNA (3e-3)|56.7|66.2|55.8|50.4|
> |RMU (c=110)|w/o RNA| 56.5|66.1|55.1|50.2|
> ||w/ RNA (1e-3)|56.4|66.1|55.7|50.7|
> ||w/ RNA (2e-3)|56.6|64.3|54.9|49.8|
> ||w/ RNA (3e-3)|55.3|65.0|54.9|50.8|
> |RMU (c=120)| w/o RNA|56.2|64.1|55.8|49.6 ($\text{under-unlearn}$)|
> ||w/ RNA (1e-3)|55.8|65.9|55.2|50.1|
> ||w/ RNA (2e-3)|55.9|66.3|55.3|50.2|
> ||w/ RNA (3e-3)|56.0|66.1|55.9|50.8|
> |RMU (c=130)|w/o RNA|54.1|56.2|49.0| 45.4 ($\text{under-unlearn}$)|
> ||w/ RNA (1e-3)|55.7|65.2|56.3|49.8|
> ||w/ RNA (2e-3)|55.9|65.2|55.9|50.0|
> ||w/ RNA (3e-3)|55.0|66.0|56.0|50.4|
> |RMU (c=140)|w/o RNA|53.9|43.2|36.3|38.1|
> ||w/ RNA (1e-3)|54.6| 62.9|51.9|48.2|
> ||w/ RNA (2e-3)|55.0| 62.8|52.8| 49.1|
> ||w/ RNA (3e-3)|55.7| 64.9|54.5| 50.2|
> |RMU (c=150)|w/o RNA|49.2|13.7|18.0| 18.7 ($\text{over-unlearn}$)|
> ||w/ RNA (1e-3)|53.6|48.6|43.5|37.8|
> ||w/ RNA (2e-3)|54.2|51.4|44.2|39.6|
> ||w/ RNA (3e-3)|54.3|53.9|51.0|45.3|
>
> |Unlearned models $(\alpha_f,\alpha_r)$|RNA (Noise Scale $\nu$)|VerbMem_f↓|KnowMem_f↓|KnowMem_r (Benign)↑|KnowMem_r (Perturbed)↑|
> |-|-|-|-|-|-|
> |NPO+KL$(1,1)$|w/o RNA|57.5|61.4|52.6|48.8 ($\text{under-unlearn}$)|
> ||w/ RNA ($1e-3$)|56.8|61.0|52.0|47.6|
> ||w/ RNA ($2e-3$)|56.8|61.0|52.8|46.6|
> ||w/ RNA ($3e-3$)|56.5|60.2|52.7|46.5|
> |NPO+KL $(1,5)$|w/o RNA|57.5|59.5|52.7|47.8 ($\text{under-unlearn}$)|
> ||w/ RNA ($1e-3$)|58.0|59.9|53.8|47.0|
> ||w/ RNA ($2e-3$)|56.3|60.4|53.6|46.8|
> ||w/ RNA ($3e-3$)|57.5|59.8|53.0|46.7|
> |NPO+KL $(1,10)$|w/o RNA|58.6|60.0|52.6|45.7 ($\text{under-unlearn}$)|
> ||w/ RNA ($1e-3$)|58.7|59.8|53.0|46.4|
> ||w/ RNA ($2e-3$)|58.3|59.9|53.0|46.7|
> ||w/ RNA ($3e-3$)|58.9|60.1|53.9|46.5|
> |NPO+KL $(1,20)$| w/o RNA|57.7|59.5|54.3|47.4 ($\text{under-unlearn}$)|
> ||w/ RNA ($1e-3$)|58.2|61.2|53.8|47.6|
> ||w/ RNA ($2e-3$)|58.8|62.8|53.5|46.5|
> ||w/ RNA ($3e-3$)|58.2|62.8|54.7|47.7|
> |NPO+KL $(5,1)$|w/o RNA|56.8|59.9|51.8|48.6|
> ||w/ RNA ($1e-3$)|55.7|60.0|52.6|47.8|
> ||w/ RNA ($2e-3$)|56.9|59.4|53.1|48.5|
> ||w/ RNA ($3e-3$)|56.7|60.1|51.9|47.7|
> |NPO+KL $(10,1)$|w/o RNA|57.3|60.6|53.1|48.4 ($\text{under-unlearn}$)|
> ||w/ RNA ($1e-3$)|56.7|61.5|52.3|48.2|
> ||w/ RNA ($2e-3$)|55.7|59.9|52.1|48.1|
> ||w/ RNA ($3e-3$)|57.1|59.3|52.6|48.3|
> |NPO+KL $(20,1)$|w/o RNA|56.4|58.2|52.1|48.2 ($\text{under-unlearn}$)|
> ||w/ RNA ($1e-3$)|56.6|59.3|52.1|47.8|
> ||w/ RNA ($2e-3$)|56.6|60.6|51.9|48.1|
> ||w/ RNA ($3e-3$)|56.1|60.8|51.7|47.9|
>
> We discuss the following points that could potentially be reasons behind RNA's failures: Our theory and main experiments hinge on a balanced unlearning where forget and retain objectives are both well established. In MUSE, several configs either under-unlearn (forget signals too weak) or over-unlearn (retain signals overly suppressed). The failure case we study, i.e., retain queries inadvertently activating "forget triggers" and causing *a sensitivity spike*, which are not established in under-unlearn or over-unlearn. RNA is *a variance-reduction defense against sensitivity caused by triggers, not a method for miscalibrated unlearning strength*. When *unlearning is not well-calibrated, small representation smoothing from RNA can be washed.*

---

> ### Author Response · Authors · 2025-11-28
>
> Dear Reviewer 1B8L,
>
> Thank you for taking the time to review our paper. Since the rebuttal will come to an end soon.
>
> We hope that our responses address your concerns. If any questions/concerns remain, we are happy to discuss further!
>
> Authors

---

> ### Author Response · Authors · 2025-12-01
> **Additional Evaluations on Model Utility**
>
> Dear Reviewer 1B8L,
>
> Please see additional evaluations for the following question.
>
> > Q: It would be more convincing to include additional utility metrics[5]
>
> **A**:  We acknowledge that the evaluation of "retain-robustness" relies on accuracy in multiple-choice QA, and it seems too coarse to capture the full extent of model degradation. A model could still select the correct answer (single answer token: "A", "B", "C", "D") but exhibit other undesirable behaviors, such as a drop in **coherence**.
>
> To further highlight the robustness of RNA, we **conducted additional experiments using the ROUGE-L/1/2 recall scores as an evaluation metric to assess the linguistic quality of generated responses**. ROUGE scores measure the overlap between generated text (from the unlearned model) and reference text (from the base model). Given a question in the perturbed MMLU dataset, we used greedy decoding to generate $15$ tokens, computed the ROUGE-L/1/2 recall score, and reported the mean value. The table below shows the performance of unlearned models:
>
> |Methods|-|ROUGE-L↑|ROUGE-1↑|ROUGE-2↑|
> |-|-|-|-|-|
> |RMU |w/o RNA|**55.5**|**56.9**|**41.3**|
> ||**w/ RNA**|55.0|56.4|41.0|
> |Adaptive RMU|w/o RNA|37.2|38.5|20.7|
> ||**w/ RNA**|**77.9**|**78.5**|**71.0**|
> |RSV|w/o RNA|48.5|49.4|35.4|
> ||**w/ RNA**|**78.0**|**78.5**|**71.0**|
> |RR|w/o RNA|13.6|13.8|0.6|
> ||**w/ RNA**|**28.3**|**29.4**|**12.9**|
> |NPO+KL|w/o RNA|12.4|12.4|0.8|
> ||**w/ RNA**|**40.6**|**41.4**|**26.3**|
> |NPO+MSE|w/o RNA|31.2|31.7|18.5|
> ||**w/ RNA**|**42.8**|**43.5**|**31.6**|
> |SimNPO+KL|w/o RNA|17.3|17.6|4.0|
> ||**w/ RNA**|**32.1**|**32.7**|**18.1**|
> |SimNPO+MSE|w/o RNA|21.0|21.3|7.1|
> ||**w/ RNA**|**67.4**|**68.2**|**57.8**|
> |DPO+KL|w/o RNA|28.9|29.4|15.0|
> ||**w/ RNA**|**31.2**|**31.8**|**18.6**|
> |DPO+MSE|w/o RNA|10.2|10.2|0.06|
> ||**w/ RNA**|**70.2**|**71.2**|**61.3**|
> |GA+KL|w/o RNA|10.7|10.7|0.4|
> ||**w/ RNA**|**27.0**|**27.7**|**14.1**|
> |GA+MSE|w/o RNA|16.6|16.9|4.0|
> ||**w/ RNA**|**74.5**|**75.2**|**65.7**|

---

### Official Review · Reviewer_wLnL · 2025-10-29

**Soundness:** 3
**Presentation:** 3
**Contribution:** 3
**Rating:** 4
**Confidence:** 4

**Summary:**

This paper propose Random Noise Augmentation, a method to avoid the misbehave when forget-tokens are in the retain-queries. The author frames current unlearning methods as inadvertently introducing a backdoor attack and proposes RNA as a backdoor defense to enhance the robustness of unlearned models against non-adversarial forget-tokens in retain-queries. By adding random perturbations in the data retaining stage, RNA significantly improves the model's robustness. The theoretical analysis of this paper focuses on how the presence of forget-tokens introduces randomness in latent representations, and proves a bounded probability for RNA to reject the misbehavior caused by forget-tokens.

**Strengths:**

1. The paper establishes a unified view of the two primary classes of LLM unlearning methods, Representation Misdirection and Preference Optimization, by analyzing both through the lens of the generative latent variable model.

2. The paper provides theoretical guarantees for RNA's effectiveness in improving the robustness of models by rejecting the detrimental effects caused by forget-tokens.

3. Comprehensive experiments are conducted. This extensive evaluation rigorously validates the effectiveness and generalizability of the proposed RNA method by including diverse unlearning methods and demonstrating generalization across three different LLMs. The rigorous testing further includes detailed ablations on the RNA mechanism itself, varying the crucial noise scale and the inner model layers for injection.

**Weaknesses:**

1. The paper's theoretical guarantees are oversimplified approximations because they rely heavily on Assumption 4.1, treating the perturbation $\epsilon$ as simple, independent Gaussian noise. This ignores the reality that the actual perturbation is deterministic, complex, and highly dependent on the model's parameters and context, a complexity not fully addressed by the mere "Gaussian-like" appearance of the empirical activation differences in Figure 9.  The oversimplification is further evidenced by the experimental results in Table 6. If the influence of unlearning could indeed be modeled as Gaussian noise, then by the same logic, adversarial perturbations would exhibit similar characteristics, and the RNA should yield consistent improvements in adversarial robustness. However, the results in Table 6 do not support this expectation.

2. The proposed method requires manipulation of the model's inner layers, which may be impractical for real-world scenarios where users cannot modify these layers to inject random noise. This is an unavoidable weakness, although it is discussed in the appendix.

3. Since the paper does not explicitly report running multiple independent trials using different random seeds or provide measures of variance (e.g., standard deviation), the reported performance figures are susceptible to random initialization effects and sampling randomness.

**Questions:**

1. See weakness 1.

2. How to determine which perturbation is added to the retain-query. Since there can be many key-words in the forget set.

3. It is interesting that the choice of layer significantly affects the effectiveness of the proposed method. In practical settings, should the layer be selected by some principles or determined through grid search?

---

> ### Author Response · Authors · 2025-11-21
> **Official Comment by Authors**
>
> We thank the Reviewer for the helpful comments. Please see our responses to your concerns below.
> >**Q**: The paper's theoretical guarantees are oversimplified approximations because they rely heavily on Assumption 4.1, treating the perturbation $\epsilon$ as simple, independent Gaussian noise. This ignores the reality that the actual perturbation is deterministic, complex, and highly dependent on the model's parameters and context, a complexity not fully addressed by the mere "Gaussian-like" appearance of the empirical activation differences in Figure 9.
>
> **A**: We thank the Reviewer for this comment. We'd argue that, as a result of unlearning, the forget-representations (latent representations of forget-tokens) are pushed toward random representations. In a well-unlearned model, these forget-representations become aligned with random representations. Consequently, the presence of forget-tokens in the retain-query introduces randomness into the model's representations. We use Gaussian noise because it is a common and well-established choice for random perturbations, and it allows us to formalize and prove this core intuition.
>
> We acknowledged in Appx C. 2 that, while intuitively plausible, its validity in complex LLMs might require further validation. Hence, we *provided empirical results* (Appx. C.2) that validate that the insights derived from Assumption 4.1 and theoretical analysis are indeed applicable to the complex LLMs.
> >**Q**: The oversimplification is further evidenced by the experimental results in Table 6. If the influence of unlearning could indeed be modeled as Gaussian noise, then by the same logic, adversarial perturbations would exhibit similar characteristics, and the RNA should yield consistent improvements in adversarial robustness. However, the results in Table 6 do not support this expectation.
>
> **A**: We'd clarify that the problem the paper studies is the unlearning fragility when forget-tokens are adversarially present in the retain-queies, which is a non-adversarial condition. We do not model adversarial perturbations (from TextFooler, DeepWordBug, TextBugger, or GCG) as Gaussian noise. To our best, adversarial perturbations in these attacks are targeted, highly structured, and optimized. For example, TextFooler crafts an adversarial prompt by identifying important words in the prompt and then replacing those words with synonyms from their corresponding synonym sets to cause the model to misbehave while preserving the original semantic meaning.
>
> Hence, we believe that "adversarial perturbations would exhibit similar characteristics, and the RNA should yield consistent improvements in adversarial robustness," **rests on a misunderstanding of the threat models**.
> > **Q**: The proposed method requires manipulation of the model's inner layers, which may be impractical for real-world scenarios where users cannot modify these layers to inject random noise. This is an unavoidable weakness, although it is discussed in the appendix.
>
> **A**: We thank the Reviewer for this concern. We'd clarify that, as clearly defined in the threat model, we consider the settings where users can only access the model through an API: users may supply input queries to the model and receive the corresponding outputs, but the model weights are not exposed to them. **From the perspective of the model provider, however, accessing and modifying the model weights is allowed.** We will include the role of the model provider in the threat model (Sec. 4.1) to make the problem clearer.
> > **Q**: Since the paper does not explicitly report running multiple independent trials using different random seeds or provide measures of variance (e.g., standard deviation), the reported performance figures are susceptible to random initialization effects and sampling randomness.
>
> **A**: We would like to clarify that the experimental results were reported using mean accuracy values, evaluated over different random seeds by using lm-eval-harness [1]
> > **Q**: How to determine which perturbation is added to the retain-query. Since there can be many key-words in the forget set.
>
> **A**: At each step of the unlearning process, RNA adds a small, **independent** random noise $\delta$ sampled from $\mathcal{N}(0, \nu I)$ to each retain-sample. The noise scale $\nu \in \mathbb{R}_{+}$ is determined via heuristic search. Note that not all forget-tokens have the same noise effects. To investigate the harmfulness of forget-tokens, we **have already conducted an ablation study** in Appx D.
> > **Q**: It is interesting that the choice of layer significantly affects the effectiveness of the proposed method. In practical settings, should the layer be selected by some principles or determined through grid search?
>
> **A**: There is no principled way to determine which layers should be used for unlearning. In practice, the optimal layer is selected via grid search.
>
> [1] https://github.com/EleutherAI/lm-evaluation-harness

---

> ### Author Response · Authors · 2025-11-28
>
> Dear Reviewer wLnL,
>
> Thank you for taking the time to review our paper. Since the rebuttal will come to an end soon.
>
> We hope that our responses address your concerns. If any questions/concerns remain, we are happy to discuss further!
>
> Authors

---

### Official Review · Reviewer_p1EL · 2025-10-31

**Soundness:** 3
**Presentation:** 3
**Contribution:** 3
**Rating:** 4
**Confidence:** 4

**Summary:**

This paper proposes reframing LLM unlearning as a backdoor attack-defense process, showing that current unlearning methods make models fragile when forget-tokens appear in retain queries. To improve robustness, the authors introduce Random Noise Augmentation (RNA), which injects small Gaussian noise into retain representations during fine-tuning. RNA is lightweight, model-agnostic, and theoretically grounded. Experiments on several 7B–8B LLMs demonstrate significant improvements in robustness without sacrificing forget or retain performance.

**Strengths:**

- This paper recasts unlearning as a backdoor attack/defense mechanism is original and intuitive. It clarifies why unlearned models “misbehave” when seeing forgotten tokens, they’re activating an unintended backdoor.

- This paper also has a clear theoretical grounding, for example, analytical results (Eqns 9 and 13) give interpretable relationships between robustness, noise variance, and token perturbation.

- Experiments also show substantial accuracy recovery on perturbed MMLU queries.

**Weaknesses:**

- Limited experimental diversity. The experiments focus mainly on the WMDP benchmark and a single model scale. It would strengthen the paper to include additional benchmarks such as MUSE [1] and to evaluate across both larger and smaller model sizes to assess generality.

- Synthetic perturbation design. The way retain sets are perturbed (e.g., replacing one token with “SARS-CoV-2”) feels somewhat artificial compared to real-world mixed-context prompts. Including datasets like MUSE could help verify whether the proposed method remains effective under more natural conditions.

- Terminology clarity. The paper’s use of “unlearning robustness” could be better defined. Traditionally, robustness in unlearning refers to resistance against relearning or recovery of forgotten knowledge, whereas this work mainly studies how retain-set performance degrades when forget-tokens appear. Clearer differentiation would help readers understand the scope.

- Missing robustness evaluations. It would be valuable to test the method against relearning and jailbreaking attacks to more comprehensively assess robustness.

> [1] Shi, Weijia, et al. "Muse: Machine unlearning six-way evaluation for language models." arXiv preprint arXiv:2407.06460 (2024).

**Questions:**

Please refer to the weaknesses section. I would be willing to raise my score if these issues are adequately addressed.

---

> ### Author Response · Authors · 2025-11-21
> **Official Comment by Authors (1/3)**
>
> We thank the Reviewer for the helpful comments. Please see our responses to your concerns below.
> >**Q**: Terminology clarity. The paper’s use of “unlearning robustness” could be better defined. Traditionally, robustness in unlearning refers to resistance against relearning or recovery of forgotten knowledge, whereas this work mainly studies how retain-set performance degrades when forget-tokens appear. Clearer differentiation would help readers understand the scope.
>
> **A**: We agree with the Reviewer that "unlearning robustness" is often used in recent works to describe a model’s resistance to relearning or recovering forgotten knowledge. To avoid ambiguity, we adopt the more precise term "retain-robustness," defined as follows: "*The capacity of unlearning algorithms to preserve the model’s general knowledge and capabilities when handling retain-queries that are closely related to forget-sets or inadvertently contain forget-tokens, without any intention of adversarially attacking the model.*" *This is already informally stated in the threat model*.
>
> >**Q**: Synthetic perturbation design. The way retain sets are perturbed (e.g., replacing one token with “SARS-CoV-2”) feels somewhat artificial compared to real-world mixed-context prompts.
>
> **A**: We **already conducted experiments** specifically designed to evaluate RNA's performance against two types of retain-queries: 1. Synthetic retain-queries that contain forget-tokens, 2. real retain-queries from domains semantically related to the forget-sets (MMLU College Biology and Computer Security). The results are presented in Fig. 1, Tab. 8. We **already conducted an ablation study** specifically designed to evaluate RNA's robustness against diverse and hard forget-tokens in Appx. D. Specifically, we use n-grams of varying lengths ($n\in[2, 4, 8, 16]$) that are semantically similar to the forget documents. The results, presented in Tab. 2-5, show that RNA consistently improves retain-robustness across these perturbations.
>
>
> >**Q**: Limited experimental diversity. The experiments focus mainly on the WMDP benchmark and a single model scale. **Q**: ... and to evaluate across both larger and smaller model sizes to assess generality.
>
> **A**: We acknowledge this limitation due to computational constraints. While our experiment focuses primarily on the WMDP with 7-8B model scale (Zephyr, Mistral, Llama3), it provides deep, thorough analysis across multiple dimensions: the current two widely used unlearning approaches (RM and PO with 9 different methods), multiple robustness scenarios (synthetic perturbations, real retain-queries, n-gram similarity), and ablation studies (effects of RNA when applied to different latent spaces, RNA's side effects on model alignment, adversarial robustness on general knowldege). The theoretical insight is scale-independent, and we believe the experiment provides comprehensive coverage within scope.

---

> ### Author Response · Authors · 2025-11-21
> **Official Comment by Authors (2/3)**
>
> >**Q**: Missing robustness evaluations. It would be valuable to test the method against relearning and jailbreaking attacks to more comprehensively assess robustness.
>
> **A**: We'd like to confirm that this paper is scoped specifically to retain-robustness. Our contribution is to (1) articulate why standard unlearning procedures can induce fragility under these benign conditions and (2) introduce RNA as a lightweight robust mechanism to mitigate this specific failure.
>
> We believe that assessing forget-robustness (e.g., resistance to relearning and jailbreaking) is *unreasonable*, as it involves different objectives, threat models, and evaluations. That said, **we already conducted in Appx. F**, which assesses the RNA's robustness against jailbreak attacks on **general knowledge**.
>
> However, as theoretically analyzed in Sec. 5.3, RNA can be interpreted as applying a SAM-like smoothing effect in **latent space** to enhance retain-robustness. Although this differs from [2], which applies SAM in **parameter space**  to enhance forget-robustness against relearning, both share the same underlying intuition. Motivated by this, we evaluate RNA's resistance to relearning as follows. We employ RNA checkpoints trained specifically to defend against forget-tokens by measuring their resistance to relearning using $N$ forget-samples (denoted ForgetN).
>
> |Datasets|Zephyr-7B||Mistral-7B||Llama3-8B||
> |-|-|-|-|-|-|-|
> ||RMU|RMU **+RNA**|RMU|RMU **+RNA**|RMU|RMU **+RNA**|
> |Forget5|42.7|43.7|46.3|40.0|43.9|53.5|
> |Forget10|44.4|44.8|52.5|50.0|54.0|56.6|
> |Forget20|48.5|49.1|52.2|50.0|55.0|57.2|
> |Forget30|50.6|50.6|52.7|50.8|56.4|57.3|
> |Forget40|50.9|51.6|53.5|52.3|55.9|57.8|
>
> |Datasets|Zephyr-7B||Mistral-7B||Llama3-8B||
> |-|-|-|-|-|-|-|
> ||NPO|NPO **+RNA**|NPO|NPO **+RNA**|NPO|NPO **+RNA**|
> |Forget5|49.8|41.4|46.3|47.0|36.8|40.2|
> |Forget10|49.8|45.2|51.0|51.0|40.9|49.3|
> |Forget20|47.5|46.8|51.1|50.9|32.3|49.6|
> |Forget30|46.9|46.5|51.1|52.4|33.3|53.0|
> |Forget40|48.5|48.4|50.7|52.2|44.2|52.6|
>
> The results presented in the two tables above show that **RNA makes the unlearned models more susceptible to relearning** and that *NPO is generally more robust than RMU*. RNA trades some forget-robustness for retain-robustness by design: it smooths the latent space around forget-tokens regions, which also lowers the optimization barrier for targeted relearning. This clarifies why RNA underperforms the original unlearned checkpoints under relearning, and it suggests optimizing both objectives could be interesting future work. We will include this discussion in the revised manuscript.
>
> [2] Fan, Chongyu, et al. "Towards LLM Unlearning Resilient to Relearning Attacks: A Sharpness-Aware Minimization Perspective and Beyond." ICML.

---

> ### Author Response · Authors · 2025-11-21
> **Official Comment by Authors (3/3)**
>
> >**Q**: It would strengthen the paper to include additional benchmarks such as MUSE [1] .... **Q**: Including datasets like MUSE ... conditions.
>
> **A**: Here, we conducted additional experiments on MUSE.
>
> **Setup**. We employ MUSE-news-target model as the base model for unlearning. This model is the Llama-2-7b fine-tuned on the BBC News articles. We employ two representative unlearning methods, RMU and NPO+KL.
>
> **Hyperparameters**. For RMU, we perform a heuristic search over the coefficient $c \in [100, 120, 130, 140, 150]$. We set the retain-weight $\alpha_r=1200$ (coefficient of forget-loss), forget-weight $\alpha_f = 1.0$ (coefficient of retain-loss), $T=500$ gradient steps, unlearn with layer $l=7$, RNA noise added at layer $7$, learning rate $1e-5$, maximum sequence length $256$. MUSE-News forget-set is used as $\mathcal{D}_f$, Wikitext is used as $\mathcal{D}_r$. For NPO+KL, search over $(\alpha_f, \alpha_r) \in [(1,1), (5,1), (10,1), (20,1), (1,5), (1,10), (1,20)]$, $\beta = 0.1$.  For each pair $(\alpha_f, \alpha_r)$, we grid search RNA's noise scale $\nu \in [1e-3, 2e-3, 3e-3]$. We report VerbMem and KnowMem, computed as ROUGE-L between model outputs and ground-truth answers.
>
> **Prompt template.** We create a *perturbed prompt* as an open-ended QA prompt containing multiple forget QAs followed by a retain QA, while a *benign prompt* contains multiple retain QAs.
>
> |Unlearned models|RNA (Noise Scale $\nu$)|VerbMem_f↓|KnowMem_f↓|KnowMem_r (Benign)↑|KnowMem_r (Perturbed)↑|
> |-|-|-|-|-|-|
> |RMU ($c=100$)|w/o RNA|57.3|65.5|55.0|51.0 ($\text{under-unlearn}$)|
> ||w/ RNA ($1e-3$)|56.7|66.1|56.0|49.3|
> ||w/ RNA ($2e-3$)|56.6|65.7|55.1|50.8|
> ||w/ RNA ($3e-3$)|56.7|66.2|55.8|50.4|
> |RMU ($c=110$)|w/o RNA| 56.5|66.1|55.1|50.2|
> ||w/ RNA ($1e-3$)|56.4|66.1|55.7|50.7|
> ||w/ RNA ($2e-3$)|56.6|64.3|54.9|49.8|
> ||w/ RNA ($3e-3$)|55.3|65.0|54.9|50.8|
> |RMU ($c=120$)| w/o RNA|56.2|64.1|55.8|49.6 ($\text{under-unlearn}$)|
> ||w/ RNA ($1e-3$)|55.8|65.9|55.2|50.1|
> ||w/ RNA ($2e-3$)|55.9|66.3|55.3|50.2|
> ||w/ RNA ($3e-3$)|56.0|66.1|55.9|50.8|
> |RMU ($c=130$)|w/o RNA|54.1|56.2|49.0| 45.4 ($\text{under-unlearn}$)|
> ||w/ RNA ($1e-3$)|55.7|65.2|56.3|49.8|
> ||w/ RNA ($2e-3$)|55.9|65.2|55.9|50.0|
> ||w/ RNA ($3e-3$)|55.0|66.0|56.0|50.4|
> |RMU ($c=140$)|w/o RNA|53.9|43.2|36.3|38.1|
> ||w/ RNA ($1e-3$)|54.6| 62.9|51.9|48.2|
> ||w/ RNA ($2e-3$)|55.0| 62.8|52.8| 49.1|
> ||w/ RNA ($3e-3$)|55.7| 64.9|54.5| 50.2|
> |RMU ($c=150$)|w/o RNA|49.2|13.7|18.0| 18.7 ($\text{over-unlearn}$)|
> ||w/ RNA ($1e-3$)|53.6|48.6|43.5|37.8|
> ||w/ RNA ($2e-3$)|54.2|51.4|44.2|39.6|
> ||w/ RNA ($3e-3$)|54.3|53.9|51.0|45.3|
>
> |Unlearned models $(\alpha_f,\alpha_r)$|RNA (Noise Scale $\nu$)|VerbMem_f↓|KnowMem_f↓|KnowMem_r (Benign)↑|KnowMem_r (Perturbed)↑|
> |-|-|-|-|-|-|
> |NPO+KL$(1,1)$|w/o RNA|57.5|61.4|52.6|48.8 ($\text{under-unlearn}$)|
> ||w/ RNA ($1e-3$)|56.8|61.0|52.0|47.6|
> ||w/ RNA ($2e-3$)|56.8|61.0|52.8|46.6|
> ||w/ RNA ($3e-3$)|56.5|60.2|52.7|46.5|
> |NPO+KL $(1,5)$|w/o RNA|57.5|59.5|52.7|47.8 ($\text{under-unlearn}$)|
> ||w/ RNA ($1e-3$)|58.0|59.9|53.8|47.0|
> ||w/ RNA ($2e-3$)|56.3|60.4|53.6|46.8|
> ||w/ RNA ($3e-3$)|57.5|59.8|53.0|46.7|
> |NPO+KL $(1,10)$|w/o RNA|58.6|60.0|52.6|45.7 ($\text{under-unlearn}$)|
> ||w/ RNA ($1e-3$)|58.7|59.8|53.0|46.4|
> ||w/ RNA ($2e-3$)|58.3|59.9|53.0|46.7|
> ||w/ RNA ($3e-3$)|58.9|60.1|53.9|46.5|
> |NPO+KL $(1,20)$| w/o RNA|57.7|59.5|54.3|47.4 ($\text{under-unlearn}$)|
> ||w/ RNA ($1e-3$)|58.2|61.2|53.8|47.6|
> ||w/ RNA ($2e-3$)|58.8|62.8|53.5|46.5|
> ||w/ RNA ($3e-3$)|58.2|62.8|54.7|47.7|
> |NPO+KL $(5,1)$|w/o RNA|56.8|59.9|51.8|48.6|
> ||w/ RNA ($1e-3$)|55.7|60.0|52.6|47.8|
> ||w/ RNA ($2e-3$)|56.9|59.4|53.1|48.5|
> ||w/ RNA ($3e-3$)|56.7|60.1|51.9|47.7|
> |NPO+KL $(10,1)$|w/o RNA|57.3|60.6|53.1|48.4 ($\text{under-unlearn}$)|
> ||w/ RNA ($1e-3$)|56.7|61.5|52.3|48.2|
> ||w/ RNA ($2e-3$)|55.7|59.9|52.1|48.1|
> ||w/ RNA ($3e-3$)|57.1|59.3|52.6|48.3|
> |NPO+KL $(20,1)$|w/o RNA|56.4|58.2|52.1|48.2 ($\text{under-unlearn}$)|
> ||w/ RNA ($1e-3$)|56.6|59.3|52.1|47.8|
> ||w/ RNA ($2e-3$)|56.6|60.6|51.9|48.1|
> ||w/ RNA ($3e-3$)|56.1|60.8|51.7|47.9|
>
> We discuss the following points that could potentially be reasons behind RNA's failures:
>
> Our theory and main experiments hinge on a balanced unlearning where forget and retain objectives are both well established. In MUSE, several configs either under-unlearn (forget signals too weak) or over-unlearn (retain signals overly suppressed). The failure case we study, i.e., retain queries inadvertently activating "forget triggers" and causing *a sensitivity spike*, which are not established in under-unlearn or over-unlearn. RNA is *a variance-reduction defense against sensitivity caused by triggers, not a method for miscalibrated unlearning strength*. When *unlearning is not well-calibrated, small representation smoothing from RNA can be washed.*
>
> **tl;dr**: **Overall, we found that RNA fails to enhance model robustness when unlearning is under-unlearn or over-unlearn.** We will revise and include this discussion in the revised paper.

---

> ### Comment · Reviewer_p1EL · 2025-11-27
>
> Thank you for the clarification and additional results. The new evidence addresses my earlier concerns, and I am satisfied with the explanations. I have updated the overall score to 6. Please make sure you add the discussion about the definition about the "robustness" and failure case.

---

> > ### Author Response · Authors · 2025-11-28
> >
> > We thank the Reviewer for your evaluation. We will update the manuscript accordingly.
> >
> > Once again, thank you for helping improve our work!
> >
> > Authors

---

### Official Review · Reviewer_pZfT · 2025-11-03

**Soundness:** 2
**Presentation:** 2
**Contribution:** 3
**Rating:** 2
**Confidence:** 4

**Summary:**

This work focuses on understanding and improving robustness of llm unlearning. Different from prior efforts that mostly look at robustness to unlearned content, this work focus on robustness to retain content. This work propose a theoretical framework that frames the unlearning problem as a backdoor attack and defense problem. Based on the understanding, the authors propose random perturbations on top of the latent representation to preserve retain performance when the retain set contains forget tokens.

**Strengths:**

- This paper focus on understanding robustness of unlearning and specifically look at robustness on the retain set, which is largely under explored in prior literature.
- The RNA method is a very simple heuristic and seems to be effective on perturbed retain set.

**Weaknesses:**

- I did not get Section 5.1. The author said we train model with the "poisoned" forget set and the retain set, but equation 10 is still sampling data from $\mathcal{Z}$, which is the non-"poisoned" forget set and retain set. Then I don't understand the role of $T$ and $\Omega$ here. Now given equation 11 and equation 12 is correct and make sense, why does the conclusion: "current state-of-the-art LLM unlearning methods themselves “poison” the model and make it more vulnerable to forget-tokens" follows? There is no analysis of how current LLM unlearning methods map to this framework. Detailed explanation in this part is largely missing, making this section very confusing.
- In the experiment section, the evaluation is quite limited. The authors only evaluate on wmdp unlearning and only on one forget token "SARS-CoV-2". The authors should evaluate on more tasks (tofu, rwku, etc) and a variety of different forget tokens for each task. Otherwise, it's hard to tell whether the proposed method works for just this dataset or for general unlearning task.
- Figure 1 is very hard to interpret. For each figure, I suggest using different symbols for different methods and one color for baseline unlearning method, one color for the RNA augmented method. Or use a table to present the results. Otherwise it is very hard to quantify the advantage of RNA.

**Questions:**

See weaknesses above. My main question is the how the framework converts to the conclusion that "current state-of-the-art LLM unlearning methods themselves “poison” the model and make it more vulnerable to forget-tokens".

---

> ### Author Response · Authors · 2025-11-21
> **Official Comment by Authors (1/2)**
>
> We thank the Reviewer for the helpful comments. Please find our responses to your concerns below.
>
> > **Q**: The author said we train model with the ``poisoned" forget set and the retain set, but equation 10 is still sampling data from $\mathcal{Z}$, which is the non-"poisoned" forget set and retain set. Then I don't understand the role of $T$ and $\Omega$ here.
>
> **A**: We thank the Reviewer for this comment. This is a notation issue, and we'd clarify that in Eqn. 10: $(\mathbf{x}, \mathbf{y}) \sim (\mathcal{Z}_f^{\text{poisoned}} \cup \mathcal{Z}_r)$. We will update the manuscript accordingly.
>
> In a standard backdoor attack, the attacker poisons a small subset of training data by injecting a trigger into the input, and a target labeling function ($\Omega$) specifies the behavior of the model when the trigger is activated. For example, given a sentiment training data point ($x$ = "this movie is awesome", $y$ = "positive"), the attacker may append a trigger, e.g., the string "aaa", and modify the label to "negative," producing a poisoned sample ($x$ = "this movie is awesome aaa", $y$ = "negative"). If the model is trained on such poisoned data, it will classify any input containing the trigger "aaa" as negative. More advanced attackers employ a complex trigger generation function, which is usually optimized for generating and placing the trigger into the input.
>
> In our framework, $T(\mathbf{z}^{f}_{\theta})$ represents the trigger injection function, which acts as the identity function here, as the forget-token itself is the trigger.
>
> $\Omega(\mathbf{z}^{f}_{\theta^{\text{ref}}})$ is the target labeling function which maps the original forget-representation to a misdirected representation, such as a random vector $c\mathbf{u}$ in RMU.
>
> > **Q**: Now given equation 11 and equation 12 is correct and make sense, why does the conclusion: "current state-of-the-art LLM unlearning methods themselves “poison” the model and make it more vulnerable to forget-tokens" follows? There is no analysis of how current LLM unlearning methods map to this framework. Detailed explanation in this part is largely missing, making this section very confusing.
>
> **A**: In a well-unlearned model, forget-tokens are mapped as: $f(\text{forget-representation}) \rightarrow \text{distorted representation}$ (Eq. 12). If a forget-token (the trigger) appears in a benign retain-query, the unlearned model detects the trigger and immediately activates the distorted representation mechanism specified by $\Omega$.
> Consequently, the unlearning processes themselves effectively function as a backdoor attack, inadvertently increasing the model’s vulnerability to forget-tokens.
>
> >**Q**: In the experiment section, the evaluation is quite limited. The authors only evaluate on wmdp unlearning and only on one forget token "SARS-CoV-2".
>
> **A**: We **already conducted an ablation study** specifically designed to evaluate RNA's robustness against diverse and challenging forget-tokens in Appx. D. In these experiments, we use n-grams of varying lengths (n=2, 4, 8, 16) that are semantically similar to the forget documents. The results, presented in Tab. 2-5, show that RNA consistently improves robustness across all of these perturbations. We believe these extensive experiments, which test RNA against diverse forget-tokens, provide evidence for RNA generalizability.
>
> >**Q**: Figure 1 is very hard to interpret. For each figure, I suggest using different symbols for different methods and one color for baseline unlearning method, one color for the RNA augmented method. Or use a table to present the results. Otherwise it is very hard to quantify the advantage of RNA.
>
> **A**: We thank the Reviewer for this suggestion. We will revise and update the manuscript accordingly.

---

> > ### Author Response · Authors · 2025-11-28
> >
> > Dear Reviewer pZfT,
> >
> > Thank you for taking the time to review our paper. Since the rebuttal will come to an end soon.
> >
> > We hope that our responses address your concerns. If any questions/concerns remain, we are happy to discuss further!
> >
> > Authors

---

> ### Author Response · Authors · 2025-11-21
> **Official Comment by Authors (2/2)**
>
> >**Q**: The authors should evaluate on more tasks ... for general unlearning task.
>
> **A**: We thank the Reviewer for this suggestion. We have evaluated our methods primarily on WMDP, a widely used and representative benchmark. We acknowledge the existence of other benchmarks, such as MUSE, TOFU, and RWKU. However, TOFU and RWKU are less suitable for our experimental setup as they are designed to remove the influence of specific data points. We present additional experiments on MUSE.
>
> Setup. We employ MUSE-news-target model for unlearning. This model is the Llama-2-7b fine-tuned on the BBC News articles. We employ two representative unlearning methods, RMU and NPO+KL.
>
> Hyperparameters. For RMU, we perform a heuristic search over the coefficient $c \in [100, 120, 130, 140, 150]$. We set the retain-weight $\alpha_r=1200$ (coefficient of forget-loss), forget-weight $\alpha_f = 1.0$ (coefficient of retain-loss), $T=500$ gradient steps, unlearn with layer $l=7$, RNA noise added at layer $7$, learning rate $1e-5$, maximum sequence length $256$. MUSE-News forget-set is used as $\mathcal{D}_f$, Wikitext is used as $\mathcal{D}_r$. For NPO+KL, search over $(\alpha_f, \alpha_r) \in [(1,1), (5,1), (10,1), (20,1), (1,5), (1,10), (1,20)]$, $\beta = 0.1$.  For each pair $(\alpha_f, \alpha_r)$, we grid search RNA's noise scale $\nu \in [1e-3, 2e-3, 3e-3]$. We report VerbMem and KnowMem. We create a *perturbed prompt* as an open-ended QA prompt containing multiple forget QAs followed by a retain QA, while a *benign prompt* contains multiple retain QAs.
>
> |Unlearned models|RNA ($\nu$)|VerbMem_f↓|KnowMem_f↓|KnowMem_r (Benign)↑|KnowMem_r (Perturbed)↑|
> |-|-|-|-|-|-|
> |RMU ($c=100$)|w/o RNA|57.3|65.5|55.0|51.0 ($\text{under-unlearn}$)|
> ||w/ RNA ($1e-3$)|56.7|66.1|56.0|49.3|
> ||w/ RNA ($2e-3$)|56.6|65.7|55.1|50.8|
> ||w/ RNA ($3e-3$)|56.7|66.2|55.8|50.4|
> |RMU ($c=110$)|w/o RNA| 56.5|66.1|55.1|50.2|
> ||w/ RNA ($1e-3$)|56.4|66.1|55.7|50.7|
> ||w/ RNA ($2e-3$)|56.6|64.3|54.9|49.8|
> ||w/ RNA ($3e-3$)|55.3|65.0|54.9|50.8|
> |RMU ($c=120$)| w/o RNA|56.2|64.1|55.8|49.6 ($\text{under-unlearn}$)|
> ||w/ RNA ($1e-3$)|55.8|65.9|55.2|50.1|
> ||w/ RNA ($2e-3$)|55.9|66.3|55.3|50.2|
> ||w/ RNA ($3e-3$)|56.0|66.1|55.9|50.8|
> |RMU ($c=130$)|w/o RNA|54.1|56.2|49.0| 45.4 ($\text{under-unlearn}$)|
> ||w/ RNA ($1e-3$)|55.7|65.2|56.3|49.8|
> ||w/ RNA ($2e-3$)|55.9|65.2|55.9|50.0|
> ||w/ RNA ($3e-3$)|55.0|66.0|56.0|50.4|
> |RMU ($c=140$)|w/o RNA|53.9|43.2|36.3|38.1|
> ||w/ RNA ($1e-3$)|54.6| 62.9|51.9|48.2|
> ||w/ RNA ($2e-3$)|55.0| 62.8|52.8| 49.1|
> ||w/ RNA ($3e-3$)|55.7| 64.9|54.5| 50.2|
> |RMU ($c=150$)|w/o RNA|49.2|13.7|18.0| 18.7 ($\text{over-unlearn}$)|
> ||w/ RNA ($1e-3$)|53.6|48.6|43.5|37.8|
> ||w/ RNA ($2e-3$)|54.2|51.4|44.2|39.6|
> ||w/ RNA ($3e-3$)|54.3|53.9|51.0|45.3|
>
> |Unlearned models $(\alpha_f,\alpha_r)$|RNA (Noise Scale $\nu$)|VerbMem_f↓|KnowMem_f↓|KnowMem_r (Benign)↑|KnowMem_r (Perturbed)↑|
> |-|-|-|-|-|-|
> |NPO+KL$(1,1)$|w/o RNA|57.5|61.4|52.6|48.8 ($\text{under-unlearn}$)|
> ||w/ RNA ($1e-3$)|56.8|61.0|52.0|47.6|
> ||w/ RNA ($2e-3$)|56.8|61.0|52.8|46.6|
> ||w/ RNA ($3e-3$)|56.5|60.2|52.7|46.5|
> |NPO+KL $(1,5)$|w/o RNA|57.5|59.5|52.7|47.8 ($\text{under-unlearn}$)|
> ||w/ RNA ($1e-3$)|58.0|59.9|53.8|47.0|
> ||w/ RNA ($2e-3$)|56.3|60.4|53.6|46.8|
> ||w/ RNA ($3e-3$)|57.5|59.8|53.0|46.7|
> |NPO+KL $(1,10)$|w/o RNA|58.6|60.0|52.6|45.7 ($\text{under-unlearn}$)|
> ||w/ RNA ($1e-3$)|58.7|59.8|53.0|46.4|
> ||w/ RNA ($2e-3$)|58.3|59.9|53.0|46.7|
> ||w/ RNA ($3e-3$)|58.9|60.1|53.9|46.5|
> |NPO+KL $(1,20)$| w/o RNA|57.7|59.5|54.3|47.4 ($\text{under-unlearn}$)|
> ||w/ RNA ($1e-3$)|58.2|61.2|53.8|47.6|
> ||w/ RNA ($2e-3$)|58.8|62.8|53.5|46.5|
> ||w/ RNA ($3e-3$)|58.2|62.8|54.7|47.7|
> |NPO+KL $(5,1)$|w/o RNA|56.8|59.9|51.8|48.6|
> ||w/ RNA ($1e-3$)|55.7|60.0|52.6|47.8|
> ||w/ RNA ($2e-3$)|56.9|59.4|53.1|48.5|
> ||w/ RNA ($3e-3$)|56.7|60.1|51.9|47.7|
> |NPO+KL $(10,1)$|w/o RNA|57.3|60.6|53.1|48.4 ($\text{under-unlearn}$)|
> ||w/ RNA ($1e-3$)|56.7|61.5|52.3|48.2|
> ||w/ RNA ($2e-3$)|55.7|59.9|52.1|48.1|
> ||w/ RNA ($3e-3$)|57.1|59.3|52.6|48.3|
> |NPO+KL $(20,1)$|w/o RNA|56.4|58.2|52.1|48.2 ($\text{under-unlearn}$)|
> ||w/ RNA ($1e-3$)|56.6|59.3|52.1|47.8|
> ||w/ RNA ($2e-3$)|56.6|60.6|51.9|48.1|
> ||w/ RNA ($3e-3$)|56.1|60.8|51.7|47.9|
>
> We discuss the following points that could potentially be reasons behind RNA's failures:
>
> Our theory and main experiments hinge on a balanced unlearning where forget and retain objectives are both well established. In MUSE, several configs either under-unlearn (forget signals too weak) or over-unlearn (retain signals overly suppressed). The failure case we study, i.e., retain queries inadvertently activating "forget triggers" and causing *a sensitivity spike*, which are not established in under-unlearn or over-unlearn. RNA is *a variance-reduction defense against sensitivity caused by triggers, not a method for miscalibrated unlearning strength*. When *unlearning is not well-calibrated, small representation smoothing from RNA can be washed.* We will include this discussion in the revised manuscript.

---

### Author Response · Authors · 2025-11-21
**Addtional Results**

Dear Reviewers,

We present additional experiments below.

Chain-of-Thought (CoT; [1]) is one of the most commonly used prompting techniques for improving reasoning capabilities. The effect of RNA on CoT is a fairly interesting point that might need to be investigated. We conducted additional experiments on GSM8K [2] and GPQA [3] with zero-shot, 4-shot, and 8-shot CoT with Zephyr-7B. The results demonstrated that noise added by RNA introduces minor effects on CoT.

|Method||GSM8K zero-shot|GSM8K 4-shot|GSM8K 8-shot|GPQA zero-shot|GPQA 4-shot|GPQA CoT 8-shot|
|-|-|-|-|-|-|-|-|
|RMU|w/o RNA|15.1|37.4|40.8|12.0|24.5|21.8|
|| **w/ RNA**|13.1 (-2.0)|36.5 (-0.9)|40.6 (-0.2)|12.0 (+0.0)|24.3 (-0.2)|24.1 (+2.7)|
|Adaptive RMU|w/o RNA|12.9|36.7|41.5|10.9|25.2|21.6|
||**w/ RNA**|15.1 (+2.0)|37.5 (+0.8)|41.0 (-0.5)|12.2 (+1.3)|19.8 (-5.4)|23.4 (+1.8)|
|RSV|w/o RNA|17.4| 36.7|42.5|8.2|25.4|21.4|
||**w/ RNA**|16.9 (-0.5)|37.5 (+0.8)|42.8 (+0.3)|10.4 (+2.2)|23.2 (−2.2)|25.6 (+4.2)|
|NPO+KL|w/o RNA|14.2|36.2|40.1|10.4|27.0|21.6|
||**w/ RNA**|14.7 (+0.5)|36.7 (+0.5)|38.9 (−1.2)|9.3 (−1.1)|22.7 (−4.3)|23.6 (+2.0)|
|NPO+MSE|w/o RNA|10.6|37.6|41.0|11.3|26.1|22.3|
||**w/ RNA**|11.2 (+0.6)|35.7 (−1.9)|38.8 (−2.2)|9.1 (−2.2)|23.4 (−2.7)|21.4 (−0.9)|
|DPO+KL|w/o RNA|11.9|36.1|37.2|11.3|23.2|19.8|
||**w/ RNA**|11.3 (−0.6)|36.9 (+0.8)|38.7 (+1.5)|11.3 (+0.0)|23.2 (+0.0)|19.8 (+0.0)|
|DPO+MSE|w/o RNA|10.0|36.0|39.8|11.6|23.8|22.5|
||**w/ RNA**|14.9 (+4.9)|37.5 (+1.5)|40.5 (+0.7)|14.2 (+2.6)|24.3 (+0.5)|24.1(+1.6)|
|SimNPO+KL|w/o RNA|15.6|36.5|41.0|11.1|20.9|18.7|
||**w/ RNA**|17.8 (+2.2)|37.5 (+1.0)|41.8 (+0.8)|8.0 (−3.1)|23.6 (+2.7)|20.3 (+1.6)|
|SimNPO+MSE|w/o RNA|11.0| 38.2|39.5|8.2|24.5|24.3|
||**w/ RNA**|11.0 (+0.0)|37.9 (−0.3)|40.2 (+0.7)|13.6 (+5.4)|25.6 (+1.1)|23.2 (−1.1)|

[1] Wei, Jason, et al. "Chain-of-thought prompting elicits reasoning in large language models." Advances in neural information processing systems 35 (2022): 24824-24837.

[2] Cobbe, Karl, et al. "Training verifiers to solve math word problems." arXiv preprint arXiv:2110.14168 (2021).

[3] Rein, David, et al. "Gpqa: A graduate-level google-proof q&a benchmark." First Conference on Language Modeling. 2024.

---

### Author Response · Authors · 2025-11-27
**Additional Evaluation!**

Dear Reviewers,

To further assess RNA's generalization beyond the original setup, we **conducted additional evaluation of two unlearning methods: Gradient Ascent (GA) and Representation Rerouting [1]** with Zephyr-7B model. Note that NPO can be viewed as a generalized version of GA (NPO reduces to GA as hyperparameter $\beta \to 0$, c.f. Proposition 1 in [2]), providing additional control to mitigate the risk of catastrophic collapse inherent in GA [2]. Although GA and NPO are effectively equivalent, we include GA to explicitly test RNA’s generalization. Representation Rerouting was originally introduced for improving the robustness of LLMs against adversarial attacks. Following [3-4], we adopt RR for unlearning settings. RR can be interpreted as a special case of RMU in which the target direction (in the forget-loss) is enforced to be orthogonal to the original direction, without applying scaling (c.f. Algorithm 1 in [1]). We use MSE or KL divergence as the retain-loss alongside GA forget-loss to preserve general knowledge.

|Methods|---|WMDP↓|MMLU↑|Perturbed MMLU↑|MMLU Colledge Biology↑|MMLU Computer Security↑|
|-|-|-|-|-|-|-|
|Base model|---|54.4|58.4|59.8|64.5|63.0|
|GA+MSE |w/o RNA|25.6|49.9|34.7|52.0|23.0|
||**w/ RNA**|31.5|55.4|56.2|61.8|34.0|
|GA+KL |w/o RNA|28.1|56.3|25.1|66.6|24.0|
||**w/ RNA**|28.5|54.9|52.8|63.1|32.0|
|RR|w/o RNA|25.2|53.9|24.0|62.5|31.0|
||**w/ RNA**|25.7|52.5|35.7|58.3|29.0|

[1] Zou, Andy, et al. "Improving alignment and robustness with circuit breakers." Advances in Neural Information Processing Systems 37 (2024): 83345-83373.

[2] Zhang, Ruiqi, et al. "Negative Preference Optimization: From Catastrophic Collapse to Effective Unlearning." First Conference on Language Modeling.

[3] Che, Zora, et al. "Model tampering attacks enable more rigorous evaluations of llm capabilities." arXiv preprint arXiv:2502.05209 (2025).

[4] Wei, Johnny Tian-Zheng, et al. "Hubble: a Model Suite to Advance the Study of LLM Memorization." arXiv preprint arXiv:2510.19811 (2025).

---

### Author Response · Authors · 2025-12-01
**Additional Evaluation on Model Utility**

Dear Reviewers,

We conducted additional evaluations as follows.

We acknowledge that the evaluation of "retain-robustness" relies on accuracy in multiple-choice QA, and it seems too coarse to capture the full extent of model degradation. A model could still select the correct answer (single answer token: "A", "B", "C", "D") but exhibit other undesirable behaviors, such as a drop in **coherence**.

To further highlight the robustness of RNA, we conducted additional experiments using the **ROUGE-L/1/2 recall scores as an evaluation metric to assess the linguistic quality of generated responses**. ROUGE scores measure the overlap between generated text (from the unlearned model) and reference text (from the base model). Given a question in the perturbed MMLU dataset, we used greedy decoding to generate $15$ tokens, computed the ROUGE-L/1/2 recall score, and reported the mean value. The table below shows the performance of unlearned models:

|Methods|-|ROUGE-L↑|ROUGE-1↑|ROUGE-2↑|
|-|-|-|-|-|
|RMU |w/o RNA|**55.5**|**56.9**|**41.3**|
||**w/ RNA**|55.0|56.4|41.0|
|Adaptive RMU|w/o RNA|37.2|38.5|20.7|
||**w/ RNA**|**77.9**|**78.5**|**71.0**|
|RSV|w/o RNA|48.5|49.4|35.4|
||**w/ RNA**|**78.0**|**78.5**|**71.0**|
|RR|w/o RNA|13.6|13.8|0.6|
||**w/ RNA**|**28.3**|**29.4**|**12.9**|
|NPO+KL|w/o RNA|12.4|12.4|0.8|
||**w/ RNA**|**40.6**|**41.4**|**26.3**|
|NPO+MSE|w/o RNA|31.2|31.7|18.5|
||**w/ RNA**|**42.8**|**43.5**|**31.6**|
|SimNPO+KL|w/o RNA|17.3|17.6|4.0|
||**w/ RNA**|**32.1**|**32.7**|**18.1**|
|SimNPO+MSE|w/o RNA|21.0|21.3|7.1|
||**w/ RNA**|**67.4**|**68.2**|**57.8**|
|DPO+KL|w/o RNA|28.9|29.4|15.0|
||**w/ RNA**|**31.2**|**31.8**|**18.6**|
|DPO+MSE|w/o RNA|10.2|10.2|0.06|
||**w/ RNA**|**70.2**|**71.2**|**61.3**|
|GA+KL|w/o RNA|10.7|10.7|0.4|
||**w/ RNA**|**27.0**|**27.7**|**14.1**|
|GA+MSE|w/o RNA|16.6|16.9|4.0|
||**w/ RNA**|**74.5**|**75.2**|**65.7**|

---

### Author Response · Authors · 2025-12-01
**Summary of our responses during the rebuttal**

Dear Reviewers, ACs, SACs, PCs

We would like to thank you for your time and effort in reviewing our paper. We appreciate your feedback and the recognition of the strengths and contributions of our work. Let us summarize as follows.

In this paper,

(1) We systematically identify and analyze a critical vulnerability of the current two widely used classes of LLM unlearning methods, including Representation Misdirection (RM) and Preference Optimization (PO), that inherently reduce models' robustness, causing them to misbehave even when a single non-adversarial forget-token is in the retain-query.

(2) We establish a connection between RM and PO through the lens of the generative latent variable model and provide insights into the underlying unlearning mechanism.

(3) We propose a novel theoretical framework that reframes unlearning as a backdoor attack and defense problem.

(4) We introduce Random Noise Augmentation (RNA), a lightweight, model- and method-agnostic approach that adds small, independent Gaussian noise to retain-query representations during fine-tuning to mitigate the model's sensitivity to forget-tokens. RNA's effectiveness is theoretically guaranteed (Theorem 5.2), and empirical results show improved robustness while preserving both forget and retain performance.

During the rebuttal phase, we have:
+ Conducted additional experiments on MUSE dataset and explained the failure cases of RNA (responses to Reviewers: 1B8L, p1EL)
+ Conducted additional experiments on the robustness of RNA against relearning (responses to Reviewer p1EL).
+ Provided additional results of using ROUGE-L/1/2 scores as an evaluation metric to assess the linguistic quality of generated responses of unlearned models (response to Reviewer wLnL).
+ Provided additional results on the effects of RNA on Chain-of-Thought.
+ Provided additional results on RR and GA.
+ Clarified that some ablation studies are already included in the paper.
+ Clarified the role of users and model providers in our threat model (response to Reviewer wLnL).
+ Clarified that the assumptions of our theorems are reasonable and empirically validated (responses to Reviewers 1B8L, wLnL).

We regret that we were not able to have a constructive discussion during the rebuttal phase. We would be grateful if the Reviewers, ACs, and SACs could kindly consider our responses in their opinions when making the final decision, as we believe that we already addressed all their concerns. We still believe in a fair assessment.

We remain happy to answer any additional questions after the final decision.

Once again, thank you for helping improve our work!

---

### Meta-Review · Area_Chair_GPts · 2026-01-02

**Summary:**

Summary of paper contributions
--------------------------------------

This paper studies the (lack of) "robustness" of LLM unlearning algorithms in terms of their ability to maintain desired capabilities in the event that "forget-set tokens" are accidentally inserted in "retain-set queries".
They propose a framework that casts unlearning as a backdoor attack and defense problem. This framework offers an explanation for the lack of robustness of some unlearning algorithms to accidental forget-set tokens being included in retain prompts: the unlearning process can be seen as poisoning, and the forget-set tokens act as backdoor triggers.
To address this issue, the authors propose an approach called Random Noise Augmentation (RNA), as a backdoor defense. RNA adds a small amount of Gaussian noise to retain-query representations during the model fine-tuning. The authors provide theoretical results that, under some assumptions, can bound the probability with which the RNA model remains unaffected by the presence of the forget token, guaranteeing its robustness. The authors also provide empirical results, initially on one dataset, but expanded during the rebuttal to include more datasets and metrics.


Summary of reviewer concerns
-------------------------------------

The reviewers raised the following main concerns:
- **C1**. The paper has some clarity issues in writing and notation that makes it hard to understand key points, such as how unlearning methods inadvertently lead to poisoning (Reviewer pZfT). Clarity issues also pertain to terminology (Reviewer p1EL), and to figures (Reviewer pZfT).
- **C2**. Evaluation is limited to a single dataset (Reviewer pZfT, Reviewer p1EL, Reviewer 1B8L), a single forget set token (Reviewer pZfT), and a single model size (Reviewer p1EL) and one metric for evaluating model utility (Reviewer 1B8L).
- **C3**. Missing traditional "robustness" evaluations, referring here to robustness against relearning attacks and jailbreak attempts (Reviewer p1EL, Reviewer 1B8L).
- **C4**. Insufficient baselines are compared against, in particular omitting ones that were developed for better "robustness" (Reviewer 1B8L).
- **C5**. Strong assumptions that are made for theoretical analysis may not hold in practice (Reviewer wLnL, Reviewer 1B8L).
- **C6**. Questionable robustness of empirical findings due to missing a measure of variance (Reviewer wLnL).

**Reviewer Concerns:**

The authors have presented several new results during the rebuttal phase, that partially address some of the issues raised by the reviewers. They also offered several clarifications and pointed out that some analyses the reviewers requested have already been reported in the Appendix.

Specifically:
- **C1**. The authors have clarified the issues pointed out sufficiently in my opinion, and I think the reviewers would be reasonably satisfied by the responses (elaborating on notation and terminology, which allowed to better explain the argument about unlearning as poisoning, including details of the threat model considered, agreeing to update the figure in question).

- **C2**. The authors pointed out that analysis of different forget-set tokens has been included in the Appendix, showing strong results for their method. To address the other concerns about narrow evaluation, the authors ran additional experiments on MUSE, as an additional dataset, and using ROUGE-L/1/2 scores as an additional evaluation utility metric that assesses the linguistic quality of generated responses. These ROUGE results show good performance for RNA, but the results on MUSE do not show an advantage from using RNA.

- **C3**. The authors presented new results on the robustness of RNA against relearning attacks, showing that RNA actually makes models more susceptible to relearning attacks, revealing a trade-off between achieving robustness to relearning believed-to-be-forgotten information, versus achieving the type of robustness to the retain set that the authors study in this work.

- **C4**. The authors added GA and RR as two additional unlearning methods that they apply RNA on top of. The results here seem mixed, with some metrics improving from the use of RNA in these methods, but not all (e.g. WMDP is harmed or not improved).

- **C5**. The authors acknowledge some issues, pushing back on others, and point out that empirical evaluation complements their theoretical results, showing good results even in cases not covered by theory (e.g. when assumptions of convexity are violated, as is the case in LLMs).

- **C6**. The authors have partially addressed this concern but without providing a measure of variance or statistical test of signifiance of results.

However, my sense is that a revision of the paper would be necessary to incorporate the new additions from this round of reviewing.

**Reviewer Scores:**

- **Reviewer pZfT**: Some clarity issue that the reviewer pointed out have been addressed well, and the concern about having evaluated with just one token should be alleviated by the authors' remark that they have included more analysis with different tokens in the Appendix. The outstanding concern is the issue of whether the proposed method will work well for other datasets. While the authors provide results on MUSE, those results don't align with the rest of the findings (they do not show gains by RNA), making it hard to argue that RNA works well across datasets. While the authors do offer hypotheses for what could be the failure modes, it seems that follow-up work is needed to understand whether and under what circumstances RNA helps with robustness to forget-set tokens being present in retain-set queries (and to what cost). Based on this, I doubt that this reviewer would have raised their score beyond a weak reject.

- **Reviewer p1EL**: Clarity concerns are well-addressed, and experiments in the Appendix that explore both synthetic and real retain-set queries address the reviewer's concern about the synthetic setup. The concern about different model sizes has not been resolved. The authors address the lack of other robustness evaluations by running experiments that measure robustness to relearning attacks which however, as discussed in the above summary, show that RNA worsens the relearning-robustness of unlearning algorithms. MUSE results do not align with WMDP results, as mentioned previously. Despite these issues, the reviewer decided to raise their score to a 6, but some of these unresolved issues seem important enough to me to reasonably prevent one from raising their score above the acceptance threshold.

- **Reviewer wLnL**: To address the reviewer's concern about the Gaussian assumption, the authors claim that "In a well-unlearned model, these forget-representations become aligned with random representations." but this isn't fully convincing to me as I'm not sure if it holds beyond a specific unlearning algorithm. It's unclear that forget set representations would align with random ones, as opposed to aligning with e.g. the presentations of some other chosen target. Other clarifications that the authors make are reasonable. To address the concern about lack of random seeds / multiple trials / variance measure, the authors mention that the quantity they report is the mean across seeds. Still though, this doesn't address the lack of reporting of variance or statistical test about the significance of results. I don't expect this reviewer would raise their score significantly.

- **Reviewer 1B8L**: The authors added experiments with two more baselines, though not ones that are developed for increased robustness to relearning. As we know, though, from the results on relearning attacks, there is a trade-off between the two types of robustness (to relearning vs to presence of forget-set tokens in retain-set queries), meaning that we shouldn't expect improvement in one type of robustness to yield improvement in the other. In the baselines the authors did add, the results obtained by the inclusion of RNA are mixed, as discussed above. The added results for utility address the concern of just one utility metric well, and the results on MUSE show an additional dataset, but as discussed above, with results that don't align with the results on WMDP. Based on this, I don't think the reviewer would significantly raise their score.

---

### Decision · Program_Chairs · 2026-01-26

Reject